# Organization of the human intestine at single-cell resolution

John W. Hickey[1,10], Winston R. Becker[2,10], Stephanie A. Nevins[2,10], Aaron Horning[2], Almudena Espin Perez[3], Chenchen Zhu[2], Bokai Zhu[1], Bei Wei[2], Roxanne Chiu[2], Derek C. Chen[2], Daniel L. Cotter[2], Edward D. Esplin[2], Annika K. Weimer[2], Chiara Caraccio[1], Vishal Venkataraaman[1], Christian M. Schürch[1,4], Sarah Black[1], Maria Brbić[5,6], Kaidi Cao[5], Shuxiao Chen[7], Weiruo Zhang[3], Emma Monte[2], Nancy R. Zhang[7], Zongming Ma[7], Jure Leskovec[5], Zhengyan Zhang[8], Shin Lin[9], Teri Longacre[1], Sylvia K. Plevritis[3], Yiing Lin[8], Garry P. Nolan[1✉], William J. Greenleaf[2✉] & Michael Snyder[2✉]

The intestine is a complex organ that promotes digestion, extracts nutrients, participates in immune surveillance, maintains critical symbiotic relationships with microbiota and affects overall health[1]. The intestine has a length of over nine metres, along which there are differences in structure and function[2]. The localization of individual cell types, cell type development trajectories and detailed cell transcriptional programs probably drive these differences in function. Here, to better understand these differences, we evaluated the organization of single cells using multiplexed imaging and single-nucleus RNA and open chromatin assays across eight different intestinal sites from nine donors. Through systematic analyses, we find cell compositions that differ substantially across regions of the intestine and demonstrate the complexity of epithelial subtypes, and find that the same cell types are organized into distinct neighbourhoods and communities, highlighting distinct immunological niches that are present in the intestine. We also map gene regulatory differences in these cells that are suggestive of a regulatory differentiation cascade, and associate intestinal disease heritability with specific cell types. These results describe the complexity of the cell composition, regulation and organization for this organ, and serve as an important reference map for understanding human biology and disease.

The human adult intestinal system is a complex organ that consists of approximately 7 m of small intestine and 2 m of large intestine. This system completes the digestive process that begins in the oral cavity and stomach, first absorbing water and small-molecule nutrients (such as sugars, monovalent ions and amino acids) in the small intestine, then accumulating larger molecules such as fibre in the large intestine, which serves as an anaerobic fermentation chamber enabling the breakdown and absorption of by-products and the synthesis, often through alimentary gut microbiota, and absorption of other nutrients such as vitamins[1].

The small intestine is phenotypically heterogeneous, comprising three morphologically distinct regions—the duodenum, jejunum and ileum[2]. The large intestine can be partitioned into the ascending, transverse, descending and sigmoid regions. Each of these anatomical regions contains an immense diversity of phenotypically and morphologically distinct cell types. Epithelial, stromal and immune cells, each comprising multiple cell types, reside throughout the intestine. Immune cells are of particular interest, as they interact with the microbiome and foreign material present in the gut[3]. Although these broad cell types are common to all portions of the intestinal system, specific cell types are known to display locational preferences. For example, Paneth cells populate the small intestine, and enteroendocrine L cells are found primarily in the ileum and large intestine[4,5]. Moreover, these cell types are spatially organized into different 'neighbourhoods' across these intestinal regions, and both the composition of these neighbourhoods and the molecular phenotypes of the underlying cellular types vary in relatively unknown ways across these anatomical regions. These differences in both the composition of functional neighbourhoods and the molecular identity of the cell states that comprise these neighbourhoods define the composition and function of the human intestine.

Here, we map many portions of the intestine at the single-cell resolution using single-nucleus RNA, open chromatin and spatial proteomic imaging technologies. Previous studies have mapped cell types using single-cell RNA-sequencing (scRNA-seq) and have catalogued cell types across the intestine[6]. We extend this research by spatially mapping cells and proteins using co-detection by indexing (CODEX)[7–10] as well

[1]Department of Pathology, Stanford School of Medicine, Stanford, CA, USA. [2]Department of Genetics, Stanford School of Medicine, Stanford, CA, USA. [3]Department of Biomedical Data Science, Stanford School of Medicine, Stanford, CA, USA. [4]Department of Pathology and Neuropathology, University Hospital and Comprehensive Cancer Center Tübingen, Tübingen, Germany. [5]Department of Computer Science, Stanford University, Stanford, CA, USA. [6]School of Computer and Communication Sciences, École Polytechnique Fédérale de Lausanne, Lausanne, Switzerland. [7]Department of Statistics and Data Science, University of Pennsylvania, Pennsylvania, PA, USA. [8]Department of Surgery, Washington University, St Louis, MO, USA. [9]Department of Medicine, University of Washington, Seattle, WA, USA. [10]These authors contributed equally: John W. Hickey, Winston R. Becker, Stephanie A. Nevins. ✉e-mail: gnolan@stanford.edu; wjg@stanford.edu; mpsnyder@stanford.edu

as mapping gene regulatory information using single-cell assay of open chromatin using single-nucleus assay for transposase-accessible chromatin with sequencing (snATAC–seq)[11]. We define the relative abundance of distinct cell types across the intestine, including the enormous complexity of epithelial cells across different intestinal regions, and the organization of cells into different multicellular structural niches. We also map gene regulatory differences in these cells that are suggestive of a regulatory differentiation cascade. These results provide important insights into cell function, regulation and organization for this complex organ and serve as an important reference for understanding human biology and disease.

## Mapping the human intestine

We mapped the cell composition, regulatory information and spatial distribution of single cells across the intestines of multiple donors using single-nucleus RNA-seq (snRNA-seq), which measures nuclear RNA transcripts in individual nuclei; snATAC-seq, which measures open chromatin in single cells; and CODEX, which stains the same tissue section with up to 54 antibody probes against different targets (usually proteins). We analysed eight sections from nine individuals: seven individuals of European ancestry (five male and two female) and two African American individuals (one male and one female). Age ranges were from 24 to 78 years. The eight regions (in order of trajectory from the stomach) were as follows: the duodenum, proximal jejunum, mid-jejunum and ileum from the small intestine, and the ascending, transverse, descending and sigmoid regions of the large intestine.

## Multiplexed imaging of the intestine

To create a spatial map of the intestine across the eight regions, we used CODEX multiplexed imaging, which enables insights into cellular interactions, composition of multicellular tissue units and spatial relationships to the overall function of the intestine[9,10]. We first validated and optimized CODEX staining, imaging and image processing for 16-mm$^2$ sections of fresh-frozen samples on one participant (B001) (Supplementary Figs. 1–3). For the other eight donors, we expanded our CODEX antibody panel by adding and validating 17 intestine-specific markers (Supplementary Information 1 and Supplementary Fig. 4) for a total of 54 antibodies that enabled the spatial identification of 25 cell types[12] (Extended Data Fig. 1c and Supplementary Figs. 5 and 6).

We used the resultant dataset (a total of 2.7 million cells) to compare the cellular composition and organization across the different tissue regions, normalizing to overall cell grouping (Fig. 1a–c). Within the stromal compartment, moving from the small intestine to the colon, we observed a decrease in endothelial cells and an increase in smooth muscle cells (Fig. 1a and Extended Data Fig. 1d). To verify that this was not an artefact of capturing more muscularis externa within samples of the colon compared with the small intestine, we calculated the percentages of all cell types within the four different pathological compartments of the intestine: the mucosa, muscularis mucosa, submucosa and muscularis externa. Indeed, even when comparing all of the cell types found within the muscularis externa, there was still a significant decrease in endothelial cells and an increase in smooth muscle cells (Extended Data Fig. 1e). Thus, not only is there less vasculature more broadly in the colon, but there is less within the muscularis externa and a higher density of smooth muscle cells.

In the immune compartment, we observed a decrease in CD8$^+$ T cells from the small intestine to the colon (Fig. 1b, Extended Data Fig. 1f and Supplementary Fig. 7a–c), consistent with previous observations[13]. Conversely, we observed an increase in the percentage of dendritic cells within the colon compared with in the small intestine that is also seen when examining total cell percentages within the mucosa (Fig. 1b and Extended Data Fig. 1f).

In the epithelial compartment, we observed a decrease in enterocytes, an increase in secretory enterocytes (goblet cells) and CD66$^+$ enterocytes and an absence of Paneth cells when moving from the small intestine to the colon (Fig. 1c, Extended Data Fig. 1g and Supplementary Fig. 7d). We also detected a rare population of CD57$^+$ enterocytes that is enriched within the duodenum compared with in other areas of the intestine (Extended Data Fig. 1h and Supplementary Fig. 7e). These gastric-like cells are enriched in areas of the duodenum within submucosal glands (Extended Data Fig. 1i and Supplementary Fig. 7f).

## Cell type associations with clinical data

We also evaluated cell type changes with donor metadata. M1 macrophage levels had the highest correlation with body mass index (BMI) (Fig. 1d) and were restricted to the mucosa (Fig. 1e). M1 macrophages are pro-inflammatory and have been implicated in chronic inflammatory disease, autoimmunity and problems with wound healing in the intestine[14–16]. Similarly, obesity increases the risk of gastrointestinal disorders[17]. Although the donors did not have histories of gastrointestinal disorders, we found that individuals with a BMI characterized as overweight (25–29.9) have a fivefold increase in M1 macrophages and individuals with obesity (BMI > 30) have an eightfold increase compared with individuals who are normal weight (18.5–25). We also observe decreases in endothelial cells (from 25% to 20%) and CD8$^+$ T cells (from 42% to 25%) in donors with a history of hypertension (Fig. 1f). High pressure due to a lower ratio of total vasculature is expected, but a substantial decrease in CD8$^+$ T cells is surprising (Supplementary Fig. 7a).

## Spatial restriction of immune cells

In addition to cell type composition, cellular density can highlight whether a cell has broad functions over large regions, is spatially restricted for specialized functions or has the need for specific cell–cell interactions. We quantified the local cell density for all cell types (Fig. 1g and Extended Data Fig. 1j,k). Visual inspection suggested that plasma cells (-0.2) with the highest same-cell type density were restricted to specific mucosal areas, followed by CD8$^+$ T cells (-0.37), then M2 macrophages (-0.5), which were diffuse throughout all areas of the intestine (Extended Data Fig. 1l). Indeed, M1 macrophage density (-0.39) was lower than its M2 counterpart. Quantification of the distribution of each macrophage subset within the different intestinal tissue units indicates a spatial restriction of macrophage subsets (Fig. 1h), suggesting important functional roles in these regions and that other macrophage subtypes among M2 macrophages exist that may also be spatially restricted. In summary, these results suggest an important role for spatial restriction of immune cell subtypes along the length of the intestine.

## Stromal multicellular neighbourhoods

To provide a global view of intercellular interactions, cellular densities and overall multicellular structures of the intestine, we performed cellular neighbourhood analysis[10] (Methods and Extended Data Fig. 1a). This revealed 18 significant multicellular structures with major epithelial, stromal and immune-based neighbourhoods (Fig. 2a,b and Extended Data Fig. 2b–d). Eight neighbourhoods were classified as stromal neighbourhoods and identified major structures within the intestine: micro- and macrovasculature, innervated stroma and smooth muscle, and innate immune hubs within the stroma and smooth muscle areas (Fig. 2a). Only the Smooth Muscle neighbourhood increased moving from the small intestine to the colon, whereas the innervated and innate immune smooth muscle neighbourhoods did not (Extended Data Fig. 2e and Supplementary Fig. 8a). This further suggests that these dense compartmentalized smooth muscle cell areas (Extended Data Fig. 2f) increase within the colon.

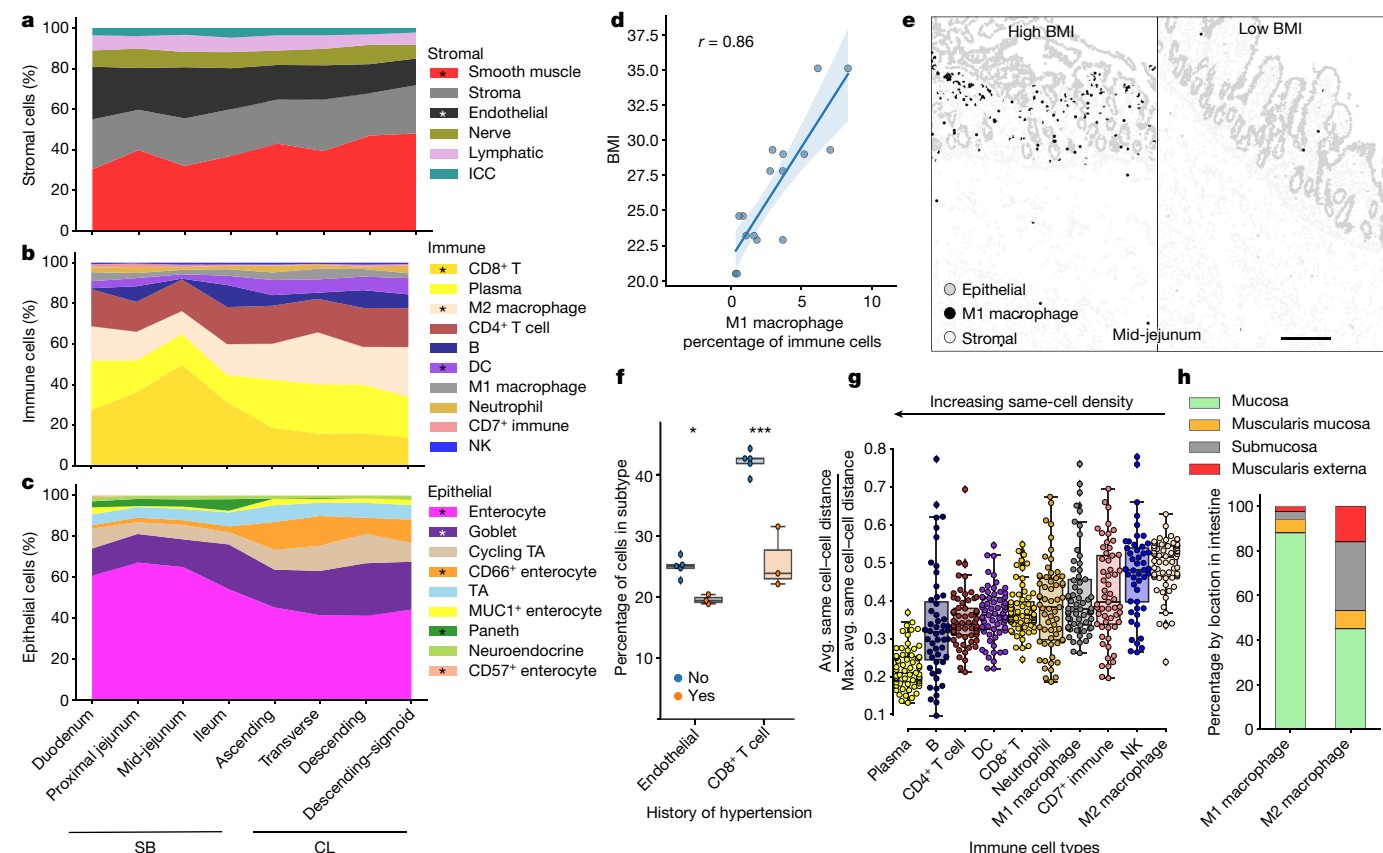

**Fig. 1 | CODEX multiplexed imaging of eight regions from the small intestine and colon to create a single-cell map of the healthy human intestine. a–c**, Cell type percentages from CODEX data averaged across eight donors. Cell types are normalized to the stromal (**a**), immune (**b**) and epithelial (**c**) compartments. Statistical analysis was performed using two-sided *t*-tests comparing the difference in cell type percentage between the small bowel (SB) and the colon (CL); *$P < 0.05$. ICC, interstitial cells of Cajal; NK, natural killer cells; TA, transit-amplifying cells. **d**, The percentage of M1 macrophages within the small bowel and colon for all donors plotted versus donor BMI (Pearson correlation $r = 0.86$). **e**, Cell type maps of the mid-jejunum from representative individuals ($n = 8$ donors) with high or low BMI with M1 macrophages (black) highlighted among stromal (light grey) and epithelial (grey) cell types also shown. Scale bar, 250 μm. **f**, Cell type percentages for endothelial and CD8[+] T cells compared for donors with or without a history of hypertension. Statistical analysis was performed using two-sided *t*-tests; *$P = 0.038$, ***$P = 0.00013$. $n = 3$–5 donors. **g**, Quantification of the same-cell density measured as an average distance of its five nearest same-cell neighbours divided by the maximal possible same-cell distance within the tissue. $n = 64$ tissue sections. **h**, The percentage of macrophage subsets across major intestinal compartments.

## Immune multicellular neighbourhoods

Congruent with our observation of high plasma cell density (Fig. 1g), we observed a Plasma-Cell-Enriched neighbourhood driven by increased density of plasma cells (Fig. 2a). This Plasma-Cell-Enriched neighbourhood also exhibits co-enrichment of CD4[+] T cells and antigen-presenting cells such as dendritic cells and macrophages (Fig. 2a) and is localized within the mucosa lamina propria (Extended Data Fig. 2g–i). These observations are consistent with observations suggesting that secretion or ligand engagement of plasma cells from antigen-presenting cells within the bone marrow can maintain long-term survival in plasma-specific niches[18–20].

Notably, despite their relatively low density in the intestine, CD8[+] T cells were enriched in two major neighbourhoods (Fig. 2a). One neighbourhood (CD8[+] T Cell-Enriched IEL (intraepithelial lymphocyte)) exhibits enrichment of both epithelial cell types and CD8[+] T cells. Thus, the neighbourhood analysis was able to separate the CD8[+] T cells that are intraepithelial lymphocytes, which are critical for rapid immunological responses against infection[21] and maintenance of epithelial integrity[22]. This CD8[+] T Cell-Enriched IEL neighbourhood was one of the neighbourhoods of which the prevalence changed from the small intestine (~30%) to the colon (~3%) (Extended Data Figs. 2c and 3a and Supplementary Fig. 8a). This is the spatial compartment from which we

observed a global decrease in CD8[+] T cell percentage (Fig. 1b) and also significantly decreases with a history of hypertension (Extended Data Fig. 3b). We suggest that the ability of CD8[+] T cells to survive or locate within intraepithelial spaces is negatively affected by hypertension.

The decrease in the CD8[+] T Cell-Enriched IEL neighbourhood is met with an increase in the Plasma-Cell-Enriched neighbourhood within the colon (Extended Data Figs. 2c and 3a). In particular, there is a significant decrease in Plasma-Cell-Enriched neighbourhoods within the ileum as compared to the colon (Supplementary Fig. 7c–d). By contrast, there is also an increase in the Adaptive-Immune-Enriched neighbourhood within the ileum as compared to the colon (Supplementary Fig. 7e–f). Notably, although proximal to the colon, the ileum has the most distinct immune microenvironment from the colon.

CD4[+] T cells contributed to five diverse multicellular neighbourhoods (Fig. 2a). This broad neighbourhood membership is fitting, given that CD4[+] T cells coordinate innate and adaptive immune responses. CD4[+] T cell, B cell and dendritic cell membership defined two different follicle-based structures. The first of these structures, which exists in outer regions of the follicle, exhibited higher enrichment of CD4[+] T cells, whereas inner regions of the follicle were enriched for B cells (Fig. 2a). The presence of the Inner Follicle (that is, the germinal centre) neighbourhood was dependent on a fully mature lymphoid follicle like a Peyer's patch within the image (Fig. 2b). However, the Outer

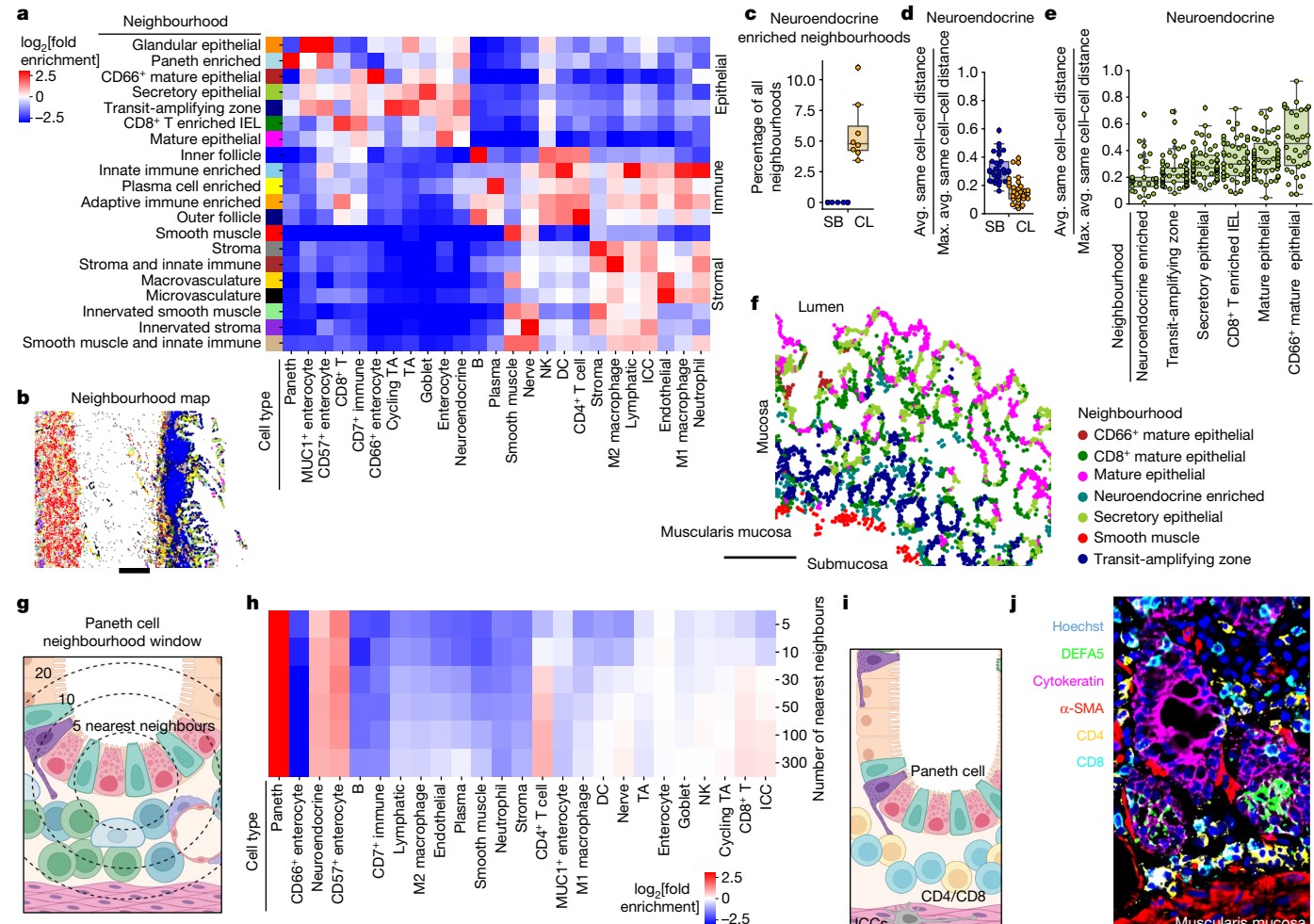

**Fig. 2 | Multicellular neighbourhood analysis of the intestine. a**, Twenty unique intestinal multicellular neighbourhoods were defined by enriched cell types as compared to the overall percentage of cell types in the samples. **b**, An example in which neighbourhoods mapped back to the tissue show overall tissue structures. Scale bar, 0.5 mm. **c**, The percentage of Neuroendocrine-Enriched neighbourhood of all of the neighbourhoods as determined by individually characterizing cellular neighbourhoods by region. $n$ = 8 donors. The box plots show the median (centre line), 25th to 75th percentile (box limits), minimum and maximum values (whiskers), and outliers (points outside 1.5× the interquartile range). **d,e**, Quantification of the same-cell density for neuroendocrine cells compared across the small bowel and colon ($n$ = 32 tissue

sections) (**d**) or the epithelial neighbourhoods as determined by individually characterizing cellular neighbourhoods by region ($n$ = 64 tissue sections) (**e**). Avg., average; max., maximum. **f**, A subset of epithelial neighbourhoods mapped back to a representative magnified region ($n$ = 8 donors) of the mucosa of a transverse colon section. Scale bar, 250 μm. **g,h**, The approach to calculate concentric increasing neighbourhoods around a Paneth cell (**g**) to generate cellular neighbourhoods for Paneth cells at increasing radii (**h**). **i,j**, Schematic (**i**) and CODEX fluorescence data illustrating a representative (1 of 32 sections from 8 donors) magnified portion of the proximal jejunum depicting colocalization of Paneth cells (DEFA5, green) and CD8+ T cells (CD8, cyan) and CD4+ T cells (CD4, yellow) in the intestinal crypt environment (**j**). Scale bar, 50 μm.

Follicle neighbourhood was found across the intestine irrespective of the presence of a fully mature follicle. Out of the 64 tissues that we imaged, 11 had mature follicles that we segmented for comparison (Supplementary Fig. 9a–b). The cell type and neighbourhood compositions differed between each follicle irrespective of location, driven primarily by percentage of dense B cells (Supplementary Fig. 9a), found within the Inner Follicle neighbourhood (Supplementary Fig. 9b). The variation in the ratio of Inner Follicle neighbourhood to Outer Follicle neighbourhood across the intestine suggests a continuum of lymphoid tissues within the intestine[23–26].

Variation in the multicellular composition of cellular neighbourhoods may indicate a core functionality as well as a need for compositional flexibility based on anatomical location. We compared the cell type compositions in the small intestine versus the colon for all neighbourhoods and found that both inner and outer follicle structures are less conserved, whereas stromal neighbourhoods are more conserved (Extended Data Fig. 3c). We observed differences in neighbourhood

composition in Paneth-Cell-Enriched (significantly less abundant in the colon than the small bowel) and Transit-Amplifying-Zone (not different in terms of abundance) neighbourhoods that are both enriched for early epithelial progenitor cells, potentially indicating different crypt microenvironments across the intestine (Extended Data Fig. 2d and Supplementary Fig. 9c,d).

## Intestinal crypt neighbourhoods

As we observed differences in neighbourhood cell type conservation, we performed neighbourhood analysis on each individual region of the intestine separately and then concatenated the results. All aggregated neighbourhoods (Fig. 2a) were identified (Extended Data Fig. 3d). We also identified two unique neighbourhoods: Neuroendocrine-Enriched, which was found only in the colon, and Neutrophil-Enriched, which was found throughout the intestine but enriched in the colon (Fig. 2c and Extended Data Fig. 3e,f).

The Neutrophil-Enriched neighbourhood was characterized by a high density of neutrophils associated with vasculature and innate immune cells, found often within stromal and smooth muscle areas (Extended Data Fig. 3d). The Neuroendocrine-Enriched neighbourhood had a mixture of epithelial and immune cell types enriched (Extended Data Fig. 3d).

The identification of the Neuroendocrine-Enriched neighbourhood suggested differential organization of neuroendocrine cells in the small intestine compared with in the colon. Indeed, neuroendocrine cells were found to be denser within the colon compared with in the small intestine (Fig. 2d). Furthermore, neuroendocrine cells are most dense within the Neuroendocrine-Enriched neighbourhood as compared to neuroendocrine cells in other epithelial neighbourhoods (Fig. 2e), and the density decreases with the maturity of the epithelial cell types that define these epithelial neighbourhoods (Fig. 2e). This suggests that the Neuroendocrine-Enriched neighbourhood represents the colon crypt neighbourhood, which is confirmed by its localization near the muscularis mucosa (Fig. 2f).

Identification of this crypt environment only within the colon and not in the small intestine is consistent with our finding of variation in early epithelial environments in our first neighbourhood analysis (Extended Data Fig. 3c), and Paneth-Cell-Enriched neighbourhoods are observed only within the small intestine (Extended Data Fig. 3e). Indeed, Paneth cells are known to be restricted to the small intestine and also to be enriched within the intestinal crypt. As a consequence, to understand whether the Neuroendocrine-Enriched neighbourhood was similar to the small intestine crypt environment, we examined the neighbours surrounding Paneth cells with increasing window size (Fig. 2g). This analysis revealed that there was a high enrichment of neuroendocrine cells, but it also underscored that interstitial cells of Cajal, CD4[+] T cells and CD8[+] T cells were enriched within the local microenvironment (Fig. 2h–j). This agrees with enrichment of CD4[+] T cells and interstitial cells of Cajal found within the Neuroendocrine-Enriched neighbourhood (Extended Data Fig. 3d).

## Hierarchical structural organization

Multicellular neighbourhood analysis revealed key differences in the structural composition across the intestine as well as in the composition of these neighbourhoods, particularly with relation to the adaptive immune system and the intestinal crypt. However, how these multicellular neighbourhoods interact with one another, and how they are spatially structured in the tissue is unclear. Understanding how multicellular groups are related is key to both defining the hierarchy of tissue organization, as well as defining key functional tissue interfaces.

We investigated higher-order structural organization using several methods. First, we clustered windows of neighbourhood compositions in a manner similar to how we clustered windows of cell types (Fig. 3a) to define neighbourhoods (Fig. 3b). This generated communities of neighbourhoods (Fig. 3c and Extended Data Fig. 4a–c), which we then leveraged to identify major tissue units such as the muscularis mucosa (Fig. 3d and Supplementary Fig. 10). The Paneth-Cell-Enriched neighbourhood was enriched within the Adaptive-Immune-Enriched community (Extended Data Fig. 4a). Indeed, concentric neighbourhood environments surrounding the Paneth-Cell-Enriched neighbourhood also showed colocalization with Outer Follicle and Adaptive-Immune-Enriched neighbourhoods (Extended Data Fig. 4d). These results support the idea that the adaptive immune system forms a conserved niche with the intestinal crypt.

To relate the various levels of spatial organization, we created a hierarchical structure network graph (Fig. 3e and Supplementary Fig. 11). Each level of this graph is connected to the next by its major contributors. Using this intuitive formalism, we can observe many levels of intestinal cell and tissue-structure organization. For example, we observed crosstalk between stromal and smooth muscle cell types and structures, which are in turn isolated from epithelial and immune components that are more entwined with one another (Fig. 3e (red bracket)).

Using this graph structure, we can also observe multilevel relationships between the structures. For example, Paneth cells (Fig. 3e (green circle)) are a rare cell subset (size) and are primarily enriched within Paneth-Cell-Enriched neighbourhoods (Fig. 3e (light blue square)), which are enriched in the Adaptive-Immune-Enriched community (Fig. 3e (orange triangle)) that, in turn, is enriched within the muscularis mucosa tissue unit (Fig. 3e (red diamond)). This relationship can be seen within the tissue, where we see the Adaptive-Immune-Enriched community localized to the bottom of the colon crypt (Fig. 3f).

Visualizing the communities also revealed spatial layering of the intestine, moving from the smooth muscle, stroma and particularly within epithelial areas (Fig. 3f). To formalize these observations of intercommunity-level spatial interactions, we created[27] a spatial context map[27] (Methods) that revealed major structural relationships between communities within the colon (Fig. 3g). For example, moving from left to right in the spatial context map parallels tissue organization in a cross-section of the intestine moving from the muscularis externa to the top of the mucosa (Fig. 3f,g). In brief, the Smooth Muscle community (red triangle) is often found alone (size of circle), indicating its compartmentalization from other communities (Fig. 3g). However, it is found next to the Stromal community (grey triangle), with which it forms an interface (grey and red triangle combination) (Fig. 3g (yellow highlighted edge)). This then forms a trio interface with the Adaptive-Immune-Enriched community (orange triangle). Moving to the right, we observe another trio that involves the Smooth Muscle, Adaptive-Immune-Enriched and Secretory Epithelial communities (Fig. 3g (green box)). This pattern continues across the Plasma-Cell-Enriched community (yellow triangle) and then the Mature and CD66[+] Epithelial community (teal triangle) (Fig. 3g).

We confirmed these community–community interactions using an established method for identifying two-combination community motifs[27] (Methods). Significant associations with just the Adaptive-Immune-Enriched community shared across the intestine demonstrate that there are connections to the Plasma-Cell-Enriched, Smooth Muscle, Secretory Epithelial, Stromal and Follicle communities but not to the Mature Epithelial community (Extended Data Fig. 5b and Supplementary Fig. 12), which aligns with our previous analyses (Fig. 3f,g). Similarly, analysis of other shared significant motifs shows that the Plasma-Cell-Enriched community intersects with both the Secretory Epithelial and Mature Epithelial communities (Extended Data Fig. 5c). We created a spatial context map for just the cells found in the mucosa (Extended Data Fig. 5d–f) and, again, observed a high-frequency intersection between the Plasma-Cell-Enriched, Secretory Epithelial and Transit-Amplifying Zone neighbourhoods (Extended Data Fig. 5f (red box)) with many connections to the Plasma-Cell-Enriched neighbourhood, implicating an important role in overall intestinal tissue structure.

In summary, these results indicate that immune-cell-enriched neighbourhoods have important roles in intestinal tissue organization. Indeed, we found enrichment of cell types moving from the smooth muscle community to the lumen (Fig. 3h and Extended Data Fig. 5g,h), which shows an increase in adaptive immune cells in the base of the crypt, restricted zones of plasma cells, and an increase in innate immune cells and CD8[+] T IELs towards the top of the intestine (Fig. 3h). In conclusion, our hierarchical mapping data further confirm compositional differences in multicellular structures between the small intestine and colon but also highlight conserved multicellular structure interactions and an important distribution of distinct cell types in subregions of the intestine.

## Single-nucleus RNA and chromatin atlas

The CODEX experiments revealed distinct cellular arrangements across intestinal regions, but included only 54 probes, potentially limiting

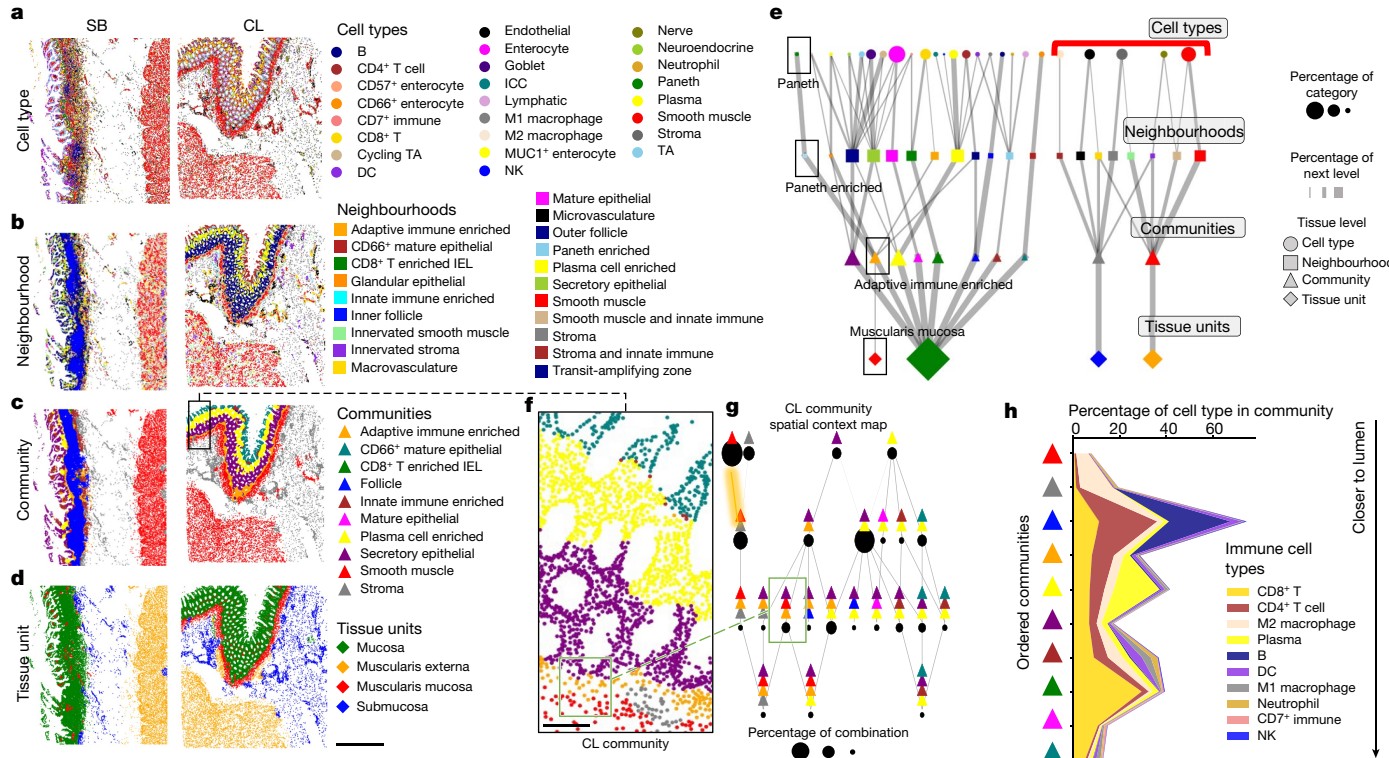

**Fig. 3 | Multilevel hierarchical structural description of the small intestine and colon. a–d**, Representation of multiple levels of hierarchical description: cell type (**a**), multicellular neighbourhood (**b**), community (based on clustering windows of cell neighbourhoods) (**c**) and tissue units (based on clustering communities) (**d**) comparing the small bowel with the colon for two representative tissue sections (from a total of 64 sections from 8 donors). Scale bar, 1 mm. **e**, Tissue hierarchy graph of the multilevel network of the tissue comprised of the different structures. Shapes correspond to structural level (cell type, neighbourhoods, communities, tissue units); colours represent individual categories as indicated in **a–d**; the size of shapes represents the percentage of tissue; and the size of connected lines represents the overall contribution to the next level of the structure when moving down the graph in increasing tissue structural hierarchy. The black rectangles highlight a single

trajectory highlighted within this Article. The red bracket indicates separation of stromal tissue units from the mucosal tissue units. **f**, Magnified mucosal area of a colon community map shown within **c**. Scale bar, 100 μm. **g**, The spatial-context maps of the colon highlighting relationships of communities across the entire sample. This structure is defined by the number of unique communities required to make up at least 85% in a given window. The circles represent the number of cells represented by a given structure. The green rectangle highlights a structure discussed in this Article and maps this structure back to **g**. The colours are as indicated in **c**. **h**, The cell type percentage of immune cells shown for each community ordered in relative order of general increasing proximity to the lumen on the basis of community spatial-context analysis.

both the number and complexity of cell types identified. To overcome these limitations, we performed 10x multiome sequencing analysis of intestinal regions from six donors and separate 10x snRNA-seq and snATAC-seq analysis of intestine regions from three additional donors (Fig. 4a, Methods and Extended Data Fig. 6a, b).

We first annotated all snRNA cells from both the snRNA and multi-ome experiments together, by dividing cells into immune, stromal and epithelial compartments, revealing a total of 10 immune, 16 stromal and 16 epithelial cell types (Fig. 4b–d), and annotated the remaining scATAC cells from three donors separately. Cell types were anno-tated by examining gene expression levels and gene activity scores of known marker genes as well as by labelling the datasets with previously published scRNA-seq data[28] (Methods).

The majority of immune cell types were identified in both the single-nucleus and CODEX data (Supplementary Table 2). To determine which cell types were significantly differentially abundant between locations, we applied two methods that control for the compositional nature of single-cell data[29,30]. We found that CD8+ T cells were more abundant in the small intestine, whereas B cells were more abundant in the colon (Fig. 4f and Extended Data Fig. 7), consistent with the CODEX results (Fig. 1b).

Within the stromal compartment, we annotated eight fibroblast subtypes—interstitial cells of Cajal (*KIT*, *ANO*), glial cells (*SOX10*, *CDH19*,

*PLP1*), neurons (*SYP*, *SYT1*, *RBFOX1*), pericytes (*NOTCH3*, *MCAM1*, *RGS5*), adipocytes (*PLIN1*, *LPL*) and three endothelial cell clusters (Fig. 4b and Extended Data Fig. 6f). Cells with high expression of *MYH11* and *ACTA2* were classified as smooth muscle/myofibroblasts. One of these clus-ters had high expression of *DES*, which we labelled *DES*high and may represent smooth muscle. We also identified fibroblasts with high levels of *WNT* agonists, such as *RSPO3*, and the *BMP* antagonist *GREM1*, which are probably present at the crypts, and fibroblasts with high expression of *WNT5B* and *BMP* transcripts thought to be present at the villi[28] (Extended Data Fig. 6f). Elsewhere, the *ADAMDEC1*high popula-tion has been referred to as S1 fibroblasts, the *WNT5B* and *BMP* signal-ling fibroblasts as S2 fibroblasts, and the *KCNN3*high population as S3 fibroblasts[6,31]. Similar to the immune cells, we observed changes in cell type abundance along the intestine (Fig. 4f and Extended Data Fig. 7). For example, two smooth muscle/myofibroblast clusters were less abundant in the small intestine than in the colon. Conversely, villus fibroblasts and endothelial cells were most abundant in the duodenum and jejunum, less abundant in the Ileum and least abundant in the colon.

As epithelial cells initially clustered on the basis of location (Extended Data Fig. 6c), we subclustered and annotated epithelial cells from different primary locations—duodenum, jejunum, ileum and colon—separately. In each intestinal region, we observed a differentiation trajectory from stem cells to mature absorptive cells (enterocytes). We divided this

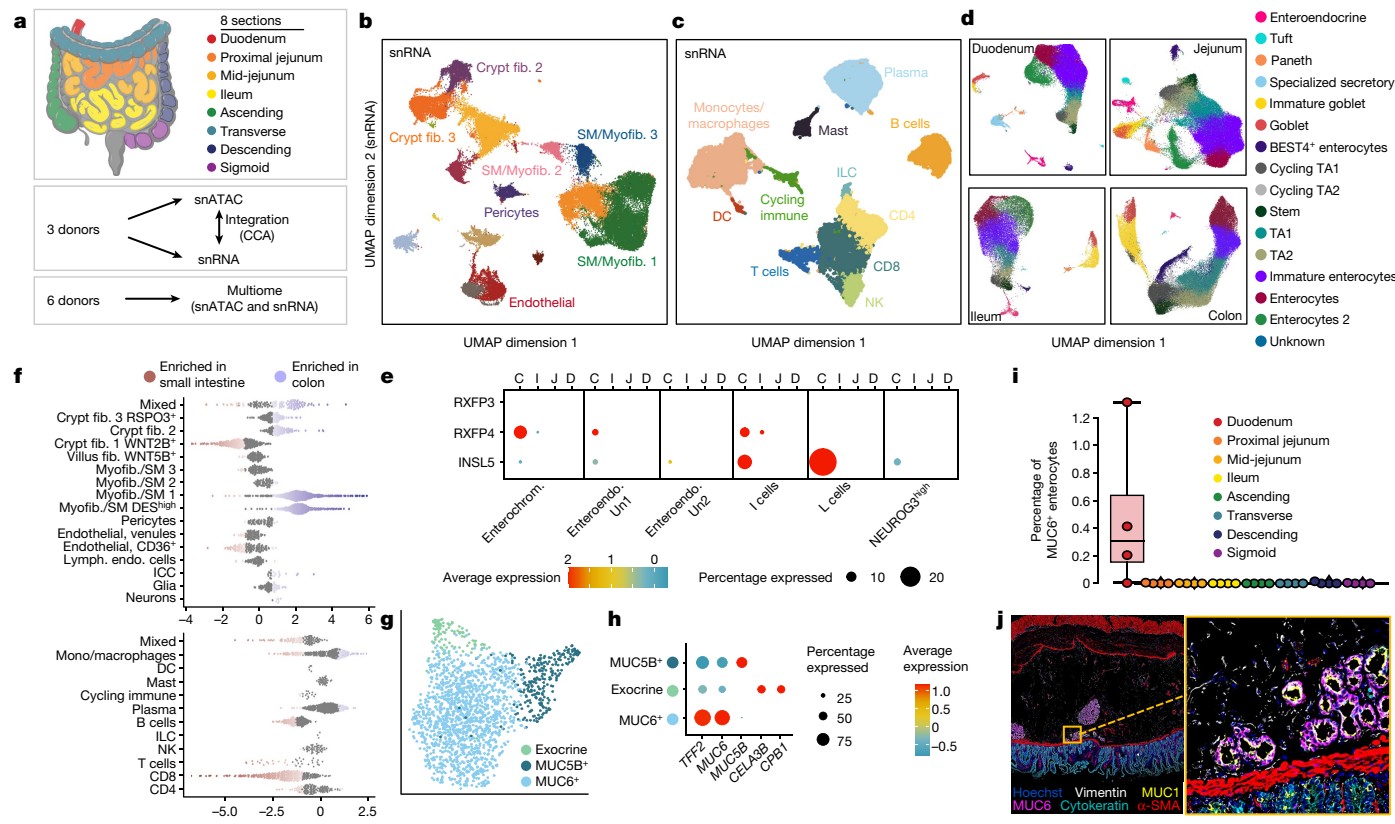

**Fig. 4 | Single-cell atlas of gene expression and chromatin accessibility in the human intestine. a**, Sections of the intestine analysed by separate snRNA-seq and snATAC-seq experiments or multiome experiments. **b,c**, UMAP representation of all snRNA stromal (**b**) and immune (**c**) cells coloured by cell type. DC, dendritic cells; fib., fibroblasts; ILC, innate lymphoid cells; myofib., myofibroblasts; SM, smooth muscle cells. **d**, UMAP representation of snRNA epithelial cells in the four primary regions of the intestine. Jejunum includes both proximal- and mid-jejunum samples. Colon includes samples from the ascending, transverse, descending and sigmoid colon. **e**, Expression of INSL5 and the INSL5 receptor, RXFP4, in different cell types in different regions of the intestine. RXFP4 was expressed in less than 2.5% of all epithelial cell types that were not included in the dot plot. C, colon; D, duodenum; I, ileum; J, jejunum; enterochrom., enterochromaffin cells; enteroendo., enteroendocrine cells.

**f**, Beeswarm plot showing the log-transformed fold change between the small intestine and colon for groups of nearest-neighbour cells from different cell type clusters. Significant changes are indicated in red and blue. Lymph. endo., lymphatic endothelial cells. **g**, Subclustering of specialized secretory cells in **d** coloured by cell type. **h**, The expression of secretory genes in specialized secretory cells defined in **g**. **i**, The percentage of MUC6⁺ enterocytes among all cell types for the four donors imaged using MUC6 antibodies. **j**, Representative CODEX fluorescence image of the duodenum (6 out of 57 markers overlaid) (left). Hoechst (nuclei), MUC6, MUC1 (also found in gland areas), cytokeratin (pan-epithelial), α-SMA (muscle) and vimentin (stromal) staining is shown. Right, the magnified area highlights the gland just below the mucosa in the submucosa. This experiment was independently repeated four times. Scale bars, 500 μm (left) and 50 μm (right).

differentiation trajectory into five cell types using similar annotations to other studies (stem > TA2 > TA1 > immature enterocyte > enterocyte)[28]. However, these cells exist along a continuum and the exact number of cell types and locations of the divisions between cell types is therefore arbitrary, and changing the resolution during clustering results in more or fewer clusters along this trajectory. In addition to absorptive cells, we also observed goblet cells, tuft cells and enteroendocrine cells in all regions of the intestine, while Paneth cells were present only in regions of the small intestine. Consistent with recent reports of BEST4⁺ enterocytes in the small and large intestine[6,32], we observed populations of BEST4⁺ enterocytes in all eight intestinal regions assayed.

To further examine the diversity of enteroendocrine cells along the intestine, we subclustered enteroendocrine cells from all intestinal regions and annotated the clusters based on expression of enteroendocrine marker genes (Extended Data Fig. 6g,h). We identified many known subtypes of enteroendocrine cells, including D cells (SST^high), I cells (CCK^high), K cells (GIP^high), Mo cells (MLN^high), S cells (SCT^high), L cells (GCG^high and PYY^high) and enterochromaffin cells (TPH1^high). Two clusters of enteroendocrine cells that did not express any of these specific markers were labelled as enteroendocrine Un1 and enteroendocrine Un2 in the subclustered dataset. L cells can be further divided on the

basis of expression of INSL5⁺, which is primarily expressed by L cells in the colon. To determine which gut cell types the INSL5⁺ L cells may be signalling, we examined the expression of RXFP4—the cognate receptor for INSL5[33,34]. We found that RXFP4 is primarily expressed by colon enterochromaffin cells and, to a lesser extent, I cells, suggesting that these are the most likely cell types that the INSL5⁺ L cells are signalling (Fig. 4e). Enteroendocrine and enterochromaffin cells were more abundant in the duodenum and jejunum compared with in the colon, and we observed shifts in the fraction of each subtype of enteroendocrine cells along the intestine (Extended Data Fig. 7). For example, D cells were most abundant in the duodenum[35]. These results provide detail on how populations of enteroendocrine and enterochromaffin cells change along the length of the intestine.

Finally, the duodenum also contained an additional cluster of cells that expressed secretory markers, but did not cluster with goblet, tuft, enteroendocrine or Paneth cells. We subclustered these cells (Fig. 4g) and identified one cluster with high expression of MUC5B, one cluster with high expression of MUC6 and TFF2, and one cluster with high expression of the exocrine markers CELA3B and CPB1 (Fig. 4h). The MUC5B⁺ and MUC6⁺ cells are probably different types of mucin-producing cells, with the MUC6⁺ cells probably representing the

cells of the Brunner's glands[36]. Notably, the *MUC5B*+ cells and exocrine cells were primarily present in only one sample in our dataset, making it possible that they are contaminating cells from a different tissue, in which expression of these markers is more common, or a rare cell type in the duodenum (Extended Data Fig. 6i). The *MUC6*+ cells were present in the majority of the samples, so we further validated their presence by labelling cells in CODEX experiments with MUC6 (Fig. 4i,j).

Within the CODEX data, we made the distinction between CD66+ enterocytes and CD57+ enterocytes. When we examined the expression of these markers in the snRNA data, we found that CD66+ enterocytes typically represent mature cell types in the colon (Extended Data Fig. 8a,b). CD57+ cells are much less abundant and *CD57* expression was primarily observed only at low levels in the cluster of MUC6+ cells, consistent with the presence of CD57+ cells in glands in the duodenum in the CODEX data (Extended Data Fig. 8a,b).

## Molecular interactions in nearby cells

The CODEX data enable the assignment of neighbouring cell types and the snRNA-seq data provide RNA expression levels in the corresponding cell types. By combining both data types, we can nominate potential ligand–receptor pairs that may facilitate cell-to-cell interactions. We identified significant pairwise cell type colocalizations and focused on those that were significantly different between the small bowel and colon (Extended Data Fig. 8c and Supplementary Table 2). Cell type pairs involving plasma cells are more colocalized in the descending-sigmoid colon tissue section than in other sections (Extended Data Fig. 8e).

Using snRNA-seq data from these six cell types, we performed differential expression analyses of ligands and receptors (Methods) and identified 48 pairs of ligands and receptors (across 41 cell–cell pairs) that are more expressed in the colon than in the small bowel (Supplementary Table 3). For example, we found that the ligand *SEMA4D* and receptor *MET* were upregulated in plasma cells and TA2, respectively, in colon tissue (Extended Data Fig. 8d), which was not observed in the small intestine (Supplementary Table 3). SEMA4D signalling has been associated with B cell aggregation and long-term survival[37]. The implication of the MET receptor on TA2 cells in the colon, compared with TA2 cells in the small intestine, is further evidenced by RAS and MAPK signalling (Extended Data Fig. 9b, Supplementary Table 4) and upregulation of plexins (Extended Data Fig. 9b), consistent with the CODEX imaging data (Fig. 3f,g and Extended Data Figs. 5a–f and 9c). Using spatial transcriptomics analysis based on Molecular Cartography[38–40] (Methods), we validated that TA cells positive for MET and plasma cells positive for SEMA4D were more colocalized in the colon (colocalization quotient (CLQ) = 1.57). We also validated nine other receptor–ligand interactions involving plasma cells that were more colocalized in the colon (Supplementary Table 5). Consequently, this indicates a potential differential survival signal that maintains the Plasma-Cell-Enriched neighbourhood in the colon.

In addition to colocalization of receptor–ligand pairs, we probed our spatial transcriptomics dataset and evaluated whether other receptor–ligand pairs were more expressed by target cells in the colon than in the small bowel. We validated 15.3% of our predictions (9 out of 58; $P < 0.002$; Methods) that have a matching cell type with snRNA-seq (Methods and Supplementary Table 6). One other example that we examined was the expression of ligand FN1 (fibronectin) in myofibroblasts and its receptor PLAUR (urokinase receptor) in enterocytes (Extended Data Fig. 9d). Overall, these results nominate potential ligand–receptor interactions that mediate specific cell type interactions in distinct regions of the intestine.

## Integration of CODEX and snRNA data

Our multiplexed imaging analysis also enables examination of the cell–cell pairs that are enriched within each neighbourhood or community regardless of tissue location. We used the same approach to calculate cell–cell colocalization in the Follicle community. This resulted in 57 colocalized cell pairs, such as the CD8+ T cell–CD4+ T cell pair, each pair containing a large list of potential receptor–ligand interactions (Extended Data Fig. 9e). To more comprehensively describe potential interactions and expressions within neighbourhoods, we integrated CODEX and snRNA-seq at the single-cell level using Max-Fuse[41], a method that we specifically designed for these challenging integration tasks: linear assignment coupled with graph smoothing and meta cell construction. The overall matching accuracy was 92.1% and cell types segregated within the co-embedding space, and the same cell types from snRNA-seq and CODEX were well blended while showing concordant RNA/protein expression patterns, indicating robust integration performance (Fig. 5a). Using this integration, we calculated the differentially expressed genes (DEGs) among the previously CODEX-defined cellular neighbourhoods with the pair-linked transcriptome information (Extended Data Fig. 9f). In particular, we examined the expression levels of DEGs involved with the follicle neighbourhoods (Fig. 5a (bottom UMAP)) and identified gene pathways that are enriched in these spatial organizations, including B-cell-receptor signalling, T-cell-polarity regulation and tolerance induction of self antigen pathways (Extended Data Fig. 9g). Thus, integrating CODEX and snRNA-seq data revealed specific expression patterns associated with distinct cellular neighbourhoods.

## Regulatory TFs in cells of the intestine

To obtain insights into the factors that control intestinal differentiation, we next investigated potential transcription factors (TFs) that regulate gene expression in intestinal cell types. We first computed ChromVar deviation scores[42] for each cell in our dataset, identifying TF motifs that are associated with chromatin accessibility in different cell types. As many TFs share similar motifs, examining TF expression in conjunction with motif activity helps to identify the specific TFs that are functional in different intestinal cell types. We next identified the TFs with the highest correlation between their gene expression and the chromatin-accessibility activity level of their putative DNA-binding motifs[43] to nominate TFs that directly drive accessibility changes. Across the intestine, this analysis revealed 61 TFs with motif activity that was strongly correlated with expression ($r > 0.5$; Fig. 5b). Broadly, we observed TFs that are active in the secretory lineage and TFs that are active in the absorptive lineage with few TFs that participate in both (for example, KLF4 in colon enterocytes and goblet cells throughout the intestine; Fig. 5b).

This analysis highlights many TFs that are known to have important roles in the intestine. For example, ASCL2, a master regulator of intestinal stem cells[44], exhibited high expression and motif accessibility in stem cells. Other TFs with high expression and motif activity in stem cells include NFIX, NFIC and HNF1B. Within the secretory lineage, POU2F3, a regulator necessary for the development of tuft cells in mice[45], was highly expressed and had high motif activity in tuft cells throughout the human intestine. When comparing the TF footprinting signal around POU2F3 motifs in tuft cells compared with enterocytes, we observed greater accessibility flanking POU2F3 motifs in tuft cells, suggesting that POU2F3 is more likely to be bound to its motif in tuft cells (Extended Data Fig. 10d). Along with POU2F3, RUNX1 and RUNX2 also exhibited high expression and accessibility in tuft cells throughout the intestine, indicating that they may also have a role in tuft cell differentiation or maintenance. Among goblet cells, KLF4, which is required for terminal differentiation into colonic goblet cells in mice[46], exhibited high gene expression and motif activity. Notably, the expression and motif activity of KLF4 was also high in differentiated absorptive epithelial cells (immature enterocytes and enterocytes) in the colon, but not in other regions of the intestine, indicating location-specific regulation.

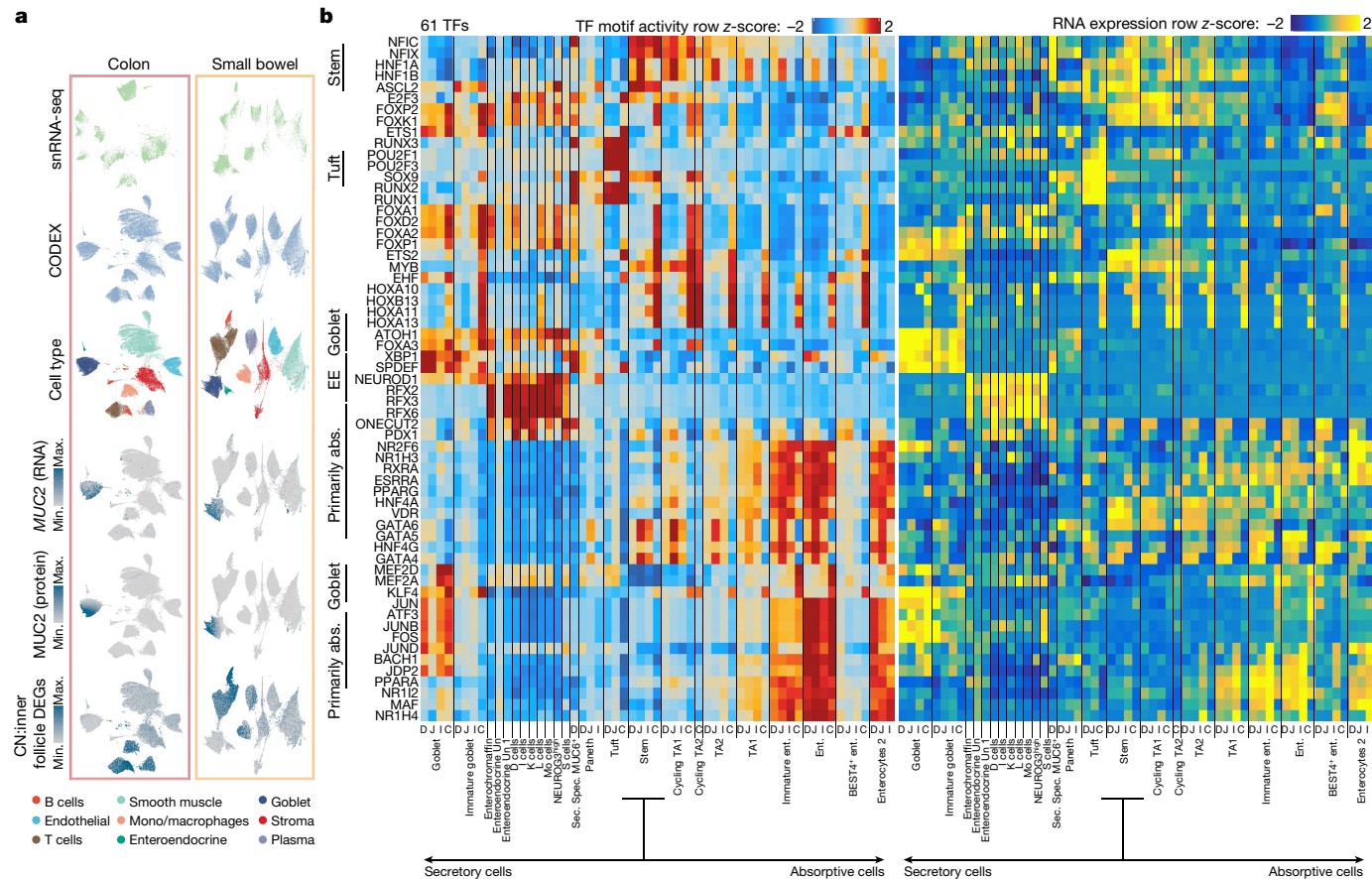

**Fig. 5 | Integration of CODEX multiplexed imaging, snRNA-seq and snATAC-seq reveals genes associated with cellular neighbourhoods and regulatory TFs in the human intestine. a**, snRNA-seq and CODEX integration using MaxFuse for both the small bowel and colon. Integrated cells are coloured by modality of origin, cell types or RNA/protein expression levels. MUC2 expression (protein and RNA) is shown on the integrated plots, as well as the top 10 DEGs of the Inner Follicle neighbourhood as determined by CODEX. Min., minimum. CN, cellular neighbourhood. **b**, TFs of which the integrated gene expression correlated with their motif activity in one region of the intestine. Row z-scores of ChromVar deviation scores are shown on the left and row z-scores of integrated TF expression are shown on the right. Ent., enterocytes. abs., absorptive.

Several regulatory TFs were nominated in goblet cells, including ATOH1 and FOXA3. ATOH1 is necessary for secretory lineage commitment in the mouse intestine[47,48], and here we provide data supporting a similar function in human goblet cells. To provide further evidence that ATOH1 drives accessibility in goblet cells, we compared ATOH1 footprints in goblet cells and enterocytes (Extended Data Fig. 10e). Indeed, we found greater flanking accessibility around ATOH1 motifs in goblet cells, consistent with greater ATOH1 binding in these cells. For FOXA3, there is evidence that FOXA3 leads to goblet cell metaplasia in the lungs[49], and our findings indicate that it probably is an important regulatory TF in the colon. We noted above that goblet cells are more abundant in the colon compared with in the small intestine, but it is unclear what biases cells to differentiate into goblet cells with greater frequency in the colon. Possible causes include differences in signalling in the crypts versus differences in stem cells between the regions. We identified motifs that are enriched in differential peaks between small intestine and colon stem cells and found that FOX motifs are enriched in peaks that are more accessible in colon stem cells (Extended Data Fig. 10a–c). Similarly, FOX TFs have greater motif deviation scores in colon stem cells compared with in small intestine stem cells (Extended Data Fig. 10a–c). As we nominate several FOX TFs as potential regulators of goblet cells, increased FOX activity may partially explain why goblet cells are more abundant in the colon.

Within enteroendocrine/enterochromaffin cells, RFX2, RFX3 and RFX6 exhibited high expression and accessibility. Of these, RFX6 is a proposed regulator of enteroendocrine cell differentiation, and loss of

RFX6 impairs enteroendocrine cell differentiation in mice[50]. Together, these results support previous findings and nominate additional TFs that may be important regulators of distinct intestinal cell types that can vary across the different regions of the intestine (for example, KLF4).

To help to validate these findings, we completed this analysis in the two separate cohorts: the multiome samples and the samples with separate snRNA and snATAC data. To examine TF expression in the latter group, we integrated the snRNA and snATAC data using canonical correlation analysis to align the datasets and assign snRNA data to each snATAC cell[51,52] (Methods). This analysis reproduced many of the findings in the multiome analysis, with 48 out of the 61 TFs originally identified in the multiome analysis reaching the same significance criteria (Extended Data Fig. 10f).

To determine which TFs may drive cell function in stromal and immune cells in the intestine, we performed the same analysis for cells in each of these compartments. Within the immune compartment, this analysis highlighted many TFs that are known to be important for cell type differentiation and maintenance, including GATA2 in mast cells[53] and PAX5 in B cells[54] (Extended Data Fig. 10g). Among stromal cells, we identified TFs associated within specific lineages, including EBF1 in pericytes—which was recently suggested to contribute to pericyte cell commitment[55]—PPARG in adipocytes[56] and SOX10 in glia[57] (Extended Data Fig. 10h). We also nominate potential regulatory TFs in interstitial cells of Cajal, including HAND1, HOXD11 and MEIS1, as well as potential regulatory TFs for different classes of intestinal fibroblasts.

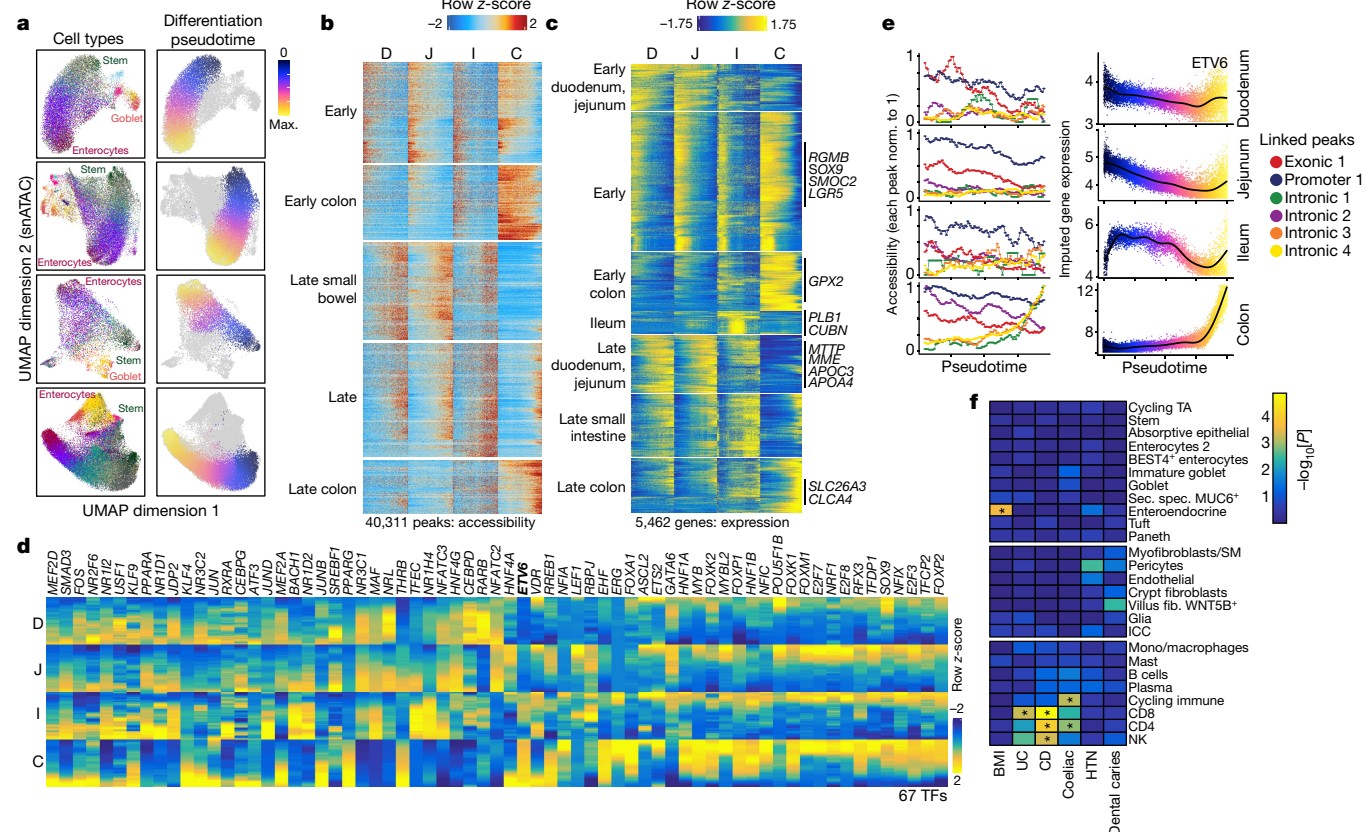

**Fig. 6 | Regulation of differentiation in the human intestine. a**, UMAP projections depicting the cells in the four primary regions of the intestine (duodenum, jejunum, ileum and colon), labelled by cell type (left) and differentiation pseudotime (right). **b,c**, Variable peaks (**b**) and genes (**c**) identified along the absorptive differentiation trajectories. The rows represent the *z*-scores of accessibility for each peak or expression for each gene. The columns represent the position in pseudotime from the start to the end for each section of the intestine. Peaks and genes were *k*-means-clustered and the clusters were labelled on the basis of the dominant time and location where they are most accessible/expressed. **d**, Integrated gene expression of TFs of which the expression is correlated with ChromVar motif activity along the differentiation trajectory. **e**, Accessibility at peaks correlated with the expression of *ETV6* along the differentiation trajectory in each region (left). Each peak is normalized to the maximum accessibility along any of the trajectories. Right, integrated gene expression of *ETV6* along the differentiation trajectory in each region is plotted on the right. Norm., normalized. **f**, Linkage-disequilibrium score regression to identify the enrichment of GWAS SNPs in cell-type-specific marker peaks. Unadjusted coefficient *P* values computed from linkage-disequilibrium score regression are plotted in the heat map. Significance is indicated by an asterisk, as determined by a Bonferroni-corrected coefficient *P* value of <0.05. *P* values for determining significance were adjusted for the number of cell classes tested.

## Absorptive differentiation trajectories

Intestinal stem cells differentiate into mature enterocytes, goblet cells and specialized cell types such as enteroendocrine, tuft and Paneth cells, renewing the epithelial lining approximately every three to seven days[58]. To map the regulatory and gene expression changes that accompany stem cell differentiation into mature enterocytes, we defined differentiation trajectories along this pathway in the single-nucleus data from the duodenum, jejunum, ileum and colon (Fig. 6a and Methods). We next identified regions of variable chromatin accessibility (peaks) and variable gene expression across these four differentiation trajectories. An example of the variable genes is *TMPRSS15*, which encodes the protein that converts trypsinogen to trypsin in the duodenum; its expression gradually increases in more differentiated cells in the duodenum (Extended Data Fig. 10m–p). We clustered these variable peaks and genes to identify sets with shared behaviour (Fig. 6b,c). The resulting clusters include peaks and genes that are open and expressed early in the differentiation pseudotime (for example, in stem cells) in all regions of the intestine, which we denote as early. This cluster includes general markers of intestinal stem cells, including RGMB, SOX9, SMOC2 and LGR5. Other clusters of genes and peaks include those that are predominantly found in differentiated small intestine cells (for example, MTTP, APOA4, APOC3, MME), in undifferentiated

colon (for example, GPX2), in ileum (such as PLB1, CUBN) and in differentiated colon (such as SCNN1B[59], SLC26A3, CLCA4).

To identify both chromatin drivers of cluster-specific regulation and relevant cluster-specific gene expression programs, we computed TF-motif enrichments in each cluster of peaks (Extended Data Fig. 10k) and KEGG pathway enrichment in each cluster of genes (Extended Data Fig. 10l). Groups of peaks accessible late in the differentiation trajectories were enriched for HNF4 and JUN/FOS motifs. As expected, genes primarily expressed late in the differentiation trajectory in the small intestine (late duodenum jejunum cluster) were enriched for multiple metabolic KEGG pathways including fat digestion and absorption and vitamin digestion and absorption.

We next identified TFs of which the gene expression is correlated with the activity of their motifs in each of the four differentiation trajectories. We found 67 TFs with expression correlated with their motif activity (*r* > 0.5) and plotted the integrated gene expression of these TFs (Fig. 6d). Many TFs display similar activity along all four differentiation trajectories. For example, ASCL2, a master regulator in intestinal stem cells, is highly expressed at the beginning of all four trajectories. Other TFs, such as ETV6, exhibit different behaviours in different regions of the intestine. ETV6 is a TF with decreased expression in colorectal cancer compared with in the normal colon, and genetic variation in ETV6 may confer colorectal cancer susceptibility[60]. We found that ETV6 is

more highly expressed in the colon compared with in the small intestine and, in contrast to in the small intestine, ETV6 expression increases in more differentiated cells in the colon.

As an example of examining how genes with unique expression patterns may be controlled, we next investigated which regulatory elements may be responsible for the variable expression of ETV6 in different regions of the intestine. We identified accessibility peaks correlated with ETV6 expression along the differentiation pseudotime in any region of the intestine (Fig. 6e). The correlated peaks most accessible in the small intestine (exonic 1 and promoter 1) became less accessible along the differentiation trajectory, consistent with the decreased expression along the differentiation pseudotime in these regions. Similar behaviour was also observed in the colon, in which these peaks became less accessible along the differentiation trajectory. However, multiple other peaks exhibited increasing accessibility in more differentiated cells only in the colon, and we speculate that these regulatory elements may drive the increased expression of ETV6 in differentiated colon cells. The same logic can be applied to identify regulatory elements that may drive changes in gene expression throughout the intestine. For example, we identified three peaks that are highly correlated with expression of TMPRSS15 and may drive its increased expression in the duodenum (Extended Data Fig. 10p). Taken together, this analysis provides a reference for the regulation of stem cell to enterocyte differentiation across the intestine.

We next tested whether disease heritability is enriched in cell-type-specific marker peaks in intestine cell types using linkage-disequilibrium score regression (Fig. 6f and Methods). We identified a significant increase in heritability for Crohn's disease and coeliac disease in T cell marker peaks, consistent with the importance of T cells in their pathogenesis[61]. We observed the most significant enrichment of heritability for BMI in enteroendocrine cells, suggesting that genetic variation may have an effect on enteroendocrine cells leading to effects on BMI. As a control, we also tested whether heritability of GWAS SNPs was enriched in an unrelated trait (dental caries) and found no cell-type-specific enrichment. These results map important disease traits to specific cell types in the intestine.

## Discussion

Here we show extensive cellular complexity of the intestine including considerable epithelial heterogeneity and new secretory cell subtypes, that the different regions of the intestine have different cell compositions, and that cells are organized into different neighbourhoods that also form communities defined by both distinct areas of epithelial and immune cells. Additionally, the open chromatin regulatory program defined key regulators and differentiation pathways used in the different regions of the intestine. Our study greatly extends previous single-cell studies by combining both spatial proteomic datasets and single-cell RNA and ATAC technologies, and the resulting datasets serve as a useful resource for the scientific community.

Our multiplexed imaging is the first in-depth spatial study of single cells in the intestine, to our knowledge. Many of our spatial analyses revealed important immune cell type organization within the intestine (Supplementary Discussion). We observed a correlation of M1 macrophages with BMI and spatial restriction of macrophage subtypes[17]. Although our donors did not have histories of gastrointestinal diseases, an increase in M1 macrophages may potentially indicate an early stage of gastrointestinal disease progression. We also observed that CD8+ T cells decreased from the small intestine to the colon primarily within the CD8+ T Cell-Enriched IEL neighbourhood. Notably, we also found that CD8+ T cells decrease in donors with a history of hypertension and within the CD8+ T Cell-Enriched IEL neighbourhood. Further investigation of T cell levels within the gut in patients with hypertension may provide cell type mechanisms for associations of hypertension with other diseases such as colon cancer[62,63].

We identified intestinal crypt multicellular neighbourhoods that were co-enriched with adaptive immune cells such as CD4+ and CD8+ T cells in the small bowel and colon. In the small bowel, stem cell crypt areas were identified by Paneth-Cell-Enriched neighbourhoods, whereas, in the colon, the stem cell crypt was identified with increasing neuroendocrine cell density towards the bottom of the crypt. CD4+ T cells were also the most enriched cell type in the Outer Follicle multicellular neighbourhood, which was present at all sites of the intestine regardless of the presence of fully developed Inner Follicle structures across neighbourhoods. This observation suggests that the immune system appears structurally poised along the intestine to generate germinal-centre-focused immune responses locally as needed. Integration of CODEX and snRNA-seq single-cell data revealed differential gene expression across these cellular neighbourhoods and highlighted important gene modules for the immune response in Inner Follicle neighbourhood structures.

We confirmed the adaptive immune and crypt association with hierarchical spatial analysis and further observed discrete immune cell composition zones across the intestine from an adaptive-immune-enriched area at the base of the crypt, then a plasma-cell-enriched area in the middle of the mucosa, with an innate-immune-enriched zone at the top. Indeed, plasma cells had the highest same-cell density and were also found to co-localize with antigen-presenting cells in a multicellular neighbourhood found in all areas of the intestine. Merging the snRNA-seq data and CODEX data showed that plasma cells and transit-amplifying epithelial cells co-localized more in the colon than in the small intestine, and this was validated by spatial transcriptomics. Restriction and layering of the intestine primarily has focused on epithelial subtypes previously, but we highlight that immune cells share a spatial restriction and zonation. This hierarchical view of multiplexed spatial data can serve as a template for other spatial atlas efforts.

This study extends the work of several scRNA gut atlases[6,32]. These previous scRNA datasets examined the diversity of cell types in the human intestine, whereas the integrated snRNA and snATAC dataset provides a detailed single-cell regulatory map of the intestine. We used this dataset to identify TFs in different cell types of which the gene expression is highly correlated with the accessibility of the motif to which they bind. This identified TFs that are known to be master regulators of different cell types, including POU2F3 in tuft cells, ASCL2 in stem cells and RFX6 in enteroendocrine cells, while also nominating many additional TFs that are probably important regulators in their respective cell types. This includes RUNX1 and RUNX2 in Tuft cells and FOXA3 and ATOH1 in goblet cells. Finally, with the inclusion of two donors from under-represented backgrounds, our study also extends single-cell analyses to include samples from such groups.

Within all regions of the intestine, intestinal stem cells differentiate into mature absorptive enterocytes. By integrating gene expression and chromatin accessibility data along the differentiation trajectories in different regions of the intestine, we nominate TFs that exhibit consistent behaviour across absorptive differentiation in all regions of the intestine. Among these are known intestinal stem cell regulators such as SOX9 and ASCL2, which are highly expressed and have high chromatin activity of their binding motifs in cells at the beginning of the differentiation trajectory in all regions of the intestine. We also observed differences in TF dynamics between the trajectories in different regions, such as ETV6, which is highly expressed in differentiated absorptive cells in the colon, but not other regions of the intestine. Notably, ETV6 expression is decreased in colorectal cancer and genetic variation in ETV6 may confer colorectal cancer risk[60]. We speculate that ETV6 is important for normal colon differentiation and its loss may prevent differentiation of colon stem cells. Examining these data also enables us to link specific regulatory elements with the expression of TFs along the trajectory. For the case of ETV6, this analysis identifies one distal and three intronic regulatory elements with similar activity

that are probably responsible for driving expression of ETV6 along the absorptive differentiation trajectory in the colon.

Together, these data provide a detailed atlas that can serve as a valuable reference for future studies of the intestine. This includes contextualizing GWAS studies of intestinal disease, similar to the analysis performed above showing heritability of BMI is enriched in enteroendocrine cells, and severing as a reference when comparing disease state such as cancer to the normal colon[64].

There are several limitations to our study. First, for each patient, we typically analysed a single sample from each intestinal region. Second, CODEX is limited to 54 markers and does not capture the entire breadth of known cell types within the intestine. Third, it is also important to note that, although there was variation of ages, all individuals were above the age of 24, limiting our analysis of the development of the intestine in children. Also, six out of the nine adults analysed were male. Although our study represents two ethnicities, the patterns across a wide range of ethnicities remain to be elucidated. We are also underpowered to ascertain sex differences, which will probably be important given differences in disease risk for male and female individuals[65]. These limitations can be addressed in the future with the investigation of more samples.

In summary, we present a detailed map of the human intestine, and the first multiplexed imaging reference for healthy small intestine and colon. In addition to biological insights, this can serve as an important reference for intestinal diseases (such as inflammatory bowel disease) as well as comparisons with other organisms.

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

# Methods

## Tissue collection and processing

This study complies with all relevant ethical regulations and was approved by the Washington University Institutional Review Board and the Stanford University Institutional Review Board. Human bowel tissues were procured from deceased organ donors. Written informed consent was obtained from the next-of-kin for all donor participants.

Organs were preserved according to surgical protocols to prepare the bowel for transplantation. In brief, a large-bore cannula was placed into the infrarenal aorta before circulatory arrest. Immediately after circulatory cessation, the abdominal viscera was rapidly cooled by flushing ice-cold HTK preservation solution through the aortic cannula and by packing the abdominal cavity with ice. The bowel was kept cold throughout the transport and dissection process until samples could be frozen for long-term storage. Tissue samples collected from the designated bowel sites were preserved by snap freezing in liquid nitrogen for snRNA-seq/scATAC, and embedded and frozen in optimal cutting temperature compound (OCT) for CODEX.

## Array creation

Imaging data were collected from four human donors, each of whom constitutes a dataset. Each dataset includes two arrays of tissues that were imaged together on the same coverslip with four tissues per array: colon (sigmoid, descending, transverse and ascending), and small intestine (ileum, mid-jejunum, proximal jejunum and duodenum). Tissues were individually frozen in OCT moulds and then cut and assembled into arrays of four tissues with known directionality such that a cross-section of each tissue would be achieved cutting the block at once. Arrays were constructed on the cryostat and sectioned at a width of 7 μm.

## Registration of samples with HuBMAP common coordinate framework

We have also registered these blocks within HuBMAP's tissue registration in a common coordinate framework[66]. In brief, male and female 3D reference objects for 11 organs including the small bowel and colon were created using Visible Human Project datasets. Using standard surgical anatomical landmarks used to collect the eight bowel sites, the tissue blocks were registered to the reference objects. The anatomical landmarks used for small bowel segments were as follows: (1) descending duodenum to the right of the pancreas head; (2) 5 cm beyond the ligament of Treitz in the jejunum; (3) 200 cm of the jejunum beyond the ligament of Treitz in the mid-bowel; and (4) 5 cm proximal to the ileocecal valve for terminal ileum. Five centimetres of bowel at each site was collected, representing approximately 20 g of tissue at each site. For the colon, the following landmarks used: (1) the right colon midway between the ileocecal valve and the hepatic flexure; (2) the transverse colon midway between the hepatic and splenic flexures; (3) the left colon midway between the splenic flexure to the appearance of the sigmoid mesentery; and (4) the sigmoid colon midway to the rectosigmoid junction where the taenia coli ceased.

## CODEX antibody conjugation and panel creation

CODEX multiplexed imaging was performed according to the CODEX staining and imaging protocol previously described[8]. Antibody panels were chosen to include targets that identify subtypes of intestinal epithelium and stromal cells, and cells of the innate and adaptive immune system. Detailed panel information is provided in Supplementary Table 7. Each antibody was conjugated to a unique oligonucleotide barcode, after which the tissues were stained with the antibody–oligonucleotide conjugates and we validated that the staining patterns matched the expected patterns already established for immunohistochemistry within positive control tissues of the intestine or tonsil. Similarly, haematoxylin and eosin morphology staining was used to confirm the location of marker staining. First, antibody–oligonucleotide conjugates were tested in low-plex fluorescence assays and the signal-to-noise ratio was also evaluated at this step, then they were tested all together in a single CODEX multicycle.

## CODEX multiplexed imaging

The tissue arrays were then stained with the complete validated panel of CODEX antibodies and imaged[8]. In brief, this entails cyclic stripping, annealing and imaging of fluorescently labelled oligonucleotides complementary to the oligonucleotide on the conjugate. After validation of the antibody–oligonucleotide conjugate panel, a test CODEX multiplexed assay was run, during which the signal-to-noise ratio was again evaluated, and the optimal dilution, exposure time and appropriate imaging cycle was evaluated for each conjugate (Supplementary Table 7). Finally, each array underwent CODEX multiplexed imaging. Metadata from each CODEX run are provided in Supplementary Table 8.

## CODEX data processing

Raw imaging data were then processed using the CODEX Uploader for image stitching, drift compensation, deconvolution and cycle concatenation. Processed data were then segmented using the CODEX Segmenter or CellVisionSegmenter, a watershed-based single-cell segmentation algorithm and a neural network R-CNN-based single-cell segmentation algorithm, respectively. The donor sample from individual B001 was segmented using the CODEX Segmenter (with parameters tuned as described previously[8]), whereas all of the other donor samples were segmented using CellVisionSegmenter. CellVisionSegmenter has been shown to work well with segmenting both dense and diffuse cellular tissues with CODEX data[67]. CellVisionSegmenter is an open-source, pretrained nucleus segmentation and signal quantification software based on the Mask region-convolutional neural network (R-CNN) architecture. Indeed, it was designed and trained on manually annotated images from CODEX multiplexed imaging data within our own group. Consequently, the only parameter that was altered was the growth pixels of the nuclear mask, which we found experimentally to work best at a value of 3. Despite this, no segmentation algorithm does a perfect job of segmentation in cases in which the boundaries identified may capture portions of neighbouring cells, and nuclear segmentation can limit quantified signal that whole-cell segmentation might be able to capture (although also imperfect from lack of consistent cell membrane stains), which has been discussed in more detail in reviews and primary sources of segmentation[67–74]. For this reason, we performed an in-depth analysis of the different data normalization techniques and unsupervised clustering methods for robust identification of cell types in CODEX intestine data. This analysis revealed that there is some segmentation noise that could affect cell type identification if using manual gating, but using $z$-normalization, Vortex or Leiden-based unsupervised clustering, over-clustering the data and manually overlaying resultant cell type clusters to the image results in cell type identification at a much higher fidelity[75].

Both the CODEX Uploader and Segmenter software can be downloaded from GitHub (https://github.com/nolanlab/CODEX), and the CellVisionSegmenter software is available at GitHub (https://github.com/bmyury/CellVisionSegmenter or https://github.com/michael-lee1/CellSeg). After the upload, the images were again evaluated for specific signal: any markers that produced an untenable pattern or a low signal-to-noise ratio were excluded from the ensuing analysis. Uploaded images were visualized in ImageJ (https://imagej.nih.gov/ij/).

## Cell type analysis

B001 and B004 cell type identification was performed according to methods developed previously[75]. In brief, nucleated cells were selected by gating DRAQ5, Hoechst double-positive cells, followed by $z$-normalization of protein markers used for clustering (some phenotypic markers were not used in the unsupervised clustering). Cells positive ($z > 1$) for greater than 35 fluorescent markers were removed

from the data. Then the data were overclustered with X-shift (https://github.com/nolanlab/vortex) or Leiden-based clustering with the scanpy Python package (v.1.9.1). These processing steps were performed based on an in-depth analysis of normalization techniques and unsupervised clustering algorithms used for CODEX multiplexed imaging data of the intestine[75]. These are not new approaches and many packages have emerged for integrating these clustering algorithms into libraries such as Squidpy[75]. Clusters were assigned a cell type on the basis of average cluster protein expression and the location within image. Impure clusters were split or reclustered after mapping back to the original fluorescence images.

## Cell type annotation using STELLAR

CODEX cell type labels were transferred to other donors using the STELLAR framework for annotating spatially resolved single-cell data, as we described previously[12]. In brief, STELLAR is a geometric deep learning method that uses the spatial and molecular cell information to transfer cell type labels across different datasets. While SpaGCN[76] and Spage2vec[77] can leverage spatial and molecular data to annotate cells, these methods are unsupervised clustering methods. As such, they require manual effort to assign cell clusters to corresponding cell types, and can also require additional manual effort for multiple iterative cluster refinements. On the other hand, STELLAR automatically identifies existing cell types and discovers cell types without requiring manual effort, so we used STELLAR as the preferred method. To apply STELLAR, we used B004 donor data as the labelled training dataset as defined in STELLAR. This dataset was curated and annotated by clustering, merging, reclustering, subclustering and assigning cells to cell types based on average marker expression. Each cluster's purity and accuracy were confirmed by location of the cell within CODEX images with corresponding fluorescence images and also H&E staining. We used B004, B005 and B006 donor dataset to train STELLAR and transfer annotations to all other donor datasets that were treated as unannotated test datasets in the STELLAR framework. We did not expect to find any new cell types across different donors so we used STELLAR to identify existing cell types across donors. All datasets were z-normalized as suggested previously[75]. We then manually confirmed the quality of STELLAR's cell type assignments in all donor datasets by looking at average marker expression profiles of predicted cell types. We found that protein marker distributions match expert hand-annotated profiles[12].

## CODEX cell type percentage normalization

We normalized the cell type percentage to the overall cell category for three reasons. First, we captured a cross-section of each individual intestinal piece; however, a few of the intestinal pieces (4 out of 64) were not representative cross-sections. Second, with a fixed CODEX imaging window, we attempted to capture the full mucosal area, as this contains the majority of cell types identified by antibodies in our panel, which led to variable amounts of the muscularis externa captured. Third, it is useful to normalize by overall cell type to understand how cell type compositions change across the three compartments.

## CODEX same-cell density analysis

CODEX same-cell density was analysed by taking the average distance of the five nearest neighbours of the same cell type for each individual cell in each imaged region. This average distance was divided by the most diffuse distance for same cell types. The most diffuse same cell distance was calculated by taking the total number of cells of a given cell type divided by the total area of the region. Thus, numbers that are closer to 1 are least dense and numbers closer to 0 are more dense with cells of the same cell type.

## Neighbourhood identification analysis

Neighbourhood analysis was performed as described previously[10,27]. In brief, this analysis involved (1) taking windows of cells across the entire cell type map of a tissue with each cell as the centre of a window; (2) calculating the number of each cell type within this window; (3) clustering these vectors; and (4) assigning overall structure on the basis of the average composition of the cluster (Extended Data Fig. 2a). In brief, determining window size cut-offs for cellular neighbourhood analysis is an important metric to be chosen. In general, smaller window sizes will identify more local or microstructures, whereas larger window sizes will lead to the identification of similarly composed structures that require a larger window size. For our neighbourhood analysis here, we chose to have window size cut-offs by selecting the ten nearest neighbours around a given cell. This number has worked well to identify conserved compositions representing a cell's immediate microenvironment or local neighbours in other tissues[10,12,27,75,78,79]. We chose this strategically to look at the microstructures at the neighbourhood level because we also were curious to understand how these microstructures work together and come together to form macrostructures of the intestine at a multihierarchical scale. In general, the size of the structure does not directly relate to the window size choice, but instead relates to how compartmentalized the conserved cell types are within a given structure. Neighbourhoods were overclustered to 30 clusters. These clusters were mapped back to the tissue and evaluated for cell type enrichments to determine overall structure and merged down into 20 unique structures.

## Neighbourhood conservation analysis

To determine neighbourhood compositional (cell type) conservation across the small bowel and colon, neighbourhood enrichment scores were found separately for both the small bowel and colon samples across all donors. This enrichment score is the average cell type percentage within the average of the neighbourhood cluster divided by the average cell type percentage for all cells in the tissue. The colon enrichment scores for each cell type were subtracted from the small bowel scores to provide the heat map that was ordered both in terms of the greatest absolute sum of differences for both neighbourhood and cell type in conservation. Thus, differences in cell type enrichment for a given neighbourhood indicate that this cell type is not compositionally conserved across the small and large intestines for this neighbourhood structure.

## Community and tissue unit identification analysis

Communities were determined similarly to how multicellular neighbourhoods were determined with some small differences. In brief, the cells in the neighbourhood tissue maps were taken with a larger window size of 100 nearest neighbours. These windows were taken across the entirety of the tissue and the vectors then clustered with k-means clustering and overclustering with 20 total clusters. These clusters were mapped back to the tissue and evaluated for neighbourhood composition and enrichment to determine overall community type. This same approach was applied to communities with a window size of 300 nearest neighbours.

## Hierarchical intestine structural graphs

Each hierarchical level was connected to the next by either what it contributed to the largest in the next level, or what made up at least 15% of this next hierarchy. The percentage of each feature in the level was represented by size of the shape. The shape and colour combination correspond to the level and feature respectively. The size of the connecting line represents the amount that a feature contributes to the next feature.

## Spatial context maps

Spatial context maps were created as previously described[27]. In brief, the spatial context analysis of neighbourhood–neighbourhood or community–community associations has some similarities with our method to identify multicellular neighbourhoods but also contains a

few key differences. First, windows of neighbourhoods or communities were calculated with either 100 or 300 nearest neighbours respectively. For the neighbourhood spatial context maps, only the cells classified in the mucosa tissue unit were included in the analysis, whereas the community spatial context maps included all cells from all tissues. Once windows were calculated (number of each cell type within the window), then the combinations representing more than 85% of the neighbourhoods within that window were selected as a combination. This combination informs about prominent associations of neighbourhoods or communities in the window, a feature that we termed spatial context. Combinations were then counted and represented in size by the size of the black circle underneath the square neighbourhood combination and only combinations that had a greater frequency than 0.1% of all combinations were plotted for visualization purposes. Each combination was then connected to each combination containing it with another combination in sublayers of the graph. For example, if the Secretory Epithelial community was found to represent 85% of a window, then this would be its own combination (purple triangle). If it is found as a combination with the Adaptive-Immune-Enriched community, then it is connected with this in a combination (purple triangle and orange triangle). Similarly, if it is found as a combination with the Plasma-Cell-Enriched community, then it is connected with this in a combination (purple triangle and yellow triangle). The single combination is therefore connected (through black edges) to these combinations (nodes) in the spatial context graph. Similarly, combinations of the Secretory Epithelial and Adaptive-Immune-Enriched communities (purple triangle and orange triangle) derive from this combination (for example, Secretory Epithelial, Adaptive-Immune-Enriched and Follicle (purple, orange and blue triangles)).

## Tissue motif identification

We used a previously developed method to identify significantly associated cellular neighbourhood-neighbourhood motifs[27]. In brief, motif identification uses segmented areas of the tissue where multiple cells of the same community are co-located, instead of individual cells (Extended Data Fig. 5a). The tissue network graphs therefore represent shared edges between instances of communities. To create a tissue graph for each treatment group, we took the union of the tissue graphs of each unique imaging region. We then created a null-set as the graph of the set of cell neighbourhood or community assignments by a sequence of valid transpositions of cell neighbourhood or community assignments. Permuting neighbourhood or community assignments and fixing the number of vertices created the maximum entropy null distribution. Only two chains with at least five instances were considered. To identify significant chains, *P* values were Bonferroni corrected by multiplying by twice the number of tests conducted in each comparison group (small bowel and CL).

## Tissue dissociation and nucleus isolation for single-nucleus experiments

Nuclei were isolated using the OmniATAC protocol[80]. Isolation of nuclei was carried out on wet ice. A total of 40–60 mg of flash-frozen tissue was gently triturated and thawed in 1 ml HB (lysis) buffer (1.0341× HB stable solution, 1 M DTT, 500 mM spermidine, 150 mM spermine, 10% NP40, cOmplete Protease Inhibitor, Ribolock) for 5 min. Tissue was then dounced 10 times with pestle A and 20 times with pestle B, or until there was no resistance from either pestle. The samples were then filtered through a 40 µm cell strainer (Falcon, 352340) and the resulting homogenate was transferred to a prechilled 2 ml LoBind tube. The samples were centrifuged in a 4 °C fixed-angle centrifuge for 5 min at 350 rcf to pellet the nuclei. After centrifugation, all but 50 µl of supernatant was removed. Then, 350 µl HB was added to the nucleus pellet for a total volume of 400 µl and the nuclei were gently resuspended using a wide bore pipet. One volume of 50% iodixanol

(60% OptiPrep (Sigma Aldrich; D1556), diluent buffer (2 M KCl, 1 M MgCl$_2$, 0.75 M Tricine-KOH pH 7.8), water) was added and the resulting solution was gently triturated. Next, 600 µl of 30% iodixanol was carefully layered under the 25% mixture. Finally, 600 µl of 40% iodixanol was layered under the 30% mixture. The sample was then centrifuged for 20 min at 3,000 rcf in a 4 °C swinging-bucket centrifuge, resulting in a visible band of nuclei. The supernatant was aspirated down to within 200–300 µl of the nucleus band. The nucleus band was then collected at 200 µl and transferred to a fresh 1.5 ml tube. The sample was diluted with one volume (200 µl) of resuspension buffer (1× PBS, 1% BSA, 0.2 U µl$^{-1}$ Ribolock). The nucleus concentration was determined using the Countess II FL Automated Cell Counter (Thermo Fisher Scientific; AMQAF1000).

## snATAC–seq

snATAC-seq targeting 9,000 nuclei per sample was performed using the Chromium Next GEM Single Cell ATAC Library & Gel Bead Kit v1.1 (10x Genomics, 1000175) and the Chromium Next GEM Chip H (10x Genomics, 1000161) or Chromium Single Cell ATAC Library & Gel Bead Kit (10x Genomics, 1000110). Libraries were sequenced on the Illumina NovaSeq 6000 system (1.4 pM loading concentration, 50 × 8 × 16 × 49 bp read configuration) targeting an average of 25,000 reads per nucleus.

## Single-nucleus transcriptome sequencing using snRNA-seq

snRNA-seq targeting 9,000 nuclei per sample was performed using Chromium Next GEM Single Cell 3′ Reagent Kits v3.1 (10x Genomics, 1000121) and the Chromium Next GEM Chip G Single Cell Kit (10x Genomics, 1000120). Libraries were pooled and sequenced on the Illumina NovaSeq 6000 system (read 1 = 28 bp, i7 index = 8 bp, i5 index = 0 bp, read 2 = 91 bp read configuration) targeting an average of 20,000 reads per nucleus.

## Single-nucleus multiome experiments

snMultiome experiments targeting 9,000 nuclei per sample were performed using Chromium Chromium Next GEM Single Cell Multiome ATAC + Gene Expression (10x Genomics, 1000283). ATAC (read 1 = 50 bp, i7 index = 8 bp, i5 index = 24 bp, read 2 = 49 bp read configuration) and RNA (read 1 = 28 bp, i7 index = 10 bp, i5 index = 10 bp, read 2 = 90 bp read configuration) libraries were sequenced separately on the Illumina NovaSeq 6000 system.

## Initial processing of single-nucleus data

Initial processing of scATAC-seq data was performed using the Cell Ranger ATAC Pipeline (https://support.10xgenomics.com/single-cell-atac/software/pipelines/latest/what-is-cell-ranger-atac) by first running cellranger-atac mkfastq to demultiplex the bcl files and then running cellranger-atac count to generate scATAC fragments files. Initial processing of snRNA-seq data was performed using the Cell Ranger Pipeline (https://support.10xgenomics.com/single-cell-gene-expression/software/pipelines/latest/what-is-cell-ranger) by first running cellranger mkfastq to demultiplex the bcl files and then running cellranger count. As nuclear RNA was sequenced, data were aligned to a pre-mRNA reference. Initial processing of the mutiome data, including alignment and generation of fragments files and expression matrices, was performed using the Cell Ranger ARC Pipeline.

## Colocalization analyses

The CODEX data were used to compute and compare the CLQ between all cell-type pairs in the small bowel versus the colon. The colocalization quotient between cell type A and cell type B was calculated using the expression $CLQ_{A\to B} = \frac{C_{A\to B}/N_A}{N_B/(N-1)}$ (ref. 81), where $C_{A\to B}$ is the number of cells of cell type A among the defined nearest neighbours of cell type B. $N$ is the total number of cells and $N_A$ and $N_B$ are the numbers of cells for cell type A and cell type B. Student's *t*-tests, adjusted for multiple-

hypothesis testing, were used identity statistically significant different CLQs between the small bowel and colon.

## Ligand and receptor analyses

The FANTOM5 database[82] and 12 more literature-supported experimentally validated ligand and receptor pairs were used to obtain the final list of validated ligand receptor pairs[83]. Non-parametric Wilcoxon rank-sum tests were used to identify differentially expressed ligands and receptors in the small bowel versus the colon (adjusted $P$-value cut-off = 0.05).

## Quality control, dimensionality reduction and clustering of snATAC–seq data

The snATAC fragments files were loaded into R (v.4.1.2) using the createArrowFiles function in ArchR[52]. Quality-control metrics were computed for each cell and any cells with TSS enrichments less than 5 were removed. Cells were also filtered on the basis of the number of unique fragments sequenced using a unique fragment cut-off that was defined for each individual sample. The sample-specific cut-offs enabled us to account for differences in sequencing depth between samples and ranged from 1,000 to 10,000, with the most common cut-off being 3,000 fragments per cell. After quality control and filtering, doublet scores for all multiome cells and all non-multiome snATAC cells were computed using the ArchR function addDoubletScores with k = 10, knnMethod = "UMAP" and LSIMethod = 1. An ArchR project was then created and doublets were filtered with filterDoublets with a filter ratio of 1.2. A small number of snATAC samples did not separate into distinct clusters of expected cell types and were removed from downstream analysis. For the non-multiome scATAC cells, an IterativeLSI dimensionality reduction was generated using addIterativeLSI, with iterations = 3, sampleCellsPre = 25000, dimsToUse = 1:25 and varFeatures = 15000. Next, clusters were added with addClusters with resolution = 1.5, nOutlier = 20, seed = 1, sampleCells = 40000 and maxClusters = 40, and the resulting clusters were divided into groups on the basis of whether the cells exhibited high gene activity scores[52], a measure of accessibility within and around a gene body, for known immune, stromal or epithelial marker genes.

## Quality control, dimensionality reduction and clustering of snRNA-seq data

After running Cell Ranger, the raw_feature_bc_matrix produced by Cell Ranger was read into R with the Seurat[84] function Read10X. The data were filtered to remove nuclei with fewer than 400 unique genes per nucleus, greater than or equal to 10,000 genes per nucleus, greater than or equal to 20,000 counts per nucleus, or greater than or equal to 5% mitochondrial RNA per nuclei. To limit the contributions of ambient RNA, we also filtered out nuclei that did not have at least three times as many counts as the median number of counts in all droplets, which should reflect the median number of counts in an empty droplet, as the large majority of droplets are empty. This limits the fraction of RNA that can come from ambient RNA in droplets that are included in the dataset. DoubletFinder[85] was run for each non-multiome snRNA sample using principal components 1–20. nExp was set to $0.076 \times$ nCells$^2$/10,000, pN to 0.25 and pK was determined using paramSweep_v3, and cells that were classified as doublets were removed before downstream analysis. snRNA data for both multiome and non-multiome cells was corrected for possible ambient RNA correction using DecontX[86].

The remaining cells from all samples were merged into a single seurat object. The data was then processed using Seurat's standard pipeline[84]. First, NormalizeData was run using the method LogNormalize and scale.factor of 10,000. Variable features were identified with Seurat's findVariableFeatures using the vst method and 2,000 features. ScaleData was then run on all genes and principal components were computed with RunPCA. To account for batch effects between different donors in clustering, the RunHarmony[87] function was run with the data being grouped by donor. RunUMAP was then used to generate a UMAP dimensionality reduction from the harmony reduction and the cells were clustered by first using FindNeighbors with reduction = "harmony" and dims = 1:20 and then FindClusters with a resolution of 1. Expression of marker genes in the resulting clusters was then used to label clusters as epithelial, stromal or immune for downstream analysis.

## Annotation of single-nucleus data

The snATAC and snRNA data were annotated in the following groups: epithelial duodenum, epithelial jejunum, epithelial ileum, epithelial colon, stromal and immune. The snATAC data were further divided into separate projects for multiome and nonmultiome cells and the RNA annotations were used for the multiome cells while the snATAC-only cells were annotated separately. For the ATAC data, the cells in each of these compartments were subset into a new ArchR project. addIterativeLSI was then run for each compartment. addHarmony was then run using the LSI dimensions as input. After computation of the harmony dimensions, the cells were clustered using addClusters and a UMAP was computed on the basis of the harmony coordinates using addUMAP. Clusters were annotated by examining gene activity scores of known marker genes. Marker genes were used for initial annotation of cell types including BEST4$^+$ enterocytes (BEST4, OTOP2), goblet cells (MUC2, TFF1, SYTL2), immature goblet cells (KLK1, RETNLB, CLCA1), stem cells (RGMB, SMOC2, LGR5, ASCL2), tuft cells (SH2D6, TRPM5, BMX, LRMP, HCK), enteroendocrine cells (SCGN, FEV, CHGA, PYY, GCG), cycling transit-amplifying cells (TICRR, CDC25C) and Paneth cells (LYZ, DEFA5).

Within the immune compartment, we identified clusters of CD4$^+$ and CD8$^+$ T cells (CD2, CD3E, IL7R, CD4, CD8), B cells (PAX5, MS4A1, CD19), plasma cells (IGLL5, AMPD1), natural killer cells (SH2D1B), macrophages/monocytes (CD14) and mast cells (HDC, GATA2, TPSAB1) (Fig. 4c and Extended Data Fig. 6d). Natural killer cells and T cells from one donor clustered separately from the other T cells in the snRNA data (Extended Data Fig. 6e), possibly because this donor was much younger (24 years) than the others in this study (Supplementary Table 1). Smooth muscle/myofibroblast clusters exhibited high expression of ACTA2, MTH11 and TAGLN. Villus fibroblasts exhibited high expression of WNT5B and some crypt fibroblasts exhibited high expression of RSPO3.

For the snRNA cells, cells were divided into six Seurat objects from the compartments listed above. Cells from the immune and stromal compartments were run through a pipeline analogous to the pipeline listed above for initial clustering (NormalizeData, ScaleData, RunHarmony, FindNeighbors and FindClusters). For cells in the four epithelial groups, we integrated the data using a different approach, first running SCTransform on the epithelial cells from each sample with assay = "decontXcounts", method = "glmGamPoi", and vars.to.regress = c("percent.mt", "percent.ribo"). We then ran the Seurat functions SelectIntegrationFeatures with features = 3000, PrepSCTIntegration, FindIntegrationAnchors using reference-based integration and dims 1–30, and finally IntegrateData. We then ran RunPCA on the integrated data followed by FindNeighbors and FindClusters to cluster the resulting integrated data.

In addition to removing probable doublets based on simulating doublets as described above, we also identified clusters with expression of markers from multiple lineages (for example, stromal and immune) during downstream clustering and annotation. For example, some cells that initially clustered with immune cells expressed higher levels of stromal genes than would be expected. For these cases, we took the following approach: we first clustered all of the cells initially classified as immune cells and identified marker genes for each cluster. We next compared the marker genes to a previously published list of colon marker genes[28] to nominate clusters that may not contain singlet immune cells. Next, we moved these cells to the stromal or epithelial compartments, in which we clustered them with all of the cells that were initially classified as epithelial or stromal cells. In this case, if the cells had high expression of immune marker genes when compared to stromal cells, we reasoned that they were most likely immune/stromal doublets and removed the cluster of cells before downstream analysis. After initial

annotation of epithelial cells, enteroendocrine cells in the snRNA data from all samples were integrated according to the SCTransform-based integration approach by running SCTranform on all epithelial cells from all samples and integrating all epithelial cells using reference-based integration as described above. After integration of all cells, we subset only the enteroendocrine cells and then computed the principal components using RunPCA and identifyied clusters using FindNeighbors and FindClusters. Known subtypes of enteroendocrine cells were then annotated based on expression of marker genes (Fig. 4 and Extended Data Fig. 6h). To annotate the MUC6+, MUC5B+ and exocrine cells, we subset the integrated duodenum data and computed the principal components using RunPCA and identified clusters using FindNeighbors and FindClusters. Exocrine cells were annotated on the basis of expression of CELA3B and CPB1, and MUC5B+ and MUC6+ cells were annotated on the basis of expression of MUC5B and MUC6 (Fig. 4h). Comparisons in cell-type abundance between regions of the intestine were done using the packages scCoda[29] and Milo[30].

## Peak calling for single-nucleus data
Peak calling was performed with MACS2 using ArchR[52]. Peak sets were defined independently for epithelial cells, stromal cells and immune cells. For epithelial cells, we wanted to generate a union peak set that captured both cell-type-specific and location-specific peaks in the epithelial compartment. To accomplish this, we divided the cells into groups, generated pseudobulk replicates for each group, called peaks on the pseudobulk replicates, generated a reproducible peak set for each group using the peaks called for the pseduobulk replicates, and then iteratively merged the peak sets for each group into a union peak set using the approach implemented in ArchR. Groups used for epithelial peak calling were the cells from each epithelial cell type—with all enteroendocrine subtypes combined, and MUC5B+, MUC6+, exocrine and unknown cells combined—in the four main regions of the intestine. Tuft cells from the small intestine were merged into a single group owing to the low number of tuft cells in the dataset. For defining the immune and stromal peak sets, cells were divided by cell type, but not location, as there were fewer cells in these compartments. Finally, an additional peak set for all cell types in the immune, stromal and epithelial cell types was defined for determining marker peaks for linkage-disequilibrium score regression.

## Integration of snRNA and snATAC data
To assign snRNA profiles to the non-multiome snATAC samples, the snRNA and snATAC datasets from the four primary regions of the intestine (duodenum, jejunum, ileum and colon) were integrated separately using the ArchR function addGeneIntegrationMatrix with reducedDims = "Harmony" and useMatrix = "GeneScoreMatrix".

## Nomination of regulatory TFs in single-nucleus data
We next identified TF regulators according to the ArchR manual for identifying TF regulators for each region, with a correlation cut-off of 0.5. TFs that met the criteria for regulators in any of the four primary regions of the intestine are plotted in Fig. 5b. This process was performed separately for the multiome datasets and the integrated singleome datasets. Cell types with few cells in each region of the intestine (for example, L cells) were combined into a single group regardless of location of origin, leading to the final cell type groupings on the x-axis of Fig. 5b. Regulators were identified separately for cells in the immune and stromal compartment with the final cell type groupings indicated on the x-axis of Extended Data Fig. 10g,h. TF footprints were computed using the ArchR functions getFootprints and plotFootprints.

## Analysis of absorptive differentiation trajectories in single-nucleus data
Absorptive differentiation trajectories for each main section of the colon (duodenum, jejunum, ileum and colon) were inferred by running

the ArchR function addTrajectory with the trajectory set as harmony clusters moving from Stem to TA2 to TA1 to immature enterocytes to enterocytes and reducedDims set to the harmony dimensions. To identify variable peaks along the trajectory, a matrix of accessibility in all peaks along the trajectory was first generated with getTrajectory with useMatrix = "PeakMatrix" and log2Norm = TRUE. Peaks with variance > 0.9 in any of the four regions were then identified with the function plotTrajectoryHeatmap with varCutOff = 0.9, returnMatrix = TRUE, scaleRows = FALSE and maxFeatures = 100000. The four matrices returned by getTrajectory were then concatenated into a single matrix and the matrix was subset to include only peaks that met the variance criteria of 0.9 in at least one of the four regions and had an absolute difference in magnitude of at least 0.2. Row z-scores for the resulting matrix were computed using the ArchR function .rowZscore. The resulting row z-scores were k-means clustered using the function kmeans with the number of clusters set to 7 and iter.max = 500. Two clusters of peaks did not show a characteristic pattern and were not included in Fig. 6. Hypergeometric enrichment of motifs in marker peaks was computed with peakAnnoEnrichment and the resulting P values are plotted in Extended Data Fig. 10k. Variable genes along the trajectory were identified with an analogous method, using GeneExpressionMatrix for the multiome data or GeneIntegrationMatrix (for the separate snATAC and snRNA data) instead of PeakMatrix when running getTrajectory and plotTrajectoryHeatmap. For gene expression, log2Norm was set to TRUE when running getTrajectory and genes were filtered to include only those with an absolute difference in magnitude of at least 0.5. Row z-scores of the resulting matrices were again k-means clustered using the function kmeans with the number of clusters set to 7 and iter.max = 500. Enrichment of KEGG pathways in these clusters of genes was determined using the limma function kegga[88], and the resulting unadjusted P values are plotted in Extended Data Fig. 10l. Plots of gene expression versus pseudotime were generated using the ArchR function plotTrajectory with the default parameters, including using imputeWeights added by addImputeWeights. TFs with correlated motif activity and RNA expression were identified with correlateTrajectories as outlined in the ArchR manual. The row z-scores of the smoothed expression of TFs along the pseudotime trajectories is plotted in Fig. 6d. TFs that were correlated with expression in any of the four trajectories were included in the heat map in Fig. 6d. Peaks correlated with gene expression were identified with addPeak2GeneLinks. In Fig. 6e, the set of peaks that were correlated with ETV6 expression with a correlation of at least 0.4 in one of the four main intestinal regions was determined. For TMPRSS15, a correlation cut-off of 0.55 was used to show the most correlated peaks in the figure. The smoothed trajectory peak accessibility for each of these peaks was then plotted along the differentiation trajectory. This process was performed separately for the multiome datasets and the integrated singleome datasets.

## Cell-type-specific linkage-disequilibrium score regression
Linkage-disequilibrium score regression is a method that aims to distinguish heritability from confounding factors such as population stratification and cryptic relatedness. To run cell-type-specific linkage-disequilibrium score regression, we first computed marker peaks for coarse cell types in our dataset. To do this, we added cell type annotations to the full ArchR project with all cells and then defined a peak set for this object by running addGroupCoverages with groupBy = "CellType" followed by addReproduciblePeakSet and addPeakMatrix. We next defined less granular cell types by merging all myofibroblast clusters and pericytes into a single group, all non-villus fibroblast clusters into a single group, all non-stem absorptive epithelial cells into a single group, cycling TA cells into a single group, all enteroendocrine cells except for EnteroendocrineUn into a single group, and lymphatic endothelial and endothelial cells into a single group. We determined marker peaks for the resulting groups of cells with getMarkerFeatures and ten selected peaks with getMarkers with

cutOff = "FDR <= 0.1 & Log2FC >= 0.5". Marker peaks from cell types with very few cells were not included (T cells, unknown, exocrine, secretory specialized MUC5B⁺, DC, ILC, adipocytes, neurons, EnteroendocrineUn) The resulting peaks were then lifted over to hg19 from hg38. We then followed the linkage-disequilibrium score regression tutorial (https://github.com/bulik/ldsc/wiki) for cell-type-specific analysis[89–91]. We used summary statistics from a number of UKBB traits (https://nealelab.github.io/UKBB_ldsc/downloads.html) related to the intestine, including non-cancer illness code;self-reported: crohns disease, non-cancer illness code;self-reported: ulcerative colitis, non-cancer illness code;self-reported: malabsorption/coeliac disease, BMI as well as traits less clearly related to the intestine including non-cancer illness code;self-reported: hypertension and diagnoses−main ICD10: K02 dental caries. Coefficient *P* values from ldsc are plotted in the heat map in Fig. 6. Significance was determined by correcting the coefficient *P* values for the number of cell types tested with Bonferroni correction with the R function p.adjust.

### Processing of single-nucleus data for ligand−receptor analysis and snRNA-CODEX integration

The ligand−receptor analysis and scRNA-seq CODEX integration were performed with an initial dataset consisting of samples from donors B001, B004, B005 and B006 before collection of the remaining data. This dataset was annotated similarly to the full dataset with the following differences. First, all analysis was carried out in R v.4.0.2. Second, the quality-control cut-offs used were slightly different, with the requirement to have three times as many RNA counts as an empty droplet not implemented in the initial dataset. Third, when running doubletFinder, pK was set as 0.09 instead of running paramSweep_v3. Fourth, scTransform was not used in the subclustering analysis for the epithelial compartments. Instead, Seurat's standard normalize and scale pipeline followed by Harmony was run to compute an integrated dimensionality reduction that was then clustered using Seurat's findNeighbors and FindClusters for the immune and stromal cells as well as the epithelial cells from the four main regions of the intestine. As the ligand−receptor analysis involved making predictions that we later attempted to validate using Molecular Cartography, this analysis could not be redone with the remaining dataset.

### Molecular Cartography validation of ligand−receptor pairs

Small intestine (duodenum) and colon (sigmoid/descending) samples from donors B004, B008 and B009 were analysed for spatial transcriptomics. The cryosections (thickness, 10 µm) from the same OCT-embedded tissue arrays were placed onto the glass slides provided by Resolve Biosciences for Molecular Cartography assay. The Molecular Cartography assay was performed by the Resolve Biosciences team with their optimized protocol 'human colon v1.3' and targeting a panel of total 100 transcripts of interest, including 63 genes from our ligand−receptor predictions (Supplementary Table 3) and 37 genes for cell type annotation (Supplementary Table 9). Segmentation was performed using DAPI signal with cellPose (https://github.com/MouseLand/cellpose; v.2.0.5) and followed by Baysor (https://github.com/kharchenkolab/Baysor; v.0.5.1). Gene counts were quantified per cell and cells were removed by area (<50 or >8,000 pixels) and total gene counts (<2). Manual cell type annotation was performed on the basis of the marker gene expression after Leiden clustering (Scanpy). As only 37 genes were used as cell type markers and the sample number ($n$ = 6) is limited, only 20 cell types were identified in this dataset. To validate the ligand−receptor repair (Supplementary Table 3), we first match the cell type with snRNA-seq annotation. Only 58 out of 152 predictions have matching cell types. We compared ligand and receptor expression (log transformed) between the colon and small intestine in their predicated cell types (one sided Wilcoxon rank-sum test). *P* values were corrected for multiple testing using the Benjamin−Hochberg procedure. In total, 15 pairs have consistent higher expression in the colon compared with

in the small intestine (both adjusted *P* < 0.05; Supplementary Table 6). Permutation tests were used to assess whether the success rate (15.5%, 9 out of 58) of our validation is higher than random. Gene labels were swapped and the same DEG procedure was repeated 10,000 times.

### snRNA-seq/CODEX single-cell matching and integrational analysis using MaxFuse

snRNA-seq cells and CODEX cells were matched and downstream integrative analysis was performed using MaxFuse, of which the methodology details were described previously[41]. In brief, MaxFuse is an algorithm that matches cells across different single-cell modalities by linear assignment, using both shared (when available) and unshared features, and implements signal boosting steps (for example, graph smoothing and meta cell construction) to enable matching cells across weakly correlated modalities (for example, RNA to protein). Although various methods are available for integration tasks on modalities with robust sharing information (such as scRNA/scATAC)[84,92,93], when such tasks involve integration between protein and sequencing modalities, with much weaker shared features available (<60 versus thousands), a specialized method is needed[94]. We applied MaxFuse to match snRNA-seq cells to CODEX cells. Cells that were previously annotated as B, T, monocyte, macrophage, plasma, goblet, endothelial, enteroendocrine, smooth muscle and stromal cells were used during this integration process, whereas other cell types were not used owing to limited sharing information across modalities. Subsequently, a shared co-space was calculated to embed both modalities, with visualization of the embedding (first 20 MaxFuse-components) using UMAP. To evaluate single-cell matching performance across RNA to protein modality, we used cell type annotation accuracy (for example, a single CODEX plasma cell matched to a snRNA-seq plasma cell) as a proxy, and both CL and small bowel matching achieved >90% accuracy. Using the single-cell-level pairing information, we transferred the transcriptome expression profile to each individual CODEX cell, and subsequently performed analysis of the DEGs across various CODEX cellular neighbourhoods. The DEGs were selected with the function FindAllMarkers in the R package Seurat, with the parameters only.pos = TRUE, min.pct = 0.3, logfc.threshold = 0.25. The genes with adjusted *P* < 0.05 and shared across the CL and small bowel datasets were shown on the heat map. Gene Ontology enrichment analysis was performed for individual CNs with the DEGs, by using the PANTHER database. We have uploaded the code that we used to perform the MaxFuse matching for the snRNA-seq and CODEX datasets within the paper here (https://github.com/shuxiaoc/maxfuse/tree/main/Archive/hubmap_nature). This includes both the dataset preparation and analysis features of the code.

### Reporting summary

Further information on research design is available in the Nature Portfolio Reporting Summary linked to this article.

## Data availability

All of the datasets (snRNA-seq, snATAC−seq and CODEX) can be visualized and accessed through a website portal (https://portal.hubmap-consortium.org/). Our landing page links all of the raw dataset IDs and the HuBMAP ID for this Collection is HBM692.JRZB.356 and the DOI is https://doi.org/10.35079/HBM692.JRZB.356. Supplementary Table 10 also lists all of the dataset IDs within the HuBMAP portal where all raw datasets are stored that can be downloaded and also viewed in a processed state. We provide the processed fluorescence CODEX multiplexed image stacks (https://doi.org/10.5061/dryad.76hdr7t1p and https://doi.org/10.5061/dryad.gmsbcc2sq). We also provide processed, quantified and annotated single-cell CODEX datasets with labelled cell types, neighbourhoods, communities, tissue units and also protein expression at Dryad (https://doi.org/10.5061/dryad.pk0p2ngrf). Processed snRNA-seq and snATAC-seq datasets are available at https://

doi.org/10.5061/dryad.8pk0p2ns8 and https://doi.org/10.5061/dryad.0zpc8672f. Unprocessed snRNA-seq and snATAC–seq datasets are available at dbGaP (phs002272.v1.p1). Source data are provided with this paper.

## Code availability

Code for analysis of the snATAC and snRNA data is available at GitHub (https://github.com/winstonbecker/scColonHuBMAP/), and code for processing CODEX multiplexed imaging data (https://github.com/nolanlab/CODEX)[8], for clustering (https://github.com/nolanlab/vortex), for transferring cell type labels with STELLAR (https://github.com/snap-stanford/stellar), for neighbourhood analysis (https://github.com/nolanlab/NeighborhoodCoordination), for tissue schematics (https://github.com/nolanlab/TissueSchematics) and for MaxFuse (https://github.com/shuxiaoc/maxfuse) is available at GitHub.

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

**Acknowledgements** We thank the patients for donating their tissues. This work was supported by the US National Institutes of Health (3U54HG010426; P50HG007735; U19AI057266; U01HG011762); Cancer Research UK (C27165/A29073); the Parker Institute for Cancer Immunotherapy (PICI0025); Hope Realized Medical Foundation (209477). J.W.H. was supported by an NIH T32 Fellowship (T32CA196585) and an American Cancer Society—Roaring Fork Valley Postdoctoral Fellowship (PF-20-032-01-CSM). S.A.N. was supported in part by the Stanford Graduate Fellowship and the NHGRI Stanford Genome Training Program 5 T32 000044. C.M.S. was supported by the Swiss National Science Foundation (P300PB_171189, P400PM_183915). Figs. 2g,i and 4a and Extended Data Figs. 1a, 2a and 6a were created using BioRender. We thank the members of the HuBMAP consortium for their many contributions to this project.

**Author contributions** S.B. and C.M.S. developed the original CODEX panel and acquired the CODEX data from B001. J.W.H., S.B. and C.C. developed gut-specific CODEX antibodies and ran the remaining donor samples. V.V. helped to establish CODEX data-processing pipelines and integration with HuBMAP data transfers. M.B., K.C., J.W.H. and J.L. created STELLAR, which was used for cell type label transfer of the CODEX data. J.W.H. processed and analysed CODEX multiplexed imaging data. R.C., D.C., S.A.N., A.H. and W.R.B. established experimental protocols and collected the snRNA-seq and snATAC–seq data. A.E.P. and W.Z. analysed CODEX and snRNA-seq data for neighbourhood crosstalk. C.Z. and J.W.H. analysed the spatial transcriptomics dataset based on Molecular Cartography. B.Z., S.C., N.Z. and Z.M. integrated CODEX and snRNA datasets using MaxFuse. W.R.B. analysed snATAC–seq and snRNA-seq data. W.R.B., J.W.H., W.J.G. and M.S. wrote the manuscript and created figures. D.C. handled data management and submission to repositories. A.H., A.K.W. and E.M. were involved in project coordination. T.L. assessed morphological staining of samples. E.D.E. assisted in metadata annotation and biological interpretation. Y.L., Z.Z. and S.L. procured samples. Y.L., S.K.P., G.P.N., W.J.G. and M.S. supervised the study. All of the authors revised the manuscript and accepted its final version.

**Competing interests** C.M.S. is a scientific advisor to, has stock options in and has received research funding from Enable Medicine. 10X Genomics holds the license to patents in which W.J.G. is listed as an inventor. W.J.G. is an equity holder of 10X Genomics, and a co-founder of Protillion Biosciences. W.J.G. consults for Guardant Health, Quantapore, Protillion Biosciences, Ultima Genomics, Lamar Health, and Erdio Biosciences. M.P.S. is a cofounder and an advisory board member of Personalis, Qbio, January AI, Mirvie, Filtricine, Fodsel, Protos, RTHM, Marble Therapeutics and Crosshair Therapeutics. G.P.N. received research grants from Pfizer, Vaxart, Celgene and Juno Therapeutics; and has equity in and is a scientific advisory board member of Akoya Biosciences. The other authors declare no competing interests.

**Additional information**
**Correspondence and requests for materials** should be addressed to Garry P. Nolan, William J. Greenleaf or Michael Snyder.

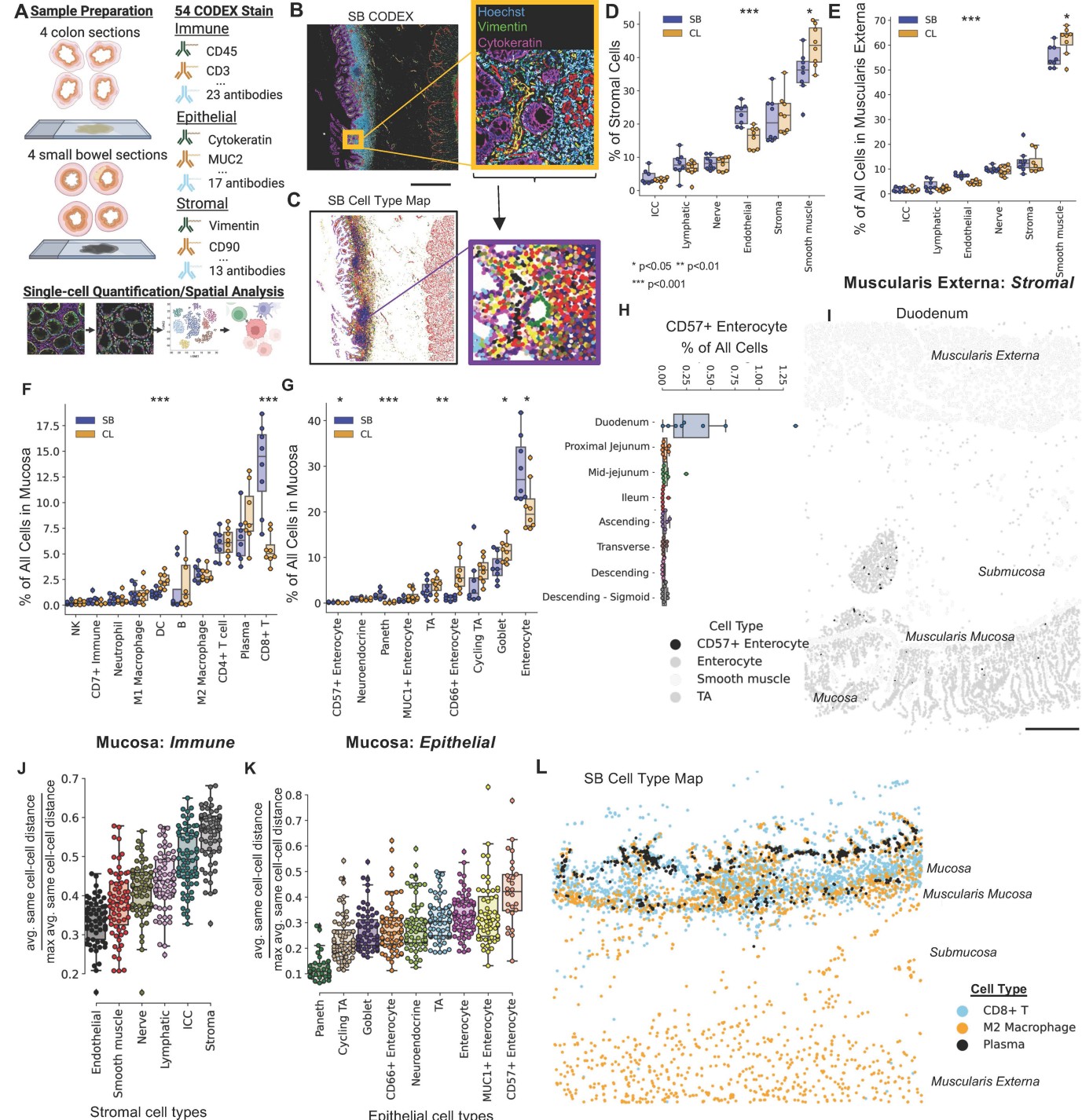

**Extended Data Fig. 1 | CODEX mulitplexed imaging across the healthy human intestine reveals changes in cell composition and organization.**
A) Schematic for how CODEX multiplexed imaging was performed on arrays of 4 different sections of either colon and small intestine from the same donor simultaneously. Image processing steps done to extract single-cell spatial data. B) An example CODEX fluorescent image of one region of the small bowel (SB) (1 of 64 tissue sections) for CODEX with 6/54 markers shown for one donor (scale bar = 1 mm and magnified insert = 100 μm) with C) accompanying cell type map following cell segmentation and unsupervised clustering. D-E) Stromal cell type percentages either as a percent of D) All stromal cells, or E) all cells restricted to the *Muscularis Externa* tissue unit. F) Immune cell type or G) epithelial cell type percentages either as a percent of all cells restricted to the *Mucosa* tissue unit. H) Percentage of CD57+ Enterocyte cells of all cell types across different areas samples from small intestine to colon. (for D-H:

* p value< 0.05, ** p value< 0.01, *** p value < 0.001 by two-sided T test, n=8 donors). (All boxplots in figures are plotted as minimum, 25 percentile, median, 75 percentile, maximum, and outliers as points outside 1.5 the interquartile range). I) Cell map of a representative section (one of 8 donors) of the Duodenum that shows CD57+ Enterocyte presence in glands in the *Submucosa* where Enterocytes and TA cells are shown in dark grey and Smooth muscle cells in light grey (other cell types not shown) (scale bar = 500 μm). J-K) Quantification of the same-cell density that is measured as an average distance of its 5 nearest same-cell neighbours normalized by the maximal possible same-cell distance within the tissue (n = 64 tissue sections) for J) stromal and K) epithelial cell types. L) A representative cell type map (one of 64 tissue sections from 8 donors) with only plasma cells, CD8+ T cells, and M2 Macrophages shown (scale bar = 500 μm).

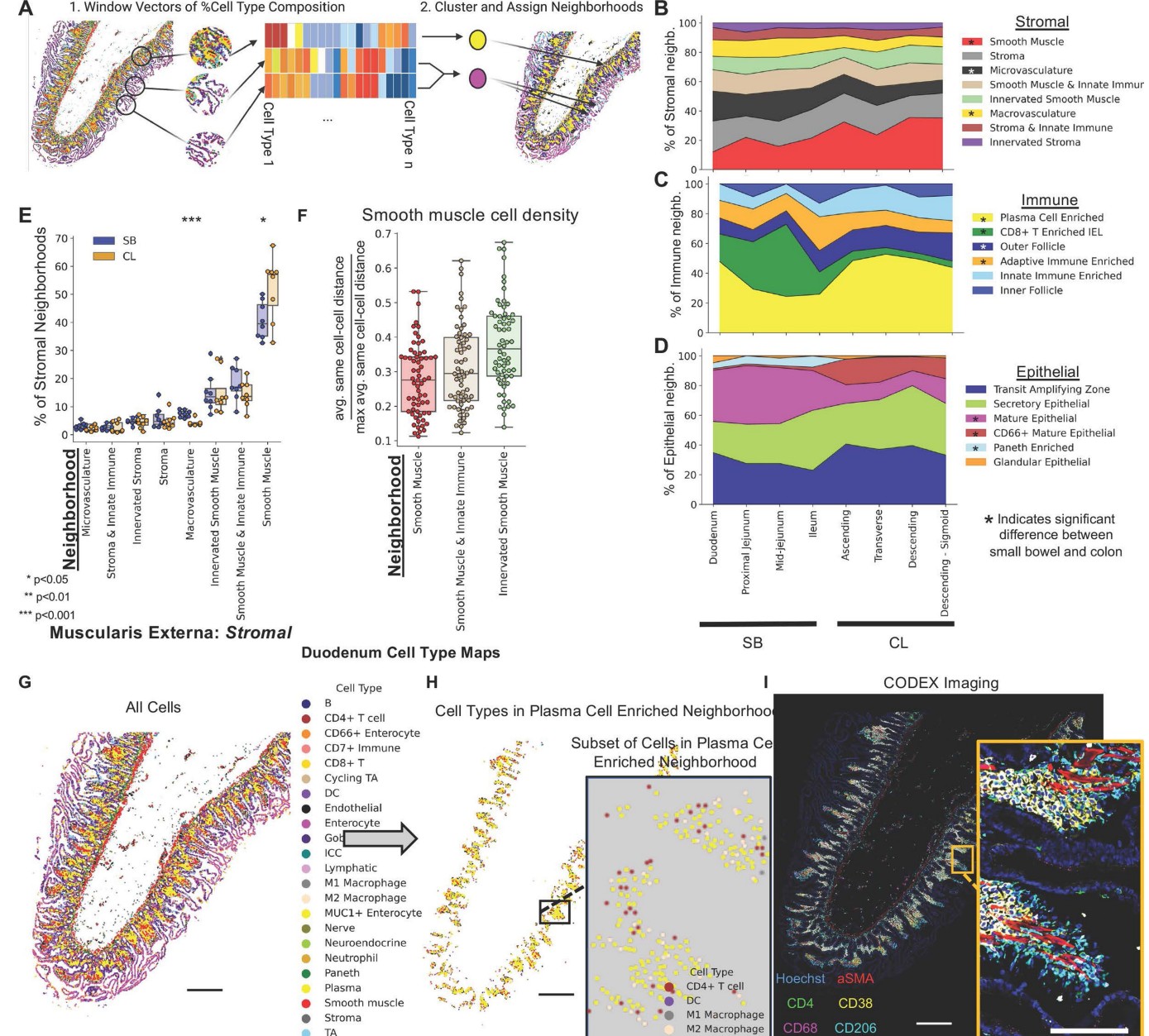

**Extended Data Fig. 2 | Multicellular neighbourhood analysis of CODEX imaging data indicates conserved cellular structures across the intestine.**
A) Neighbourhood analysis was done by taking a window across cell type maps and vectorizing the number of cell types in each window, clustering windows, and assigning clusters as cellular neighbourhoods of the intestine. B-D) Neighbourhood percentages from CODEX data averaged normalized by B) stromal, C) immune, and D) epithelial compartments. Asterix indicates p-value less than 0.05 difference in cell type percentage from the small bowel (SB) to the colon (CL) by two-sided T test. E) Stromal multicellular neighbourhood percentages either as a percent of all neighbourhoods restricted to the *Muscularis Externa* tissue unit (* p value< 0.05, *** p value < 0.001, n = 8 donors,

by two-sided T test,). F) Quantification of the same-cell density for just smooth muscle cells within different smooth muscle multicellular neighbourhoods (x axis) (n = 32 tissue sections). G-H) Cell type maps for a region of the small intestine (one of 64 tissue sections imaged from 8 donors) with G) all cell types plotted for the whole tissue (scale bar = 500 µm), H) cells contained within the plasma cell neighbourhood (scale bar = 500 µm), and a magnified area of denoted by rectangle showing subset of cell types (scale bar = 50 µm).
I) CODEX fluorescent imaging with subset of fluorescent markers overlaid for the same tissue as G (Hoechst=Blue, CD4=Green, CD68=magenta, CD38=yellow, CD206=cyan, CD138=grey), (scale bar = 500 µm with magnified insert scale bar = 100 µm).

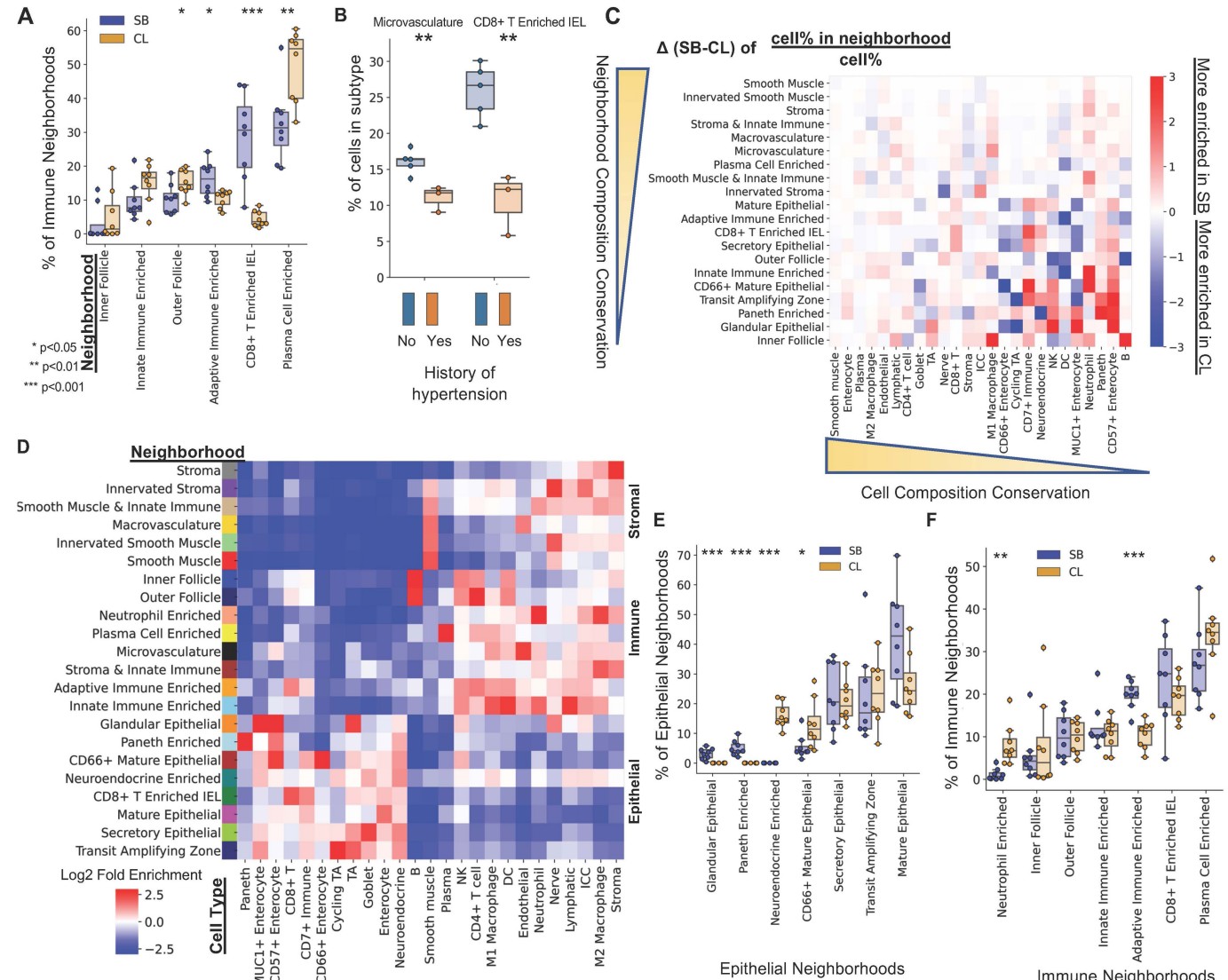

**Extended Data Fig. 3 | Change in multicellular neighbourhoods across the intestine.** A) Immune neighbourhood percentages as a percent of immune neighbourhoods. B) Neighbourhood percentages for *Microvasculature* and *CD8+ T cell IEL* neighbourhoods compared for donors with or without a history of hypertension (Microvasculature p value = 0.0065<, CD8+ T Enriched IEL p value = 0.0017 by two-sided T test, n = 3-5 donors). C) Difference in composition in neighbourhood by cell type for all neighbourhoods based on subtracting the log2 fold enrichment of each cell type found within that neighbourhood compared to average percentages in the tissue in SB from CL. Neighbourhoods and cell types are ordered by summing the absolute value of all rows and columns to denote conservation of a neighbourhood. D) 22 unique intestinal multicellular neighbourhoods (y axis of heatmap) were defined by enriched cell types (x axis of heatmap) as compared to overall percentage of cell types in the samples with 2 unique neighbourhoods not identified with overall neighbourhood analysis. E) Epithelial neighbourhood percentages as a percent of epithelial neighbourhoods from multicellular analysis performed on each individual region of the intestine separately. F) Immune neighbourhood percentages as a percent of immune neighbourhoods from multicellular analysis performed on each individual region of the intestine separately. (* p value< 0.05, ** p value< 0.01, *** p value < 0.001 by two-sided T test, n = 8 donors).

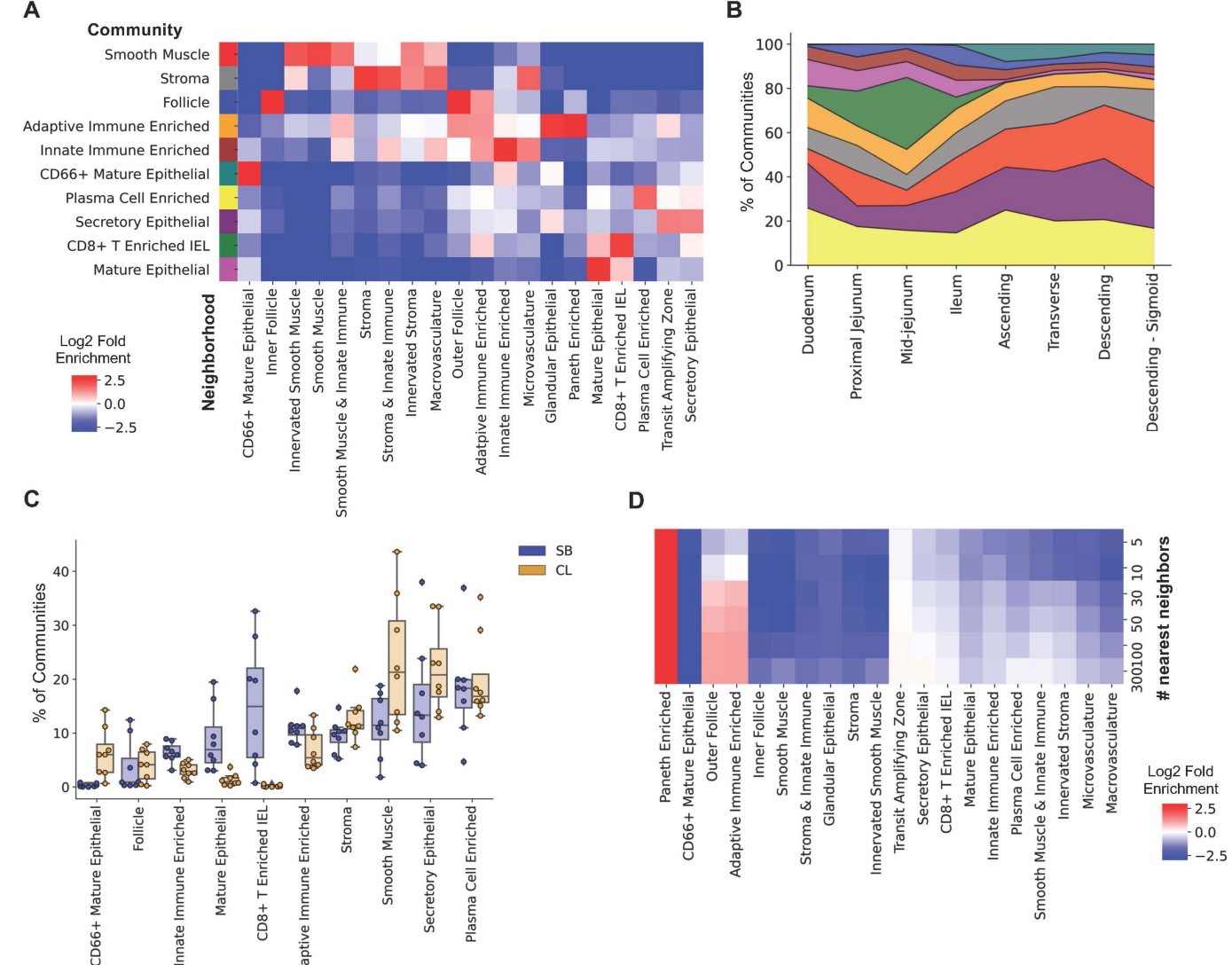

**Extended Data Fig. 4 | Multi-neighbourhood community analysis of the intestine.** A) Community analysis was done by taking a window across neighbourhood maps and vectorizing the number of each neighbourhood type in each window, clustering windows, and assigning clusters as multi-neighbourhood communities of the intestine. 10 unique intestinal multi-neighbourhood communities (y axis of heatmap) were defined by enriched neighbourhood types (x axis of heatmap) as compared to overall percentage of neighbourhood types in the samples. B) Quantification of neighbourhood types across each section of the intestine (colour legend within panel A). C) Community percentages as a percent of all communities for small intestine and colon (* p value< 0.05, ** p value< 0.01, *** p value < 0.001, n = 8 donors). D) Concentric multi-neighbourhood analysis surrounding only neighbourhoods labelled as *Paneth Cell Enriched* neighbourhoods, with number of nearest neighbours for a given row in the heatmap.

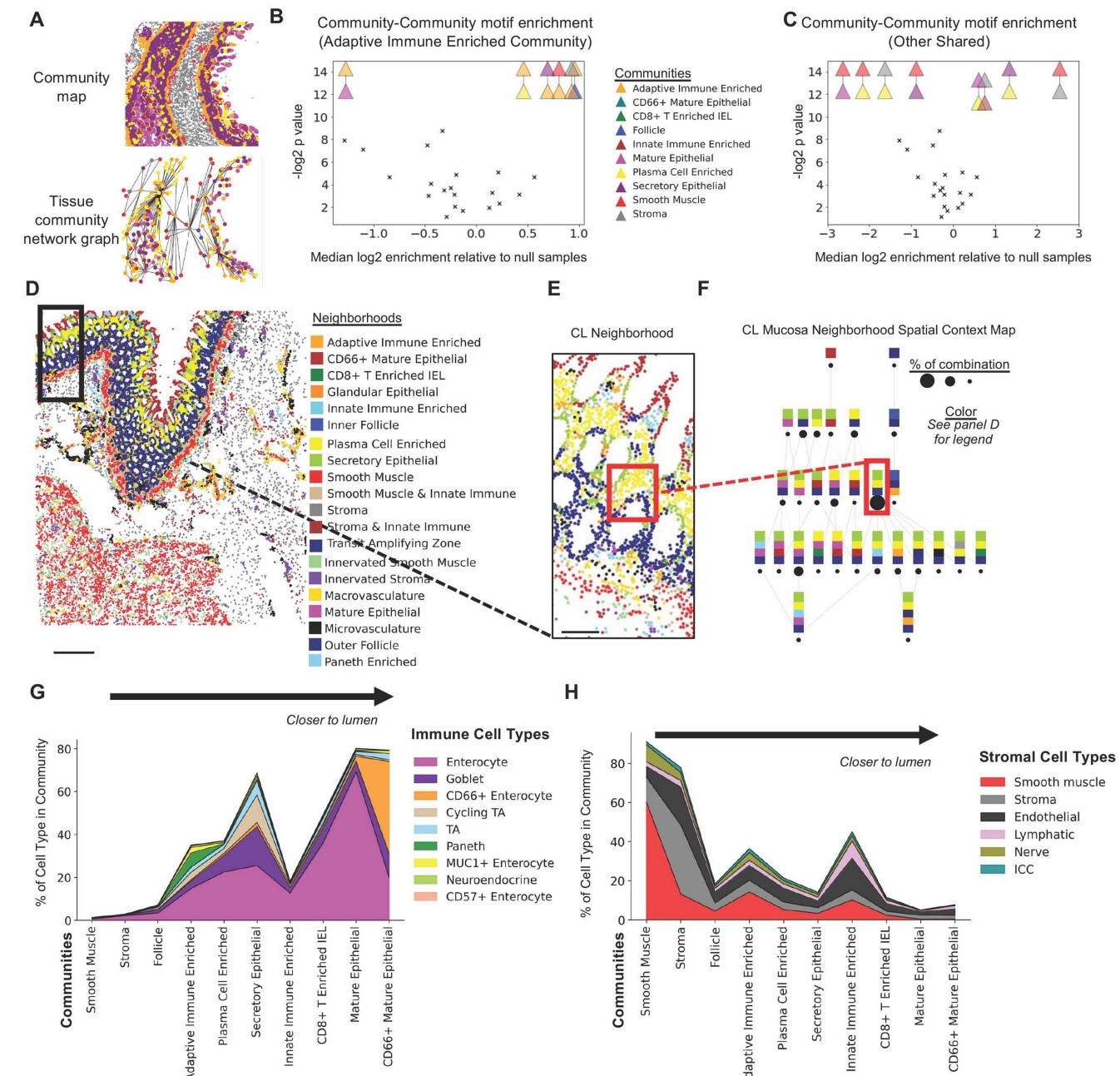

**Extended Data Fig. 5 | Multicellular neighbourhood and community interactions across the intestine.** A) Schematic of a representative community map and corresponding tissue community network graph (one of 64 tissues). B-C) Community-community motifs that are significantly enriched in both the small intestine and colon as compared to a null distribution of motif instances created from random permutation of tissue graph labels, where B) shows only those motifs that interact with the *Adaptive Immune Enriched* community and C) shows all other shared motifs between the SB and CL. Motifs indicated by shape and colour indicate those motifs that have significant p value versus those that are indicated with an x in the graph; p values were Bonferroni corrected by multiplying by twice the number of tests conducted in each comparison group. Colour legend is also the same as panel B. D) Representative neighbourhood map (one of 64 tissue sections from 8 donors) with (scale bar = 500 μm)

E) Region magnified as in the main figure of the mucosal area of a colon community map, but this time with the multicellular neighbourhoods coloured (see panel D for legend) (scale bar = 100 μm). F) Spatial context maps of the CL highlighting relationships of multicellular neighbourhoods across just the neighbourhoods found within the tissue unit *Mucosa*. This structure is defined by the number of unique neighbourhoods required to make up at least 85% in a given window. Circles represent the number of cells represented by a given structure. Red rectangle highlights a structure discussed in the manuscript and maps this structure back to panel K. Colour legend is also the same as panel D. G-H) Cell type percentage of G) epithelial and H) stromal cells shown for each community ordered in relative order of general increasing proximity to the lumen based on community spatial context analysis.

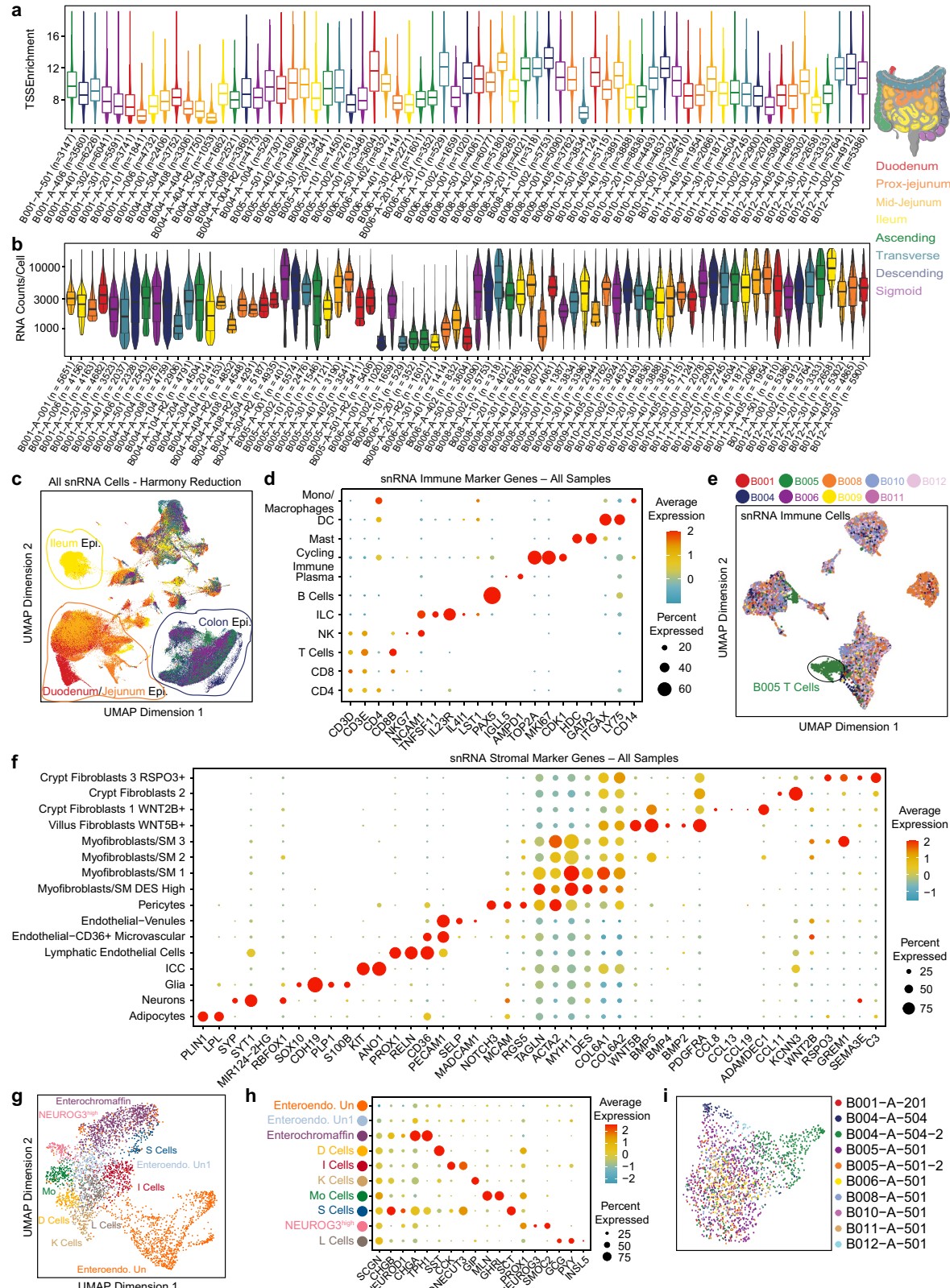

**Extended Data Fig. 6 | Quality control and clustering of single-nucleus data.**
(a,b) Violin plots of TSS enrichment (a) and RNA counts/cell (b) for different samples included in the study. Samples are coloured by the location from which they were obtained. (c) UMAP projection of all snRNA cells coloured by location. (d) Dotplot representation of expression of immune marker genes by immune cell types. (e) UMAP projection of scRNA immune cells coloured by donor.

(f) Dotplot representation of expression of stromal marker genes by stromal cell types. g) Sub-clustering of enteroendocrine cells from all regions of the intestine. h) Dotplot representation of the expression of subtype specific enteroendocrine and enterochromaffin marker genes in different enteroendocrine cell types in our datasets. I) Sub-clustering of specialized secretory cells coloured by sample.

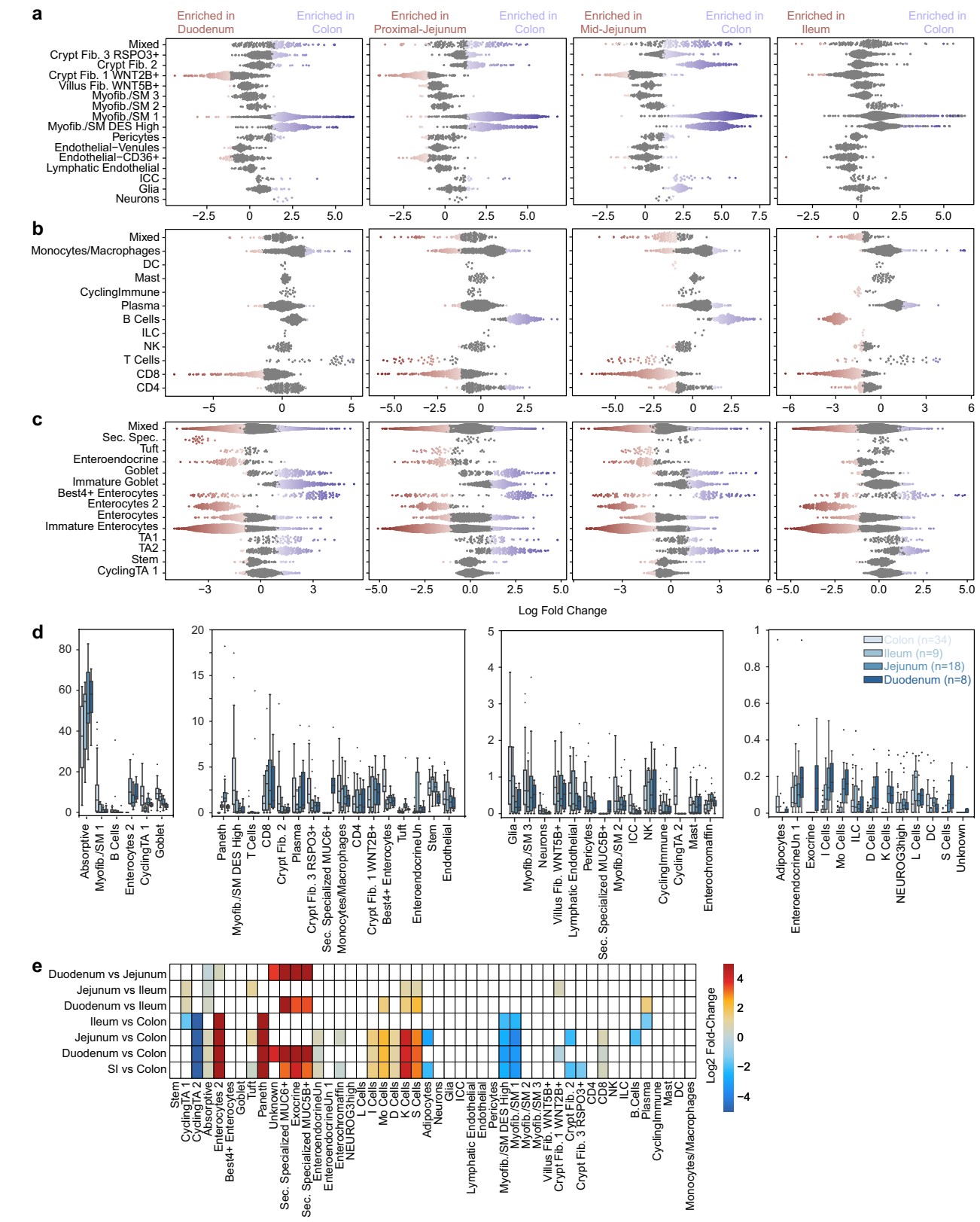

**Extended Data Fig. 7 | Differential cell type abundance.** a–c) Beeswarm plot showing the log-fold change between the three main regions of the small intestine and colon for groups of nearest neighbour cells from different cell type clusters in the stromal (a), immune (b), and epithelial (c) compartments computed with Milo. Significant changes are indicated in colour. d) Boxplots comparing the fraction of all cells in each sample composed of each cell type for samples from the colon, ileum, jejunum, and duodenum. e) Log2FC in abundance of each cell type between the regions listed on the y-axis as estimated with scCODA. Only significant results at an FDR of 0.05 are shown, with all nonsignificant differences plotted as white.

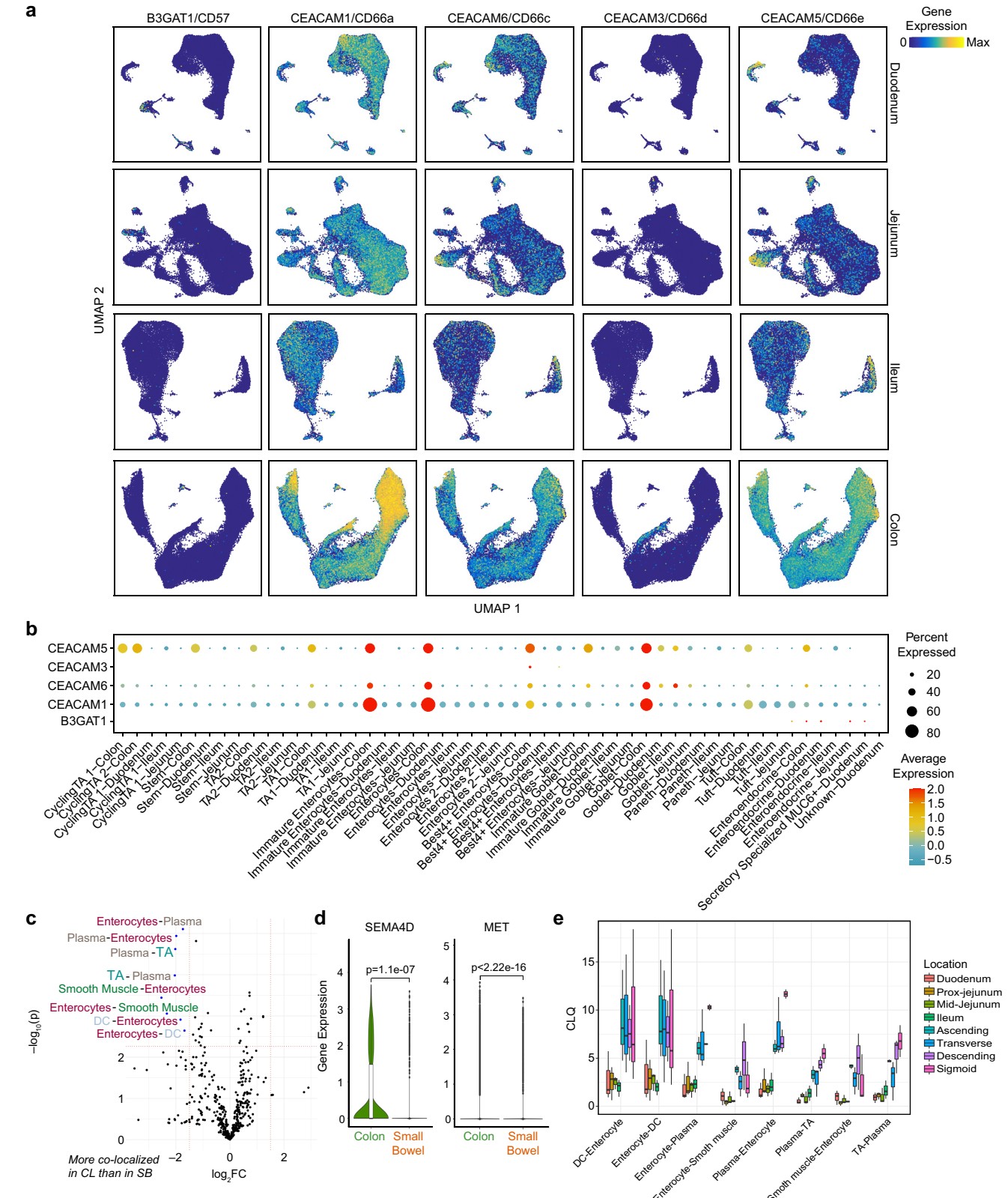

**Extended Data Fig. 8 | Pairing of CODEX multiplexed imaging and snRNA-seq for cell-cell colocalization analysis.** a) Expression of B3GAT1 and four CEACAM transcripts plotted on the UMAP manifold of epithelial cells from the duodenum, jejunum, ileum, and colon. b) Dotplot representation of the expression of B3GAT1 and four CEACAM transcripts by different epithelial cell types in different regions of the intestine. c) Large colon (CL) and small bowel (SB) show differences in cell-cell co-localization patterns; annotated cell-pairs are more colocalized in the colon compared to the small bowel (Student's T test, two-sided, corrected for multiple hypothesis testing with the Benjamini Hochberg procedure). d) SEMA4D ligand expression in plasma cells and MET receptor gene expression in TA2 cells, showing higher expression in colon than small bowel (one-sided Wilcoxon Rank-Sum Test). e) Differences in pairwise cell-type colocalization patterns across tissue locations (n = 3 samples for each location).

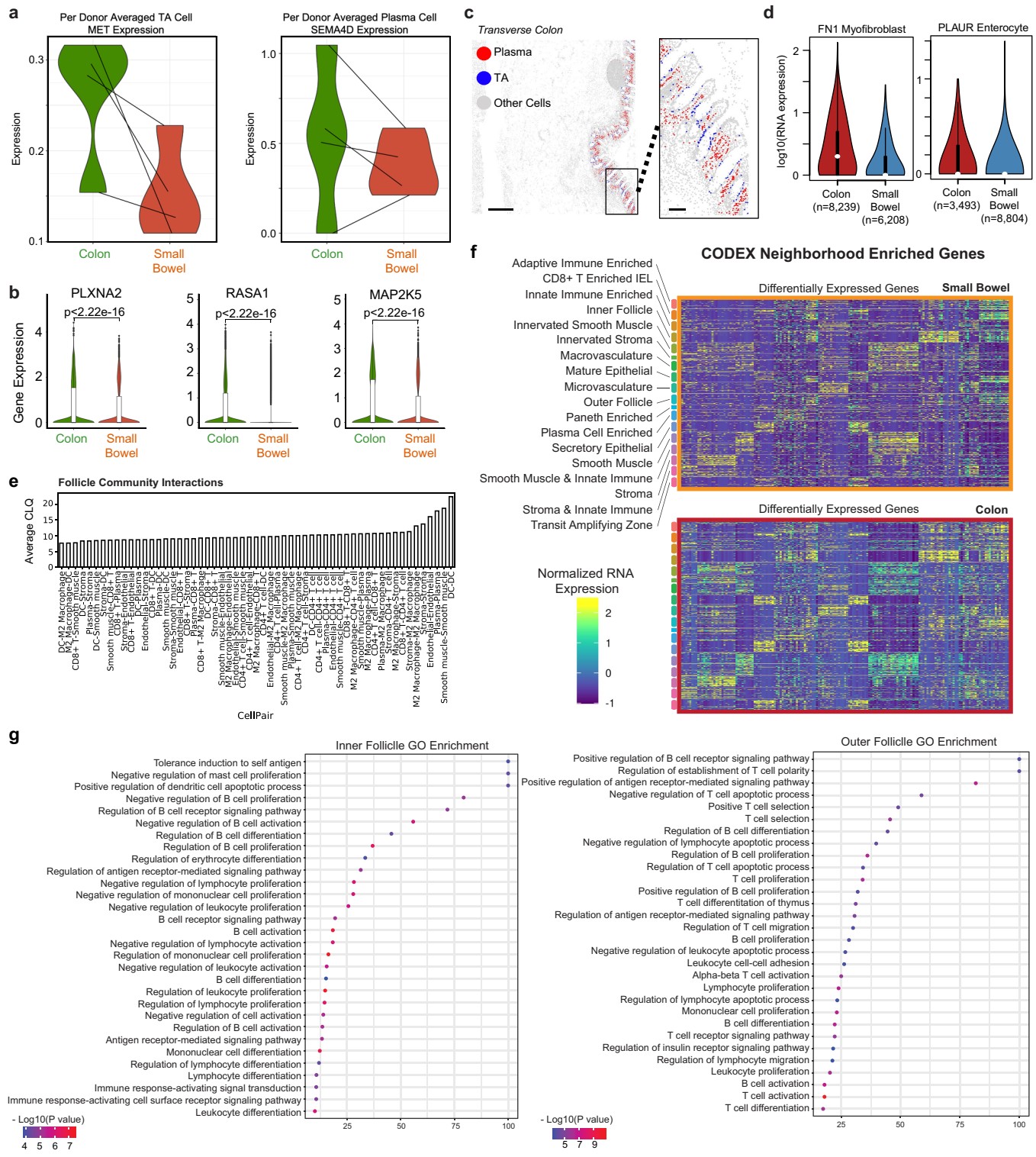

**Extended Data Fig. 9 | Integration of snRNA-seq and CODEX datasets.**
a) Average MET expression for all TA cells from a given donor and average
SEMA4D expression for all plasma cells from a given donor. b) PLXNA2, RASA1,
and MAP2K5 expression in TA2 cells in large colon (CL) and small bowel (SB)
(one-sided Wilcoxon Rank-Sum Test). c) Representative image of a donor's
transverse colon with plasma cells (red), TA cells (blue), and other cell types
(grey) highlighted with also a magnified area indicated with rectangle.
(left scale bar = 500 μm; right scale bar = 100 μm) d) Example of receptor

ligands expressed at higher levels in the colon than the small intestine (SI) for
the FN1 by myofibroblast and PLAUR by enterocyte. e) Colocalization quotient
(CLQ) for all cell types found within the follicle community. f) Heatmap of
differentially expressed genes (from MaxFuse matched snRNA-seq cells)
among individual CODEX cells, grouped based on previously determined
cellular neighbourhoods from CODEX analysis. g) Gene Ontology Enrichment
analysis for cellular neighbourhoods "inner folliclle" and "outer folliclle",
based on the differentially expressed genes shown in f.

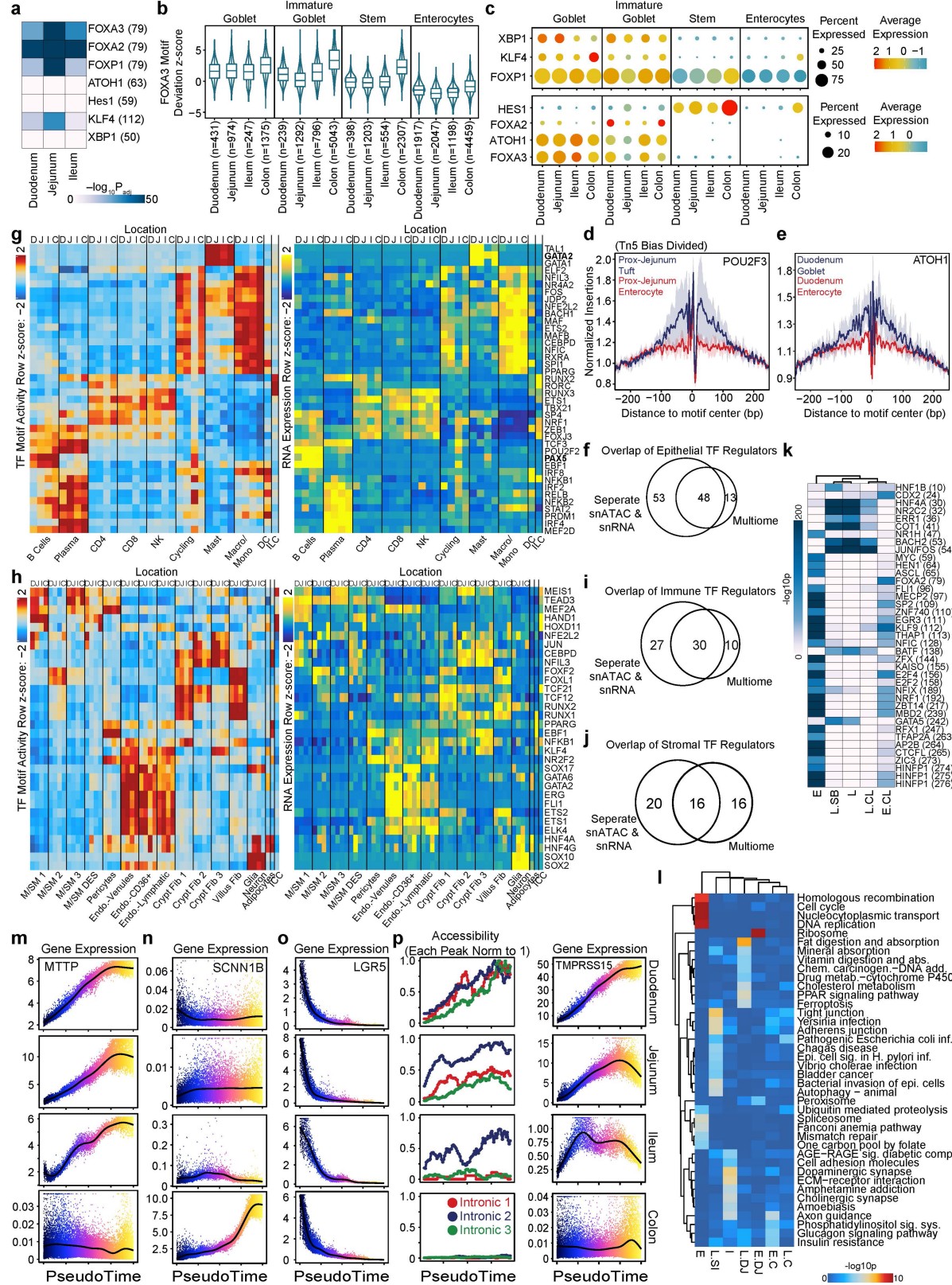

**Extended Data Fig. 10** | See next page for caption.

**Extended Data Fig. 10 | Regulatory TFs in the intestine.** a) Hypergeometric p-values of selected TF motifs in differential peaks between colon stem cells and stem cells from other regions of the intestine. Colour represents the log10 adjusted p-value computed with ArchR. b) Violin plots of motif deviation scores for FOXA3 for goblet cells, immature goblet cells, stem cells, and enterocytes in different regions of the intestine. c) Dotplot representation of expression of different transcription factors in goblet cells, immature goblet cells, stem cells, and enterocytes in different regions of the intestine. d) TF motif footprints for POU2F3 in proximal jejunum tuft cells and enterocytes. e) TF motif footprints for ATOH1 in duodenum goblet cells and enterocytes. Error bands in d and e represent the standard deviation. f) Overlap of epithelial regulators identified with the multiome data and separate snRNA and snATAC datasets.(g, h) Heatmap representation of transcription factors whose integrated gene expression was correlated with their motif activity in one region of the intestine for immune (g) and stromal (h) cell types. Row z-scores of ChromVar deviation scores are shown on the left and row z-scores of integrated TF expression are shown on the right. i, j) Overlap of immune (i) and stromal (j) regulators identified with the multiome data and separate snRNA and snATAC datasets. k) Hypergeometric p-values of TF motifs enriched in the clusters of peaks identified in b. l) Enrichment of KEGG pathways in the clusters of genes identified in c. Uncorrected p-values as determined by kegga are plotted. (m–o) Integrated gene expression of MTTP (m) and SCNN1B (n) and LGR5 (o) along the differentiation trajectory. (p) Accessibility at peaks correlated with the expression of TMPRSS15 along the differentiation trajectory in each region is plotted on the left. Each peak is normalized to the maximum accessibility along any of the trajectories. Integrated gene expression of TMPRSS15 along the differentiation trajectory in each region is plotted on the right.

# Reporting Summary

## Statistics

For all statistical analyses, confirm that the following items are present in the figure legend, table legend, main text, or Methods section.

| n/a | Confirmed | |
|---|---|---|
| ☐ | ☒ | The exact sample size (*n*) for each experimental group/condition, given as a discrete number and unit of measurement |
| ☐ | ☒ | A statement on whether measurements were taken from distinct samples or whether the same sample was measured repeatedly |
| ☐ | ☒ | The statistical test(s) used AND whether they are one- or two-sided<br>*Only common tests should be described solely by name; describe more complex techniques in the Methods section.* |
| ☒ | ☐ | A description of all covariates tested |
| ☐ | ☒ | A description of any assumptions or corrections, such as tests of normality and adjustment for multiple comparisons |
| ☐ | ☒ | A full description of the statistical parameters including central tendency (e.g. means) or other basic estimates (e.g. regression coefficient) AND variation (e.g. standard deviation) or associated estimates of uncertainty (e.g. confidence intervals) |
| ☐ | ☒ | For null hypothesis testing, the test statistic (e.g. *F*, *t*, *r*) with confidence intervals, effect sizes, degrees of freedom and *P* value noted<br>*Give P values as exact values whenever suitable.* |
| ☒ | ☐ | For Bayesian analysis, information on the choice of priors and Markov chain Monte Carlo settings |
| ☒ | ☐ | For hierarchical and complex designs, identification of the appropriate level for tests and full reporting of outcomes |
| ☐ | ☒ | Estimates of effect sizes (e.g. Cohen's *d*, Pearson's *r*), indicating how they were calculated |

*Our web collection on statistics for biologists contains articles on many of the points above.*

## Software and code

Policy information about availability of computer code

| | |
|---|---|
| Data collection | Code for generating fragments files for scATAC and counts matricies for single cell RNA was obtained from 10x genomics (go.10xgenomics.com/scATAC/cell-ranger-ATAC and https://support.10xgenomics.com/single-cell-gene-expression/software/pipelines/latest/what-is-cell-ranger). Versions for processing of the singleome datasets were cellranger-3.1.0 and cellranger-atac-1.2.0. cellranger-arc-1.0.1 was used to process the multiome samples.<br>And code for processing CODEX multiplexed imaging data (https://github.com/nolanlab/CODEX) |
| Data analysis | macs2 2.1.1.20160309 – Software for peak calling<br>R version 4.1.2 – R environment for analysis of single cell data<br>ArchR - 1.0.1 - Software for analysis of snATAC-seq data.<br>Seurat_4.1.0 - Software for analysis of snRNA-seq data.<br>DoubletFinder_2.0.3 – Software for doublet removal for scRNA-seq<br>BSgenome.Hsapiens.UCSC.hg38_1.4.3 – Package containing genomic DNA sequences<br>harmony_0.1.0 - R package used for integration of single-cell data.<br>ggplot2_3.3.5 - R package used for plotting of single cell data.<br>miloR_1.2.0 - R package for analysis of differential abundance.<br><br>Python 3.9.0 - Python version for scCODA analysis<br>sccoda version 0.1.8 - Python package for analysis of differential abundance.<br>scanpy Python package (version 1.9.1) - analyzing single cell CODEX data<br>R version 4.2.0 – R version for GO enrichment analysis<br>limma_3.52.2 – Software used for GO enrichments<br><br>R version 4.0.2 – R environment for initial analysis of single-cell data<br>Seurat_4.0.1 – Software for initial analysis of snRNA-seq data. |

Custom code for analyzing the snATAC and snRNA data is available on GitHub (https://github.com/winstonbecker/scColonHuBMAP).

Code for clustering (https://github.com/nolanlab/vortex), the code for transferring cell type labels with STELLAR (https://github.com/snap-stanford/stellar), code for neighborhood analysis (https://github.com/nolanlab/NeighborhoodCoordination), and code for tissue schematics (https://github.com/nolanlab/TissueSchematics) is available on github.

For manuscripts utilizing custom algorithms or software that are central to the research but not yet described in published literature, software must be made available to editors and reviewers. We strongly encourage code deposition in a community repository (e.g. GitHub). See the Nature Portfolio guidelines for submitting code & software for further information.

## Data

Policy information about availability of data

All manuscripts must include a data availability statement. This statement should provide the following information, where applicable:

- Accession codes, unique identifiers, or web links for publicly available datasets
- A description of any restrictions on data availability
- For clinical datasets or third party data, please ensure that the statement adheres to our policy

All of the published datasets in this study can be visualized and assessed through a website portal (https://portal.hubmapconsortium.org/). We have created a landing page with links to all the raw dataset IDs and the HuBMAP ID for this Collection is HBM692.JRZB.356 and the DOI is:10.35079/HBM692.JRZB.356. Supplemental Table 10 also lists all the dataset IDs within the HuBMAP portal where all raw datasets are stored that can be downloaded and also viewed in a processed state. We also provide the processed and annotated single-cell CODEX datasets with labeled cell types, neighborhoods, communities, tissue units, and also protein expression via Dryad doi https://doi.org/10.5061/dryad.pk0p2ngrf.

# Field-specific reporting

Please select the one below that is the best fit for your research. If you are not sure, read the appropriate sections before making your selection.

☒ Life sciences          ☐ Behavioural & social sciences          ☐ Ecological, evolutionary & environmental sciences

For a reference copy of the document with all sections, see nature.com/documents/nr-reporting-summary-flat.pdf

# Life sciences study design

All studies must disclose on these points even when the disclosure is negative.

| | |
|---|---|
| Sample size | Sample sizes were set to maximize diversity across 8 regions of the intestine while also capturing sufficient depth in single-cell techniques that are able to capture differences in composition between cell types at resolution using multiome and CODEX multiplexed imaging modalities. |
| Data exclusions | All datasets generated that did not fail experimentally (e.g. overloaded sample) were included in the study. |
| Replication | Experimental assays were not replicated in this study, but the same measurements were made for all 9 donors at all 8 regions of the intestine. For CODEX multiplexed imaging 8 of the 9 donors included the full panel and were used for generating main figure analysis. snRNAseq and snATACseq were completed on 3/9 samples and multiome analysis was completed on 6/9 samples. |
| Randomization | Randomization was not relevant for this study as there were not multiple groups requiring randomization. |
| Blinding | No blinding was performed in this study because there were no experimental groups. |

# Reporting for specific materials, systems and methods

We require information from authors about some types of materials, experimental systems and methods used in many studies. Here, indicate whether each material, system or method listed is relevant to your study. If you are not sure if a list item applies to your research, read the appropriate section before selecting a response.

## Materials & experimental systems

| n/a | Involved in the study |
|---|---|
| ☐ | ☒ Antibodies |
| ☒ | ☐ Eukaryotic cell lines |
| ☒ | ☐ Palaeontology and archaeology |
| ☒ | ☐ Animals and other organisms |
| ☐ | ☒ Human research participants |
| ☒ | ☐ Clinical data |
| ☒ | ☐ Dual use research of concern |

## Methods

| n/a | Involved in the study |
|---|---|
| ☒ | ☐ ChIP-seq |
| ☒ | ☐ Flow cytometry |
| ☒ | ☐ MRI-based neuroimaging |

# Antibodies

**Antibodies used**

We provide a detailed antibody information and metadata for all the antibodies used for CODEX (>60) within Supplementary Table 7 of the submission. Here is it below:

antibody_name rr_id uniprot_accession_number lot_number dilution conjugated_cat_number conjugated_tag Validation

Anti-MUC-2 antibody AB_791261 Q02817 7/5/19  1/200 custom 63-Alexa 488 validated for IHC/IF by manufacturer

Anti-MUC-1/EMA antibody AB_2864392 P15941 10/7/20  1/100 custom 15-Alexa 488 validated for IHC/IF by manufacturer

Anti-Synaptophysin antibody AB_10010435 P08247 1/22/20  1/100 custom 69-Alexa 488 validated for IHC/IF by manufacturer

Anti-CD15 antibody AB_395800 P22083 11/2/20  1/200 custom 70-Alexa 488 validated on fresh frozen human lymphoid tissue; Eur. J. Immunol. 2021

Anti-ITLN1 antibody AB_2129678 Q8WWA0 1/22/20  1/200 custom 72-Alexa 488 validated for IHC/IF by manufacturer

Anti-Vimentin antibody AB_393716 P08670 3/20/18  1/200 custom 7-Alexa 488 validated on fresh frozen human lymphoid tissue; Eur. J. Immunol. 2021

Anti-CD11c antibody AB_395792 P20702 1/22/18  1/200 custom 44-Alexa 488 validated in lab with fresh frozen human intestine tissue with negative and positive controls

Anti-BCL-2 antibody AB_2864404 P10415 7/30/19  1/200 custom 41-Alexa 488 validated for IHC/IF by manufacturer

Anti-CD38 antibody AB_2561794 P28907 4/13/18  1/500 custom 66-Alexa 488 validated on fresh frozen human lymphoid tissue; Eur. J. Immunol. 2021

Anti-a-SMA antibody AB_2572996 P62736 9/10/19  1/200 custom 8-Alexa 488 validated for IHC/IF by manufacturer

Anti-CD66 antibody AB_394166 P13688 3/20/18  1/200 custom 5-Alexa 488 validated on fresh frozen human lymphoid tissue; Eur. J. Immunol. 2021

Anti-CD68 antibody AB_1089058 P34810 8/12/19  1/100 custom 48-Alexa 488 validated for IHC/IF by manufacturer

Anti-CD7 antibody AB_1659214 P09564 3/23/18  1/200 custom 58-Alexa 488 validated on fresh frozen human lymphoid tissue; Eur. J. Immunol. 2021

Anti-CD45RO antibody AB_314418 P08575 8/12/19  1/100 custom 36-Alexa 488 validated for IHC/IF by manufacturer

Anti-Collagen IV antibody AB_305584 P02462 4/13/18  1/100 custom 33-Alexa 488 validated on fresh frozen human lymphoid tissue; Eur. J. Immunol. 2021

Anti-SOX9 antibody AB_2665492 P48436 7/5/19  1/100 custom 26-Cy3 validated for IHC/IF by manufacturer

Anti-GATA-3 antibody AB_2108590 P23771 3/23/18  1/100 custom 55-Cy3 validated for IHC/IF by manufacturer

Anti-Lefty antibody AB_2797977 O00292 2/26/20  1/100 custom 71-Cy3 validated in lab with fresh frozen human intestine tissue with negative and positive controls

Anti-CHGA antibody AB_2864388 P10645 7/25/19  1/100 custom 2-Cy3 validated for IHC/IF by manufacturer

Anti-CD4 antibody AB_2561907 P01730 1/22/18  1/200 custom 28-Cy3 validated on fresh frozen human lymphoid tissue; Eur. J. Immunol. 2021

Anti-HLA-DR antibody AB_2562826 P04233 7/24/18  1/200 custom 11-Cy3 validated on fresh frozen human lymphoid tissue; Eur. J. Immunol. 2021

Anti-CD44 antibody AB_312953 P16070 1/22/20  1/200 custom 14-Cy3 validated for IHC/IF by manufacturer

Anti-CD3 antibody AB_314056 P07766 3/23/18  1/200 custom 20-Cy3 validated on fresh frozen human lymphoid tissue; Eur. J. Immunol. 2021

Anti-CD90 antibody  AB_940393 P04216 3/23/18  1/200 custom 68-Cy3 validated on fresh frozen human lymphoid tissue; Eur. J. Immunol. 2021

Anti-CD21 antibody  AB_11219188 P20023 1/22/18  1/500 custom 21-Cy3 validated on fresh frozen human lymphoid tissue; Eur. J. Immunol. 2021

Anti-CD57 antibody AB_535988 Q9P2W7 2/22/18  1/100 custom 30-Cy3 validated on fresh frozen human lymphoid tissue; Eur. J. Immunol. 2021

Anti-CD34 antibody AB_1732014 P28906 12/8/19  1/100 custom 80-Cy3 validated on fresh frozen human lymphoid tissue; Eur. J. Immunol. 2021

Anti-CD36 antibody AB_395846 P16671 2/13/18  1/200 custom 49-Cy3 validated in lab with fresh frozen human intestine tissue with negative and positive controls

Anti-Cytokeratin antibody AB_2616960 Q04695 2/22/18  1/200 custom 67-Cy3 validated on fresh frozen human lymphoid tissue; Eur. J. Immunol. 2021

Anti-CD117 antibody AB_2131466 P10721 2/9/18  1/200 custom 74-Cy3 validated on fresh frozen human lymphoid tissue; Eur. J. Immunol. 2021

Anti-CD19 antibody AB_395810 P15391 3/13/20  1/200 custom 75-Cy3 validated on fresh frozen human lymphoid tissue; Eur. J. Immunol. 2021

Anti-CD45 antibody AB_314390 P08575 5/21/18  1/500 custom 56-Cy3 validated on fresh frozen human lymphoid tissue; Eur. J. Immunol. 2021

Anti-CD69 antibody AB_314837 Q07108 4/13/18  1/500 custom 24-Cy3 validated for IHC/IF by manufacturer

Anti-Somatostatin antibody AB_2890053 P61278 2/3/20  1/100 custom 57-Cy3 validated for IHC/IF by manufacturer

Anti-CD49a antibody AB_1236385 P56199 2/17/20  1/50 custom 46-Cy3 validated for IHC/IF by manufacturer

Anti-CD161 antibody AB_1501090 Q12918 1/22/20  1/200 custom 53-Cy3 validated for IHC/IF by manufacturer

Anti-MUC6 antibody AB_2864391 Q6W4X9 8/23/19  1/100 custom 81-Cy5 validated in lab with fresh frozen human intestine tissue with negative and positive controls

Anti-CD31 antibody AB_395837 P16284 2/22/18  1/200 custom 42-Cy5 validated on fresh frozen human lymphoid tissue; Eur. J. Immunol. 2021

Anti-CD49f antibody AB_2296273 P23229 3/23/18  1/50 custom 51-Cy5 validated on fresh frozen human lymphoid tissue; Eur. J. Immunol. 2021

Anti-CDX2 antibody  AB_2864406 Q99626 11/22/19  1/200 custom 62-Cy5 validated for IHC/IF by manufacturer

Anti-CD127 antibody AB_10718513 P16871 2/13/18  1/100 custom 61-Cy5 validated for IHC/IF by manufacturer

Anti-CD8 antibody AB_1877104 P01732 10/7/20  1/200 custom 43-Cy5 validated on fresh frozen human lymphoid tissue; Eur. J. Immunol. 2021

Anti-CD16 antibody AB_395804 P08637 2/22/18  1/25 custom 52-Cy5 validated on fresh frozen human lymphoid tissue; Eur. J. Immunol. 2021

Anti-CD123 antibody AB_395455 P26951 2/13/18  1/25 custom 59-Cy5 validated on fresh frozen human lymphoid tissue; Eur. J.

Immunol. 2021
Anti-CD279 (PD-L1) antibody AB_2864409 Q15116 2/22/18  1/50 custom 79-Cy5 validated for IHC/IF by manufacturer
Anti-NKG2D (CD314) antibody AB_492956 P26718 8/29/19  1/100 custom 77-Cy5 validated for IHC/IF by manufacturer
Anti-CD206 antibody AB_571923 P22897 3/29/19  1/200 custom 25-Cy5 validated for IHC/IF by manufacturer
Anti-aDefensin 5 antibody AB_2864387 Q01523 8/30/19  1/200 custom 60-Cy5 validated for IHC/IF by manufacturer
Anti-CD138 antibody AB_2561790 P18827 3/29/19  1/100 custom 76-Cy5 validated for IHC/IF by manufacturer
Anti-CK7 antibody AB_2864389 P08729 7/25/19  1/200 custom 3-Cy5 validated for IHC/IF by manufacturer
Anti-PGP9.5 antibody AB_2890054 P09936 1/22/20  1/200 custom 23-Cy5 validated for IHC/IF by manufacturer
Anti-Podoplanin antibody AB_1595511 Q86YL7 4/13/18  1/100 custom 32-Cy5 validated on fresh frozen human lymphoid tissue; Eur. J. Immunol. 2021
Anti-CD56 antibody  AB_395904 P13591 2/22/18  1/100 custom 29-Cy5 validated for IHC/IF by manufacturer
Anti-CD154 antibody AB_314825 P29965 8/22/18  1/200 custom 38-Cy5 validated for IHC/IF by manufacturer
Anti-Ki67 antibody AB_396287 P46013 4/13/18  1/25 custom 6-Cy5 validated on fresh frozen human lymphoid tissue; Eur. J. Immunol. 2021
Anti-CD163 antibody AB_1088991 Q86VB7 3/29/19  1/100 custom 45-Cy5 validated for IHC/IF by manufacturer
Anti-CD294 antibody AB_10639863 Q9Y5Y4 3/23/18  1/100 custom 65-Cy5 validated for IHC/IF by manufacturer
Anti-CD25 antibody AB_1107617 P01589 7/18/18  1/100 custom 57-Cy3 validated for IHC/IF by manufacturer
Anti-OLFM4 antibody AB_2785318 Q6UX06 10/7/20  1/100 custom 65-Cy3 validated for IHC/IF by manufacturer
Anti-Lysozyme antibody AB_776115 P61626 7/16/20  1/25 custom 81-Cy3 validated in lab with fresh frozen human intestine tissue with negative and positive controls
Anti-CD33 antibody AB_314342 P20138 10/7/20  1/100 custom 23-Cy3 validated for IHC/IF by manufacturer
Anti-FAP antibody AB_2532994 Q12884 2/22/20  1/25 custom 79-Cy5 validated for IHC/IF by manufacturer
Anti-CD98 antibody AB_2302070 P08195 7/16/20  1/25 custom 55-Cy5 validated for IHC/IF by manufacturer
Anti-CD147 antibody AB_314586 P35613 7/16/20  1/100 custom 71-Cy5 validated for IHC/IF by manufacturer

| Validation | We provide a detailed antibody information and metadata for all the antibodies used for CODEX (>60) within Supplementary Table 7 and methods of validation from source vendor, prior publications, and within our primary data and in the above box. |
| --- | --- |

# Human research participants

Policy information about studies involving human research participants

| Population characteristics | Individuals at Washington University in St. Louis were identified to participate in this study. We analyzed eight sections from nine individuals: seven European-ancestry (five males and two females), one African American male, and one African American female. Age ranges were from 24 to 78 years. |
| --- | --- |
| Recruitment | Patients without known intestinal diseases were recruited for this study. Participants were recruited for research after next-of-kin consented for organ donation. Given that presentation for organ donation after death is random, we do not expect biases to be present. Our experience is that the demographic composition of recruited participants largely reflects the composition of the local population (St. Louis, MO). |
| Ethics oversight | This study complies with all relevant ethical regulations and was approved by the Washington University Institutional Review Board and the Stanford University Institutional Review Board. Human bowel tissues were procured from deceased organ donors. Written informed consent was obtained from next-of-kin for all donor subjects.Participants were recruited for research after next-of-kin consented for organ donation. Given that presentation for organ donation after death is random, we do not expect biases to be present. Our experience is that the demographic composition of recruited participants largely reflects the composition of the local population (St. Louis, MO). |

Note that full information on the approval of the study protocol must also be provided in the manuscript.

