## [Peer Review File · Nature]

Manuscript Title: Organization of the Human Intestine at Single Cell Resolution

Reviewer Comments & Author Rebuttals

Reviewer Reports on the Initial Version:

Referees' comments:

Referee #1 (Remarks to the Author):

Proposed gap:

Discussed in lines 21-24 : “Moreover, these cell types are spatially organized into different “neighborhoods” across these intestinal regions, with both the composition of these neighborhoods and molecular phenotypes of underlying cellular types varying in relatively unknown ways across these anatomical regions.”

Novelty:

Combine RNA measurements with chromatin measurements and protein measurements in space to characterize cellular and molecular distribution of circuits along the human gut axis

General positive comments:

- The datasets cover multiple data modalities (spatial, transcriptomic, chromatin accessibility), sample many regions of the intestine, and come from several individuals.
- Analysis of the individual datasets uncovers cell types, cell states, cell neighborhoods, differentiation trajectories, and gene regulatory networks.
- The CODEX antibody marker panel was greatly expanded to detect major gut epithelial cell subtypes.
- The integration of the scRNA-Seq and scATAC-Seq is carefully performed.

The cohort chosen for the study is small covering only 4 individuals. Furthermore, the cohort is not balanced well for age, sex, and race. The CODEX data feel disconnected from the RNA-seq and ATAC-seq data presented later in the paper and the two datasets do not build well off one another. Very little novelty is presented in the first 4 figures where the conclusions drawn are underwhelming. The statistically significant observations made in these first 4 figures are almost all already known/reported. The data presented in figures 5 and 6 are intriguing and figure 5 can be extended to include TF analysis of immune and stromal subsets in addition to the reported data on epithelial cells. The paper would also benefit from better utilizing the different regions of the intestine collected when presenting the analysis of the CODEX data. Focussing the majority of analysis on SI as a whole vs Colon as a whole is overly simplistic and sells the data short. The authors should make a

better effort to tie the observations made via the spatial analysis using CODEX with the second part of the paper using RNA-seq and ATAC-seq to dive deeper into the molecular networks across the length of the intestine.

Major comments:

- Cell proportion data from CODEX are difficult to interpret given the heterogeneity in tissue sectioning between intestine regions within an individual as well as across individuals (shown in Figs S2 and S6). The only clear signals seem to be expected ones such as more CD8 T cells in the small intestine and more enterocytes in the small intestine given that the small intestine has villi and the colon does not.
- Tissue sections presented for CODEX data are not comparable in their orientation and inclusion of all layers of the intestine (shown in Figs S2 and S6) within the small or large intestine for one individual as well as across the three individuals presented. How did the authors choose which sections of duodenum, jejunum, ileum to process from the organ donors? Can the authors place these gut sections in any of the common coordinate frameworks provided atlas consortia (e.g. HubMAP, HCA, ...)?
- Figure 1 and 2 should be combined with most of figure 1 data moved to the supplement as the observations in figure 1 are not so easily interpretable for reasons above nor the conclusions novel outside of perhaps figure 1F.
- For follicle based CODEX observations was a follicle present in each section of each of the 8 segments for an individual for all 3 individuals? It is unclear how many follicles are being studied for these analyses.
- In figures 1-3 where CODEX data is presented the authors do not seem to leverage the different regions of the small intestine and colon in the analysis presented. Comparisons are made at the level of all SI and all Colon but it is less clear which segments within those tissues are driving the differences. Is this limitation due to the heterogeneity in tissue sections for the various sections taken making it difficult to find signal allowing to make conclusions about the duodenum vs jejunum or sigmoid vs transverse colon for example?
- For the categorization of the CODEX data into neighborhoods, communities, and major communities how do the authors control for the limitation of a 7-micron cross section not being able to accurately represent the true spatial distribution/orientation of cells to one another since it is highly unlikely for any given cut to be in the perfect orientation such that one can be confident that any two cells or neighborhoods or communities are truly adjacent in space within the tissue? Controlling for transverse vs cross-sectional cuts and any angle in the cut that is not in the same plane along the length of the tissue axis being cut. This can play a role in conclusions drawn such as in Figure 3A where the suggestion is the immune major communities occupy different spaces in the small vs large intestine.
- The authors jump from Fig 4 to Fig 5 prior to completing discussion of data in Fig 4. Furthermore, there is no clear logic as to why Fig 5 focuses on enteroendocrine cells in A-C and then goes on to do a broad TF analysis which is the title and focus of the figure.
- Why do the authors limit the regulatory TF analysis in Fig 4 to only epithelial cells? Can this not be applied to immune and stromal cell populations along the length of the intestine? This would make a more comprehensive figure and can substitute for the inclusion of the enteroendocrine analysis in A-

C.

- The gut data collection is excellent, but the data sets are currently best suited as a reference for future studies of the gut. The authors should give examples of how this atlas can contextualize other existing or future single cell and spatial studies of healthy and disease gut. This can be at the level of genetics (e.g. GWAS genes for IBD, CD), or comparisons to other disease gut datasets (cell types, compositions etc...)
- The authors should contextualize their findings with other existing gut atlases, for example: A single-cell survey of the small intestinal epithelium (Nature, 2017), Cells of the human intestinal tract mapped across space and time (Nature, 2021), A proximal-to-distal survey of healthy adult human small intestine and colon epithelium by single-cell transcriptomics (bioRxiv, 2021).
- Within each atlas (spatial, transcriptomic, chromatin accessibility), most of the biological findings into gut biology are already known. The findings that different gut regions have different compositions and organizations is expected, and none of the new findings are experimentally validated.
 - o Validate by IHC that there are MUC5B+ and MUC6+ secretory cells specific to the duodenum.
 - o Validate SEMA4D expression in plasma cells and MET expression in TA2 cells in colon vs small intestine.
 - o What gut cell types are the Insulin-like peptide 5 L cells signalling to, and is the cognate receptor expressed?
 - o Validate potential master regulator TFs (Fig 5D): RUNX1 and RUNX2 in Tuft cells, FOXA3 and ATOH1 in goblet cells, and ZBTB18 in Paneth cells
- Are there further ways in which the single cell transcriptomic and spatial data can be integrated, beyond the ligand-receptor interaction analyses?
- The spatial data collected here is unique, and one valuable contribution would be a power analysis of the data. How does the detection of cell types and cell neighborhoods depend on the number of individuals and tissue sizes / FOVs samples? This can be done empirically through downsampling of the data.

Minor comments:

- The abstract focuses only on the colon directly (the authors make no direct mention of the small intestine) although the paper presents data on both the small intestine and colon. The abstract should be amended to reflect this.
- Fig 1
 - o It is unclear which 3 of the 4 donors were used to generate the CODEX data
 - o Data in 1F should be split into the three major cell types like in 1D. Immune cells are mobile and less geographically restricted in tissues whereas enterocytes are geographically restricted to form the architecture of the organ (ie crypts and villi etc). This makes comparing different cell lineages on the same graph less meaningful.
 - o The authors use their geometric deep learning method STELLAR (Annotation of Spatially Resolved Single-cell Data with STELLAR (bioRxiv, 2021)) to annotate several of the spatial datasets. More detail is needed for the reader to understand how well this works. In particular, how was the model trained and tested (which individuals and which tissue regions), and did the model discover any new cell types or classify any as unknown? This is of particular importance given how different the tissue

samples appear across the 4 patients (Supp Fig 6).

- Fig 2

- o Page 8 line 13 states the plasma cell enriched neighborhood has co-enrichment with CD4 T cells and APCs. This observation would be strengthened by a visual plot of these cell types in space to appreciate this enrichment as was done for same cell enrichment in Figure 1E.

- o Page 8 line 31 should call out Figure 2D not Figure 2C

- o The heatmap presented in Fig 2F is confusing. It is unclear how the authors defined cell composition conservation. Given the authors mostly discuss the differences in cell neighborhoods between SI and CL, it may be simpler to present just the neighbor composition conservation, or at least keep it in a separate figure from the cell composition conservation.

- Fig 3

- o In Fig 3B, the tSNE embeddings are different between neighborhood and major community vs community. All three tSNE embeddings should be the same as they are built from the neighborhood vectors. Without this, we cannot see how well cell neighborhood vectors cluster into the communities.

- o Fig 3D

Labeled as epithelial spatial context map, but I think the authors included both the epithelial and immune major community in this analysis.

The box colors make it confusing as to what are corresponding regions. I think the green and black boxes (left) and the orange and blue boxes (right) refer to the same regions respectively.

Please add a legend for the cell neighborhood colors.

Although the authors have a publication on tissue schematics and spatial context maps, they should describe the method here as it is novel and not trivial.

- Fig 4

- o What are the markers for “Epithelial” in 4F when all cells here are presumably epithelial cells

- o Page 16 line 24 states one of the donors T cells cluster apart from the others. What figure shows this? Explanation is age may be the explanation. Did authors observe similar outlier behavior for this individual in CODEX data for T cells?

- o Authors identify a duodenum specific subset of Muc6 and Tff2 expressing cells. These two genes are highly expressed in the stomach (see GTEx data set). Is it possible duodenum sample preps have contaminating cells from the stomach?

- o The authors also note the MUC5B+ secretory cells in the duodenum are surprising. The authors should consider validating the presence of this population. Interestingly, the human protein atlas does not report MUC5B expression in the duodenum

(<https://www.proteinatlas.org/ENSG00000117983-MUC5B/tissue>)

- Fig 6

- o Fig 6F heatmap needs four labels for the duodenum, jejunum, ileum, and colon regions.

- Please show how cell type composition and neighborhoods within a spatial tissue region vary across individuals. This is not shown in Supp Fig 7,8 but Supp Fig 7C comes close.

- It is unclear how the authors tested for statistically significant differences in cell type composition across samples, and this is important for many of their claims. The test should control for the compositional nature of the data (good references include Differential abundance testing on single-cell data using k-nearest neighbor graphs (Nature Biotechnology, 2021), scCODA is a Bayesian model for compositional single-cell data analysis (Nature Communications, 2021).

- What are ITLN enterocytes?

- Can the authors provide further detail on how they decided how many cell neighborhoods to define in their spatial data? A tSNE of the cell composition vectors for all the cells may make it clearer how distinctly single cells separate into the cell neighborhoods.
- Why is the most diffuse same cell distance defined as taking the number of total cells of a given cell type divided by the total area of the region. Intuitively I would think the most diffuse same cell distance should decrease as the total cells of a given cell type increase, and should increase as the total region area increases.
- Why was the identification of the cell neighborhoods performed using all eight tissue regions together, rather than analyzing the regions individually? My concern is that a cell neighborhood unique to a tissue region would be missed in the joint analyses done here.
- In Fig 1F, why is a same-cell density measurement reported for Paneth cells in the colon, given their expected absence? Is the annotation of Paneth cells in the colon a result of considering all gut tissue regions together rather than individually when identifying cell types?
- The CD8+ T enriched IEL neighborhood in Fig 2B consists of CD8+ T cells and enterocytes, TA, and cycling TA cells? As TA cells in the small intestine tend to be at the base of the crypt and enterocytes tend to be in the villus, how are these epithelial cells all in the same cell neighborhood with the CD8+ T cells?
- Why is the frequency of inner follicles not highest in the mid-jejunum / ileum (Fig 2D), as that is where we expect Peyer's patches? Do the CODEX measurements identify the Peyer's patches?
- Can the authors please comment on the limitations of the CODEX measurements with regards to cell type detection. What expected cell types are missing due to rarity or lack of appropriate antibody markers, e.g. ILCs, neurons, Tregs, Th1, Th2, M cells, ...?
- "Using this intuitive formalism, we observe crosstalk between stromal and smooth muscle cell types and structures, which are in turn isolated from epithelial and immune components that are more entwined with one another." Crosstalk generally indicates functional interactions between cell types, whereas the data only can indicate co-associations. The authors can strengthen the case for crosstalk by looking for an enrichment of ligand-receptor interactions within the cells of a neighborhood or community (leveraging their scRNA-Seq data).
- When introducing the single cell RNA-Seq and ATAC-Seq analysis in the text, it was unclear that the 10x Multiome assay was applied. I surmised this from the methods, but this should be clarified in the text.
- Single nucleus RNA-Seq often has high levels of ambient RNA detected, confounding analysis. Was the level of ambient RNA ever assessed, perhaps by looking at the specificity in expression of lineage markers, and were any methods run to correct ambient RNA or remove empty droplets beyond what is incorporated in Cellranger?
- Please include in the data availability statement what kind of data is available (raw, processed, images) and whether there are portals to visualize and query the final datasets (spatial tissue maps, UMAP embeddings).
- When doing differential expression analysis for genes expressed in colon vs small bowel (e.g. Fig 4G,H and Supp Fig 16), the 4 tissue donors should be treated as the independent replicates and not the individual single cells. Using Wilcox with single cells as the replicates leads to over inflated p-values. A pseudobulk DE gene method would be appropriate: Confronting false discoveries in single-cell differential expression (Nature Communications, 2021), muscat detects subpopulation-specific state transitions from multi-sample multi-condition single-cell transcriptomics data (Nature Communications, 2020).

- When introducing LD score regression, the authors should provide explanation as to what confounding biases (cryptic relatedness, population stratification) this method controls for.
- The authors run LD score regression using the chromatin accessibility data. Did they also try using the gene expression data? Is there a way to combine results from analyses done using both modalities?

Referee #2 (Remarks to the Author):

In this manuscript Hickey et al. present a spatially-resolved cell atlas of the human gastrointestinal tract. They use both single nucleus RNAseq and single cell ATACseq to identify cell states and regulators, as well as CODEX multiplexed imaging, to identify neighborhoods of co-localized cell types. This forms a comprehensive resource that will serve as an important reference map for future human studies. The amount of collected data is impressive. The manuscript is very well written and manages to highlight several interesting and novel findings in a highly approachable manner. I particularly liked the spatial analyses that highlighted some intriguing differences between the small and large intestine including the differences in CD8+ cell abundances, the differential localization of immune cells in the crypt apex/villus vs. crypt base, as well as the unbiased identification of regulators along differentiation trajectories. I have a few, generally minor comments that should be addressed in a revision:

- 1)The spatial analysis of Figure 3 is very interesting, however the authors should add graphs showing the differential localization of distinct cell types in the crypt/villus zone in the small intestine and in the crypt base/crypt apex zones in the large intestine. For example, are CD8+ cells, CD4+ cells, neutrophils and dendritic cells more abundant in the villus compared to the crypt in the small intestine? Similarly for the plasma cell co-localization with transit amplifying cells, it would be helpful to show this more explicitly, ideally with some blow-up images of example crypt/villus zones.
- 2)As the authors note, secretory cell fractions increase dramatically from the small to the large intestine. A major open question is what dictates these differential differentiation outcomes – are there inherent differences in the stem cells between the two segments, or rather differences in niche signals? Crypt stem cell differentiation into secretory or absorptive cells, in mice at least, depends on notch activity – notch active cells differentiate into absorptive cells whereas notch-low cells differentiate into secretory cells. Do the authors find differential activity of Notch-related pathway TFs such as Hes1 and Atoh1 between the small and large intestinal stem cells? Are there other differences in the stem cell TF activity profiles that could prime them to this skewed differentiation towards secretory cells in the large intestine?
- 3)The authors make a distinction between CD57 enterocytes, CD66 enterocytes and another class of enterocytes in the CODEX data, with differing frequencies along the small-to-large intestinal axis. It would be helpful to show the expression of CD57 and CD66 mRNAs on the single cell atlas of Figure 4. Do these sub-types of enterocytes have differential gene expression programs or crypt-villus localization patterns?
- 4)L cells are usually defined by their expression of GCG (GLP1) rather than PYY.
- 5)What are the differential markers of the four types of EC cells identified?
- 6)Figure 4E – please add legends, the colors are unclear.
- 7)Figure 5 – on page 23 row 10 there is a typo – should be right instead of left.

- 8)Page 28 row 3 – ‘we analyzed many different regions’ – rephrase to tone down.
9)I could not find the data in the portal mentioned in the data availability statement.
10)Page 46, rows 27-28 – please use an objective criterion to filter out doublets.

Referee #3 (Remarks to the Author):

The image analysis presented in the paper consists of three components:

1. Cell segmentation, using two different published tools: the CODEX Segmenter and CellVisionSegmenter.
2. Cell classification, using STELLAR, presented in a co-submitted manuscript f
3. Statistical analysis of local neighborhoods at different scales.

It is stated in the paper that both the CODEX Segmenter and CellVisionSegmenter are used, and references to a GitHub page are included, but no information on where and for what parts of the datasets the two respective cell segmentation methods are used, nor is there any information on how the methods were tuned and what effect faulty cell segmentation may have on the downstream analysis. All segmentation methods have their limitations, especially in parts of the tissue where cell density is high. If different methods are used on different data sets it may very much affect downstream analysis. A better description of when and where different segmentation methods were used, as well as their limitations and effect on cell classification and neighborhood analysis should be discussed.

The co-submitted paper describing STELLAR is well written and describes the combined use of per-cell marker intensity measurements and each cell's neighbors as input to a graph neural network for cell classification. It is trained using a manually annotated training set, and it is nice in the sense that it also includes a confidence measurement making it possible to find cell deviating from those present in the training set, making it possible to find new cell types.

However, STELLAR is not unique in using graph convolutional neural networks to learn latent low-dimensional cell or cell neighborhood representations that jointly capture spatial and molecular similarities. The idea has been published before:

Hu et al. "SpaGCN: Integrating gene expression, spatial location and histology to identify spatial domains and spatially variable genes by graph convolutional network." *Nature methods* 18.11 (2021): 1342-1351.

Solorzano et al. "Machine learning for cell classification and neighborhood analysis in glioma tissue." *Cytometry Part A* 99.12 (2021): 1176-1186.

Partel et al. "Spa2vec: Unsupervised representation of localized spatial gene expression

signatures." *The FEBS journal* 288.6 (2021): 1859-1870.

A discussion of similarities and differences as compared to these methods is missing.

It is worth noting that, also this paper completely lacks a description of cell segmentation approaches and the potential impact of faulty cell segmentation results on downstream cell classifications. Furthermore, a discussion on the method's sensitivity to variation in staining intensities often observed between samples is missing.

The final cluster approach is not novel, and has been published before in different versions, with one popular implementation provided (together with many other spatial statistics tools) in the SquidPy toolbox

Palla, G., Spitzer, H., Klein, M. et al. Squidpy: a scalable framework for spatial omics analysis. *Nat Methods* (2022). <https://doi.org/10.1038/s41592-021-01358-2>

Overall, I believe the paper as a whole is interesting and well written, and apart from a discussion of limitations of cell segmentation, the technical soundness is good. In many ways, it is a nice application of a number of known methods to a nice dataset to explore variations in cell types and cell communities in the intestine. The novelty is however limited.

Referee #4 (Remarks to the Author):

Hickey, Becker, Nevins et al. provide a detailed single cell description of colon tissues from four patients across eight different tissue sites. Healthy tissue samples from autopsies are a rare but important resource for the understanding of human tissue and homeostasis. Utilizing CODEX multiplexed imaging they map the single cellular organization at multiple scales from neighbourhoods of ten cells to large environments and identify differences in cellular organization and immune co-localization along the colon. Further, they profiled the single cell gene regulation and expression of these regions of the colon by integrating snRNAseq and snATACseq from the same tissue samples and revealed more detailed descriptions of cellular content. By combining spatial co-localization measurements and region specific single cell sequencing, they nominate a list of receptor-ligand interactions that are involved in different cell neighbourhoods and present SEMA4D-MET as an example of plasma cell – TA2 signaling in the colon. Next, through integration of gene expression and accessible DNA binding motifs, active transcription factors were identified in different cell types and these findings expanded to focus on groups of genes with similar expression changes across cell types and differentiation pseudotime. Transcription factor motifs enriched in similarly expressed targets were used to identify putative regulators of differentiation and function. Finally, they test if disease heritability associated genes (SNP-associated) are enriched in specific cell types linking enteroendocrine cells with BMI and T cells with inflammatory diseases.

Overall, this is a detailed example of the power of a systems biology approach to map tissues which includes both spatial, transcriptome and chromatin accessibility. There is excellent usage and integration of snRNAseq and snATACseq but this has only minorly been connected to the tissue

neighbourhoods and communities identified using CODEX imaging. It is a challenge to communicate such a detailed and broad study, but specific areas of clarification and presentation will help make this work more accessible to a general audience. In addition, methodological details and methods as well as the importance of making these datasets accessible as the offers have set up through the HubMap consortium are important aspects of its dissemination as, alone, the dataset is limited to a small number of patients.

Comments:

Multi-layered cell neighborhood and community analysis

1. How are cutoffs for window sizes or differences between communities determined? This would be important for determining the robustness of multi-cellular feature metrics and to enable the quantification of such features across different datasets or tissues. How is the decision made on where to cut a cluster of clusters?

a) Does altering the window size change the definition of a multi-cellular feature? Are different sized tissue structures more accurately identified at different scales? A neighborhood of 10 or 100 closest cells appears arbitrary (The answer was also not apparent in Schürch et al. 2020).

b) How do less or more prevalent cell types influence multi-cellular environment clustering? This would also appear to be dependent on the granularity of the underlying single cell clustering.

2. a) How does tissue orientation and cut impact cell or neighborhood quantification in the CODEX data? Various images are taken from different planes in the tissue resulting in images with different portions of epithelial tissue in an image or different cells across a crypt. Does this impact the proportion of cells or communities quantified in an image or the community that each cell is in? Some plots could be switched to cellular density measurements cells/area instead of as a percentage of cells.

b) There are also areas of the tissue which have been damaged during processing, how are these artifacts corrected for in spatial analysis.

Visualization and interpretability of tissue maps across scales

The hierarchical maps of tissue organization are of great interest but their current presentation is very difficult to interpret. A reorganization or new approach to these plots could provide much more information to a reader. Inclusion of labels and not just colors would be of great help. It would also be useful to be able to easily compare them across different settings. I understand that this is challenging data to present in this format but as a key finding of the paper an interpretable version is needed.

Integration and validation of single cell imaging and sequencing results

1. There is excellent integration of the snRNAseq and snATACseq results but the multiplexed imaging data and sequencing are only minorly linked through the identification of interacting cells and their potential receptor-ligand pairs. Unfortunately, this analysis seems underdeveloped as it is limited to only communities that differ between bowel and colon leaving behind an opportunity to model the molecular interactions present in specific neighborhoods. This information is only listed in a small table and should be presented in the main body as it is more insightful than the numerous UMAP plots of expected cell types. As has been completed in one example, it would be fantastic if the authors were able to plot potential signaling networks in different neighborhoods or even identify

cell signaling factors or responses in the single nuc data that is associated with a spatial location or interaction.

2. With such detailed phenotyping completed in the snRNAseq and snATACseq a missing link in the paper is the validation of these results using CODEX and the localization of the identified phenotypes and signaling events to specific tissue locations. Why are mast, DCs, and neutrophils not in both datasets?

Detailed comparison of communities or communities in different regions

Very interesting work detailing the role of different transcription factors through differentiation is nicely presented. Is it feasible to profile transcriptional changes that are associated with different neighborhoods or communities? Are there changes in cell function that are associated with the identified changes in tissue organization? As it stands, the association between the neighborhoods of figures 1-3 and the single cell data of figures 4-6 remain independent biological entities.

Author Rebuttals to Initial Comments:

Referee #1 (Remarks to the Author):

Proposed gap: Discussed in lines 21-24 : “Moreover, these cell types are spatially organized into different “neighborhoods” across these intestinal regions, with both the composition of these neighborhoods and molecular phenotypes of underlying cellular types varying in relatively unknown ways across these anatomical regions.”

Novelty: Combine RNA measurements with chromatin measurements and protein measurements in space to characterize cellular and molecular distribution of circuits along the human gut axis

General positive comments:

- The datasets cover multiple data modalities (spatial, transcriptomic, chromatin accessibility), sample many regions of the intestine, and come from several individuals.
- Analysis of the individual datasets uncovers cell types, cell states, cell neighborhoods, differentiation trajectories, and gene regulatory networks.
- The CODEX antibody marker panel was greatly expanded to detect major gut epithelial cell subtypes.
- The integration of the scRNA-Seq and scATAC-Seq is carefully performed.

The cohort chosen for the study is small covering only 4 individuals. Furthermore, the cohort is not balanced well for age, sex, and race. The CODEX data feel disconnected from the RNA-seq and ATAC-seq data presented later in the paper and the two datasets do not build well off one another. Very little novelty is presented in the first 4 figures where the conclusions drawn are underwhelming. The statistically significant observations made in these first 4 figures are almost all already known/reported. The data presented in figures 5 and 6 are intriguing and figure 5 can be extended to include TF analysis of immune and stromal subsets in addition to the reported data on epithelial cells. The paper would also benefit from better utilizing the different regions of the intestine collected when presenting the analysis of the CODEX data. Focussing the majority of analysis on SI as a whole vs Colon as a whole is overly simplistic and sells the data short. The authors should make a better effort to tie the observations made via the spatial analysis using CODEX with the second part of the paper using RNA-seq and ATAC-seq to dive deeper into the molecular networks across the length of the intestine.

We thank the reviewer for enthusiasm and feedback on the manuscript. We increased the number of samples by completing CODEX (2.7 million cells) and multiome measurements across 8 sites of the intestine for 5 additional new healthy donors. In total, we now analyze eight sections from nine individuals: seven European-ancestry (five males and two females), one African American male, and one African American female. Age ranges were from 24 to 78 years. Furthermore we addressed integration of the CODEX and snRNA/ATAC-seq datasets by validating original snRNA/ATAC-seq findings with CODEX data, expanding the receptor ligand analysis to neighborhoods, and integrating snRNA/ATAC-seq cells with the CODEX multiplexed imaging using a newly developed technique. We expanded the TF analysis in Figure 5 to the

immune and stromal cell subsets, nominating potential regulatory TFs in those intestine cell types. We also further investigated differences within different areas within the intestine. As a result of responding to reviewer comments we generated many new findings and moved survey analysis from the first 4 figures to the supplementary information.

Major comments:

1. Cell proportion data from CODEX are difficult to interpret given the heterogeneity in tissue sectioning between intestine regions within an individual as well as across individuals (shown in Figs S2 and S6). The only clear signals seem to be expected ones such as more CD8 T cells in the small intestine and more enterocytes in the small intestine given that the small intestine has villi and the colon does not.

We agree that with only 3 donors, variation in tissue sectioning could influence interpretation of cell type abundance. To address this we collected CODEX multiplexed imaging data from an additional 5 donors from the 8 sites of the intestine with a total of 2.7 million cells and also modified our tissue harvesting protocol to get a higher success rate of tissue section alignment.

With the additional 5 donors (8 donors total), the data now show additional differences between the small bowel and the colon, while substantiating previous cell type differences across the small intestine and colon. One example is the novel discovery that there are differences between the percentages of dendritic cells in the small bowel and colon, and we highlight this in the manuscript.

Supplemental Fig. 7C

While not all of our tissue sections were not perfectly oriented, the majority of the initial 3 donors (21/24) are good cross sections from the harvested tissue region. We now reference this within the text of the manuscript as a source of variation. Also, some of these tissues may be good cross-sections, but may not contain all layers of the intestine because we attempted to maximize the epithelial area within the imaging window since this is where the bulk of our antibodies targeted cells existed. In addition, the intestine has invaginations that push epithelial areas out such as regions below where there is no muscularis externa.

From Supplementary Fig. 5A

For this reason, we normalized cell type percentage to stromal, immune, and epithelial compartments. To further segment distinct areas of the intestine as opposed to cell type compartments, we have now defined “tissue units” of the intestine. This allows us to characterize specific frequencies across the mucosa, muscularis mucosa, submucosa, and muscularis externa. We observe similar changes in cell type frequency that we would expect to change across the different tissue compartments. For example we see the same significant changes of epithelial and immune cell subsets within the mucosal areas (Enterocyte, Paneth, CD57+ Enterocyte, CD66+ Enterocyte, CD8+ T, DC, etc.) and also within the muscularis externa (smooth muscle, Endothelial, etc.). Here we show one of the graphs from this analysis of the mucosal areas with the percentages of the epithelial cell types plotted as a percentage of all cell types within this region for each donor, comparing the SB and CL.

Supplementary Fig. 8B

We have improved our collection of the new donor samples such that all but one were good cross sections of the intestine. One reason we could not set up a perfect cross section across all areas of the intestine for every single tissue was due to how we prepared the tissue arrays to minimize batch effects and increase efficiency for CODEX for our first 3 donors. This enables us to do more samples with less artifacts in the data. In our previous setup protocol we separately froze each tissue piece in OCT, and then combined these four tissue pieces within a single OCT block as described in Fig. 1A of the manuscript. Since we embedded each piece of tissue in OCT at the time of collection, sometimes the small piece of tissue would get distorted and so when the array was created it did not align with the other four tissues. We have improved our collection of the 5 new donor tissues by freezing much larger sections of the colon to minimize distortion. We also directly cut these large tissues and placed large tissues into arrays with known directionality. As documented by the picture below, we have now added this information to the Methods section. In total (60/64) sections are appropriately cross-sectioned.

We have also now registered these blocks within HuBMAP's tissue registration in a common coordinate framework [pmid 34750582, doi 10.1038/s41556-021-00788-6] and have added this to the Methods section. Briefly, male and female 3D reference objects for 11 organs including the small bowel and colon were created using Visible Human Project datasets. Using standard surgical anatomical landmarks used to collect the eight bowel sites, the tissue blocks were registered to the reference objects. The anatomical landmarks used for small bowel segments were: 1) descending duodenum to the right of the pancreas head, 2) 5 cm beyond the ligament of Treitz in the jejunum, 3) 200 cm of the jejunum beyond the ligament of Treitz in the mid-bowel, and 4) 5 cm proximal to the ileocecal valve for terminal ileum. 5 cm of bowel at each site will be collected, representing approximately 20 gm of tissue at each site. For colon, the landmarks used were: 1) right colon midway between the ileocecal valve and the hepatic flexure; 2) transverse colon midway between the hepatic and splenic flexures; 3) left colon midway between the splenic flexure to the appearance of the sigmoid mesentery; and 4) sigmoid colon midway to the rectosigmoid junction where the taenia coli ceased.

2. Tissue sections presented for CODEX data are not comparable in their orientation and inclusion of all layers of the intestine (shown in Figs S2 and S6) within the small or large intestine for one individual as well as across the three individuals presented. How did the authors choose which sections of duodenum, jejunum, ileum to process from the organ donors? Can the authors place these gut sections in any of the common coordinate frameworks provided atlas consortia (e.g. HubMAP, HCA, ...)?

Thank you for this comment. For heterogeneity of the tissue sections and the common coordinate framework, see the answer to Reviewer #1 question #1.

3. Figure 1 and 2 should be combined with most of figure 1 data moved to the supplement as the observations in figure 1 are not so easily interpretable for reasons above nor the conclusions novel outside of perhaps figure 1F.

We have moved many of the findings from Figure 1 to supplemental material and highlight new findings within new Figure 1E, G, H, I, J, K, L panels and Fig. 2F, G, H, I, K, M that we think add novel conclusions based on the associations with clinical metadata, density of cell types, organization of macrophage subsets, unique neuroendocrine neighborhoods of the colon, density analysis across neighborhoods for neuroendocrine cells, and paneth cell neighborhood anchor analysis. This is in addition to the novel results of detailed cell type composition across the intestine and first description of multicellular neighborhoods found across the intestine in Figures 1 and 2. These new analyses and results are discussed in detail within the text, but we share here briefly:

- We observe an increase within DC percentages going from the small intestine to the colon as shown in Supplementary Fig. 7D:

- We detail a correlation between M1 macrophages and BMI across all areas of the intestine Fig. 1G, H:

- Cell density analysis of cell types revealed a high density of plasma cells that we found to be an important cellular neighborhood (discussed within Fig. 2/3) as shown in Fig. 1 J, K:

- We further observed differences in M1 versus M2 macrophage density and localization where M1 macrophages are primarily restricted to the *Mucosa* tissue unit as shown in Fig. 1 L:

- We observe decreases in cell type frequency for CD8+ T cells and endothelial cells in the intestine samples from donors who have had a history of hypertension as shown in Fig. 1 I:

- This difference in cell type percentage was further revealed by multicellular neighborhood analysis to be mostly changes to CD8+ T cells in *CD8+ T cell IEL* neighborhoods and endothelial in *Microvasculature* neighborhoods as opposed to other neighborhoods (e.g., *Macrovasculature*) as shown in Supplementary Fig. 11G:

- We observe a unique neuroendocrine enriched neighborhood within areas of the colon but not the small intestine when performing neighborhood analysis at an individual scale as shown in Fig. 2G:

- Neuroendocrine density was shown to be increased in the colon as compared to the small intestine in Fig. 2H:

- We also observe that neuroendocrine density decreases moving up the epithelial neighborhoods of the intestine, with the highest density in the newly identified *Neuroendocrine Enriched* neighborhood and this was shown to be at the bottom of the colonic crypt as shown in Fig. 2 I, J:

- The *Neuroendocrine Enriched* neighborhood composition was similar to the crypt region of the small intestine where it had an enrichment for similar cell types of the adaptive immune system. This adaptive immune enriched zone is shown by a concentric neighborhood analysis around paneth cells as shown in Fig. 2 K-N:

4. For follicle based CODEX observations was a follicle present in each section of each of the 8 segments for an individual for all 3 individuals? It is unclear how many follicles are being studied for these analyses.

Not all sections have a follicle. In fact a majority of sections do not have follicles. We used our *Community* definition of follicles to be able to separate and segment out follicle areas of the tissue and identified 11/64 sections have mature follicles. These occur both in the small intestine (3) and the colon (8). For this reason we have limited discussion about the composition of the *Follicle* community (*Inner Follicle+Outer Follicle*) within main figures, and focus on analysis of the *Outer Follicle* that is found throughout samples irrespective of a fully mature follicle found. We also highlight that the variation of these segmented follicles could likely reflect different stages or the follicle based on greater B cell percentages reflecting dynamic processes as shown in Supplemental Fig. 12 A, B.

5. In figures 1-3 where CODEX data is presented the authors do not seem to leverage the different regions of the small intestine and colon in the analysis presented. Comparisons are made at the level of all SI and all Colon but it is less clear which segments within those tissues are driving the differences. Is this limitation due to the heterogeneity in tissue sections for the various sections taken making it difficult to find signal allowing to make conclusions about the duodenum vs jejunum or sigmoid vs transverse colon for example?

We thank the reviewer for this suggestion. We now compute statistics comparing the different regions. Some interesting differences arise when we compare these cell types across the different regions. For example, CD57+ Enterocytes, a rare epithelial cell type, are nearly restricted to the Duodenum as seen in Supplemental Fig. 8C below:

We now highlight this within the text as well and discuss that it is a part of the gland structures that MUC6 cells are found within the single-cell RNA sequencing data.

Another comparison that we made across the different tissue regions was that of the Ileum and its differential of immune cell neighborhoods with an increase in adaptive immune enriched cell neighborhoods and a decrease in plasma cell enriched neighborhoods. This is now presented in Supplemental Fig. 11 C-F.

Many changes are found broadly at the comparison level of the SB to CL even when comparing multiple regions such as CD8+ T cells. This is now shown in Supplemental Fig. 7E.

6. For the categorization of the CODEX data into neighborhoods, communities, and major communities how do the authors control for the limitation of a 7-micron cross section not being able to accurately represent the true spatial distribution/orientation of cells to one another since it is highly unlikely for any given cut to be in the perfect orientation such that one can be confident that

any two cells or neighborhoods or communities are truly adjacent in space within the tissue? Controlling for transverse vs cross-sectional cuts and any angle in the cut that is not in the same plane along the length of the tissue axis being cut. This can play a role in conclusions drawn such as in Figure 3A where the suggestion is the immune major communities occupy different spaces in the small vs large intestine.

CODEX multiplexed imaging is a platform that most often uses a 2D imaging approach (not confocal). Thus, having a 7-micron section is a conventional size slice for characterizing cell type composition in a 2D plane. And for consistent embedding and optimization for cutting cross-sectional cuts of the intestine see the answer to Reviewer #1 question #1. In short we have verified that the majority (60/64) of our sections are cross-sections of the intestine.

Furthermore, regardless of the orientation, the neighbors of the cells as detected by CODEX in 2D are in adjacent space within the tissue. Our neighborhood analysis takes the nearest neighbors of a single cell within a 2D slice and these are used as vectors and are clustered to determine conserved structures. Consequently the neighborhoods defined will incorporate these neighbors into the calculation and identification of conserved compositions. Indeed, the neighborhoods we first found from our analysis of just 3 donors are still evident when expanded to all 8 donors providing confidence that neighborhoods we detected are conserved structures.

Based on suggestions we have switched from including major communities which were just aggregations of communities into major structures in the intestine. In contrast we cluster now in an unsupervised fashion to detect major tissue units of the intestine: muscularis externa, muscularis mucosa, submucosa, and mucosa at an even higher level that align with routinely pathologically identified areas of the intestine. These are now described in Figure 3D and Supplemental Figure 15:

Moreover, we performed neighborhood analysis on the 8 individual regions of the intestine in addition to just characterizing neighborhoods separately (See response to Reviewer #1 question #40). The majority of these neighborhoods and their compositions are conserved across regions solidifying our confidence in our identified neighborhoods and communities.

7. The authors jump from Fig 4 to Fig 5 prior to completing discussion of data in Fig 4. Furthermore, there is no clear logic as to why Fig 5 focuses on enteroendocrine cells in A-C and then goes on to do a broad TF analysis which is the title and focus of the figure.

We have consolidated Figure 4 and Figure 5A-C into a single main text figure and now focus only on the TF analysis in Figure 5.

8. Why do the authors limit the regulatory TF analysis in Fig 4 to only epithelial cells? Can this not be applied to immune and stromal cell populations along the length of the intestine? This would make a more comprehensive figure and can substitute for the inclusion of the enteroendocrine analysis in A-C.

We now complete a similar analysis to identify regulatory transcription factors in the immune and stromal cell populations. We have added figures from this additional analysis to the supplement and no longer include the enteroendocrine analysis in figure 5A-C. We have added the following text to the manuscript and the new figure panels are reproduced below.

“To determine which TFs may drive cell function in stromal and immune cells in the intestine, we performed the same analysis for cells in each of these compartments. Within the immune compartment, this highlighted many TFs known to be important for cell type differentiation and maintenance including GATA2 in Mast cells and PAX5 in B cells (Figure SXa). Among stromal cells, we identified TFs associated within specific lineages, including EBF1 in pericytes, which was recently suggested to contribute to pericyte cell commitment, PPARG in adipocytes, and Sox10 in glia (Figure SXb). We also nominate potential regulatory TFs in ICCs, including HAND1, HOXD11, and MEIS1 as well as potential regulatory TFs for different classes of intestinal fibroblasts.”

Supplemental Figure 27: Regulatory transcription factors in the immune and stromal cells of the human intestine.

a, b) Heatmap representation of transcription factors whose gene expression was correlated with their motif activity in immune cells (a) and stromal cells (b) in at least one region of the intestine. Row z-scores of ChromVar deviation scores are shown on the left and row z-scores of TF expression are shown on the right.

9. The gut data collection is excellent, but the data sets are currently best suited as a reference for future studies of the gut. The authors should give examples of how this atlas can contextualize other existing or future single cell and spatial studies of healthy and disease gut. This can be at the level of genetics (e.g. GWAS genes for IBD, CD), or comparisons to other disease gut datasets (cell types, compositions etc...)

We now give examples of how this atlas can be used to contextualize future single cell and genetic studies of the healthy and diseased gut, and reference an example where colon polyp data was projected onto a normal colon reference to identify features of tumorigenesis. We have added the following to the discussion:

“Together, this data provides a detailed atlas that can serve as a valuable reference for future studies of the intestine. For example, polyp and cancer data can be compared to the normal reference derived here to identify features of tumorigenesis and malignant transformation⁷⁷. Additionally, this data also provides a reference to contextualize future GWAS studies of intestinal disease, similar to the analyses performed above.”

10. The authors should contextualize their findings with other existing gut atlases, for example: A single-cell survey of the small intestinal epithelium (Nature, 2017), Cells of the human intestinal tract mapped across space and time (Nature, 2021), A proximal-to-distal survey of healthy adult human small intestine and colon epithelium by single-cell transcriptomics (bioRxiv, 2021).

We have numerous additional callouts to these papers throughout the text. For example, we map the annotations of the fibroblast subtypes we define to the labels in previous work. We also note previous observations about BEST4+ enterocytes throughout the intestine in the text. We have also added a paragraph to the discussion explicitly contextualizing this work with the previous gut atlas papers:

“This work follows and extends the work of several scRNA gut atlases, including an atlas of the small intestine in mice (Nature 2017) and two recent atlases describing the small and large intestine in humans (Nature 2021, Cellular and Molecular Gastroenterology and Hepatology 2022). The first of these human studies additionally provides information on intestinal development in addition to mapping out cell types and features of the adult human intestine. Our work supports many of the findings in these previously published atlases. For example, the two recent human gut atlases both identified Best4+ enterocytes, first identified in the colon (Smillie et al 2019), in human small intestine and large intestine, which we observe in our data as well. While these previous scRNA datasets have examined the diversity of cell types in the human intestine, our integrated snRNA and snATAC dataset provides a detailed single cell regulatory map of the intestine. With the inclusion of two donors from underserved

backgrounds, our work also extends single-cell analyses to include samples from such groups.”

11. Within each atlas (spatial, transcriptomic, chromatin accessibility), most of the biological findings into gut biology are already known. The findings that different gut regions have different compositions and organizations is expected, and none of the new findings are experimentally validated.

We have expanded the analysis and results to include several new findings as discussed in response to Reviewer #1 question #3, 5, 14, & 40 and Reviewer #2 question #1. We also now include experimental validation of several findings including the presence of MUC6+ cells in the duodenum (Figure 4k and 4l) and receptor ligand interactions we previously observed, see Reviewer #1 question #12 & 13. We have also expanded the TF analysis to further support the regulatory epithelial TFs that we identified previously and now expand the regulatory TF analysis to nominate regulatory TFs from immune and stromal cells in the intestine.

12. Validate by IHC that there are MUC5B+ and MUC6+ secretory cells specific to the duodenum.

We have validated the presence of populations of MUC6+ secretory cells in the duodenum with CODEX imaging as well as with molecular cartography. We now include figure panels for these validation experiments in Figure 4k and 4l. These figure panels are reproduced below:

13. Validate SEMA4D expression in plasma cells and MET expression in TA2 cells in colon vs small intestine.

We used Molecular Cartography provided by Resolve Biosciences to validate the higher expression of receptor and ligand pairs in colon as predicted in Supplemental Table 3 (Method). We performed this analysis in a subset (n=6) of samples with 37 probes to describe cell types which enabled detection of 20 cell types that are present in about half (58 of 152) of our predicted pairs. 63 probes were used for the receptor and ligand expression of those that were predicted from our analysis. For the SEMA4D and MET receptor interaction pair we find that TA cells positive for MET and plasma cells positive for SEMA4D were colocalized (CLQ=1.57). In analysis of all receptor-ligand pairs we can confirm 9 pairs (out of 58, p value < 0.002, permutation test) have higher expression in colon than small bowel. An example is shown below (Supplementary Figure 25b) for ligand FN1 and its receptor PLAUR. Their expression in myofibroblast and enterocytes, respectively, are upregulated in the colon compared to the small bowel, consistent with our prediction. Both genes are indicators for inflammation. We have included this validation in the main text and supplementary data.

14. What gut cell types are the Insulin-like peptide 5 L cells signaling to, and is the cognate receptor expressed?

The cognate receptor for INSL5 is RXFP4 (see <https://www.nature.com/articles/srep29648> and <https://pubmed.ncbi.nlm.nih.gov/18236022/>), which we found was primarily expressed in enterochromaffin cells in the colon and I cells in the colon. This is consistent with INSL5 being primarily expressed by L cells in the colon, but not other regions of the intestine. We have added the following figure showing which cell types express the INSL5 receptor, which are likely the cells the INSL5+ L cells are signaling to figure 4.

We describe this figure in the manuscript as follows:

“L cells can be further divided based on expression of INSL5+, which is primarily expressed by L cells in the colon. To determine which gut cell types the INSL5+ L cells may be signaling, we examined the expression of RXFP4, the cognate receptor for INSL5. We found that RXFP4 is primarily expressed by colon enterochromaffin cells and, to a

lesser extent, I cells, suggesting that these are the most likely cell types that the INSL5+ L cells are signaling (Figure 4g).”

15. Validate potential master regulator TFs (Fig 5D): RUNX1 and RUNX2 in Tuft cells, FOXA3 and ATOH1 in goblet cells, and ZBTB18 in Paneth cells

We have approached validating these findings in multiple ways. First, we further examined the literature to highlight any evidence for these as master regulatory TFs. For a few of the TFs mentioned there is data in similar cell types in other tissues or in other species. For example, FOXA3 leads to goblet cell metaplasia in lung (<https://doi.org/10.1164%2Frccm.201306-1181OC>), and here we show that in addition to being a potential master regulator TF in the lung, it may also be in the intestine. For ATOH1, there is substantial data that ATOH1 is necessary for epithelial differentiation to the secretory lineage in mouse intestine (See Yang Q, Bermingham NA, Finegold MJ, Zoghbi HY. Requirement of Math1 for secretory cell lineage commitment in the mouse intestine. *Science*. 2001;294:2155–8. and Vandussen KL, Samuelson LC. Mouse atonal homolog 1 directs intestinal progenitors to secretory cell rather than absorptive cell fate. *Dev Biol*. 2010;346:215–23.)

Second, we now collect and include single-cell data from over twice as many samples to further validate our findings. This includes multiome data from 6 donors and a cohort of separate snRNA and snATAC data from 3 donors. When we complete the same analysis looking for regulatory TFs from these two groups of data, we find that the majority of the regulatory TFs identified in the multiome analysis are also found in the separate snATAC and snRNA analysis (48 of 63 TFs), including four of the five TFs listed above (ATOH1, FOXA3, RUNX1, and RUNX2). The overlap in regulatory TFs is summarized in the figure below.

Finally, we provide additional bioinformatic evidence that these are regulatory TFs in their respective cell types. For example, we now examine transcription factor footprints for these TFs in different cell types and find that there is greater accessibility flanking ATOH1 motifs in Goblet cells when compared to enterocytes.

We have added the above figures to the manuscript and summarized the findings in the text as follows:

“ATOH1 has been shown to be necessary for secretory lineage commitment in mouse intestine^{58,59}, and here we provide data supporting a similar function in human goblet cells. To provide further evidence that ATOH1 is binding its motif and driving accessibility in goblet cells, we compared ATOH1 footprints in Goblet cells and Enterocytes (Figure 5c). Indeed, we found greater flanking accessibility around ATOH1 motifs in Goblet cells, consistent with greater ATOH1 binding in goblet cells. For FOXA3, there is evidence that FOXA3 leads to goblet cell metaplasia in lung⁶⁰, and our findings support that it is likely also an important regulatory TF in colon”

“To help validate these findings, we completed this analysis in two separate cohorts. In the multiome samples as described above and in the remaining samples that we collected separate snRNA and snATAC data. To examine TF expression in the latter group of samples, we integrated the snRNA and snATAC data using canonical correlation analysis to align the datasets and assign snRNA data to each snATAC cell (Methods)^{54,55}. This analysis reproduced many of the findings in the multiome analysis, with 48 of the 61 TFs originally identified in the multiome analysis reaching the same significance criteria (Figure 5d).”

16. Are there further ways in which the single cell transcriptomic and spatial data can be integrated, beyond the ligand-receptor interaction analyses?

We have expanded the cell-cell interaction analysis with receptor ligand pairs - See Reviewer #1 question 45. We have also integrated single cell data with CODEX multiplexed imaging data

using a newly designed algorithm - See Reviewer #4 question #3C. We also now use CODEX to validate some of our findings in the initial single-cell datasets, including the presence of MUC6 in the intestine - See Reviewer #1 question #12.

17. The spatial data collected here is unique, and one valuable contribution would be a power analysis of the data. How does the detection of cell types and cell neighborhoods depend on the number of individuals and tissue sizes / FOVs samples? This can be done empirically through downsampling of the data.

Detection of both cell type and neighborhood differences across the small intestine and colon were largely consistent between our original 3 donors and 8 donors. We now performed power analysis for the cell type and neighborhood analysis with our dataset of 8 donors to calculate sample size with power=0.8 and alpha=0.05 as suggested by the reviewer. This matched and is now shown in Supplementary Fig. 8F and 12F:

Minor comments:

18. The abstract focuses only on the colon directly (the authors make no direct mention of the small intestine) although the paper presents data on both the small intestine and colon. The abstract should be amended to reflect this.

We have changed references to colon to references to the intestine in the abstract.

• Fig 1

19. It is unclear which 3 of the 4 donors were used to generate the CODEX data

All donors were characterized by CODEX but B001 was stained with the base panel that did not include the gut-specific markers. We now explicitly mention B001 is the patient not included in the aggregate analysis.

20. Data in 1F should be split into the three major cell types like in 1D. Immune cells are mobile and less geographically restricted in tissues whereas enterocytes are geographically restricted to form the architecture of the organ (ie crypts and villi etc). This makes comparing different cell lineages on the same graph less meaningful.

We thank the reviewer for this suggestion and now plot these in 3 separate graphs in Figure 1J and Supplementary Fig. 9A, B (*not shown here*).

21. The authors use their geometric deep learning method STELLAR (Annotation of Spatially Resolved Single-cell Data with STELLAR (bioRxiv, 2021)) to annotate several of the spatial datasets. More detail is needed for the reader to understand how well this works. In particular, how was the model trained and tested (which individuals and which tissue regions), and did the model discover any new cell types or classify any as unknown? This is of particular importance given how different the tissue samples appear across the 4 patients (Supp Fig 6).

We manually annotated small intestine and colon data from one donor from B004. This was a nontrivial task that we did not elaborate on. To create an accurate cell typing for this CODEX dataset, we spent over 60 hours of work to cluster, merge, recluster, subcluster, and assign cell types based on average marker expression. Each cluster's purity and accuracy was confirmed by location of the cell within CODEX images with corresponding fluorescent images and also H&E staining. In particular this dataset was especially curated as we compared a variety of

normalization techniques and unsupervised clustering algorithms on the accuracy of the clustering (<https://doi.org/10.3389/fimmu.2021.727626>).

After curating the training dataset and validating its accuracy, we then applied STELLAR using this annotated small intestine and colon data as a labeled training set. We then used small intestine and colon data from other donors as an unlabeled test dataset. For each donor, we trained a separate model. The model did not classify any cell types as unknown which agreed well with our broad cell type characterizations that spanned the breadth of our markers. We would not expect to find a new cell types across donors within this CODEX dataset since we trained with both small intestine and colon data. We additionally confirmed the quality of STELLAR's predictions by looking at average marker expression profiles of predicted cell types and found that protein marker distributions match expert hand-annotated profiles (Fig 3d in co-submitted manuscript). In response to the Referee's feedback, we have included a paragraph with more details in the Methods section.

- Fig 2

22. Page 8 line 13 states the plasma cell enriched neighborhood has co-enrichment with CD4 T cells and APCs. This observation would be strengthened by a visual plot of these cell types in space to appreciate this enrichment as was done for same cell enrichment in Figure 1E.

We thank the reviewer for this suggestion. We now plot these cell types in space to appreciate this enrichment within the plasma cell neighborhoods in Supplemental Fig. 10:

23. Page 8 line 31 should call out Figure 2D not Figure 2C

We have now corrected this error.

24. The heatmap presented in Fig 2F is confusing. It is unclear how the authors defined cell composition conservation. Given the authors mostly discuss the differences in cell neighborhoods between SI and CL, it may be simpler to present just the neighbor composition conservation, or at least keep it in a separate figure from the cell composition conservation.

We thank the reviewer for this suggestion. We have moved this plot to supplemental information and clarified in the text that we define neighborhood conservation as a comparison of the cell type enrichment score for each neighborhood when comparing the small and large intestine. Thus, it is necessary to present this figure as a heatmap to show the cell type composition of each neighborhood. While it is also useful to see which cell types vary the most across all neighborhoods (cell type conservation) we have stressed this fact within the manuscript to decrease confusion.

• Fig 3

25. In Fig 3B, the tSNE embeddings are different between neighborhood and major community vs community. All three tSNE embeddings should be the same as they are built from the neighborhood vectors. Without this, we cannot see how well cell neighborhood vectors cluster into the communities.

We thank the reviewer for catching this, the tSNE embeddings were different, but we have removed these as it is not quantitative nor central to the updated manuscript.

o Fig 3D

26. Labeled as epithelial spatial context map, but I think the authors included both the epithelial and immune major community in this analysis.

This is correct. Now we present two spatial context maps and clarify with the titles what type of cells are used to create them. One uses all the communities from the colon and is labeled as “CL Community Spatial Context Map.” And the other uses the neighborhoods from just the mucosa Tissue Unit and is labeled “CL Mucosa Neighborhood Spatial Context Map”

27. The box colors make it confusing as to what are corresponding regions. I think the green and black boxes (left) and the orange and blue boxes (right) refer to the same regions respectively.

We thank the reviewer for the suggestion and now use consistent colorings for the regions that we want to highlight.

28. Please add a legend for the cell neighborhood colors.

We have rearranged the figure to make it more clear what colors represent across cell neighborhood figures.

29. Although the authors have a publication on tissue schematics and spatial context maps, they should describe the method here as it is novel and not trivial.

We have described these spatial context maps in much greater detail within the Methods section of the manuscript, added additional commentary within the manuscript while describing results, and also highlighted areas of the figure to direct attention.

• Fig 4

30. What are the markers for “Epithelial” in 4F when all cells here are presumably epithelial cells

While repeating the analysis with additional data we were able to remove this annotation and now use more specific classifications for all cell types.

31. Page 16 line 24 states one of the donors T cells cluster apart from the others. What figure shows this? Explanation is age may be the explanation. Did authors observe similar outlier behavior for this individual in CODEX data for T cells?

This was shown in Figure 4D, but the donor identity was not explicitly shown. We now include a supplemental figure showing that this cluster is composed of cells entirely from a single donor, which is reproduced below:

In the CODEX data, we did not observe T cells to cluster independently, likely because we only had major cell type markers for T cells (CD45, CD3, CD8, CD4) rather than phenotypic features that may differentiate these cell types.

32. Authors identify a duodenum specific subset of Muc6 and Tff2 expressing cells. These two genes are highly expressed in the stomach (see GTEx data set). Is it possible duodenum sample preps have contaminating cells from the stomach?

We now take multiple steps to validate the presence of these cells in the duodenum. First, we now generate over twice as much data, and we observe the presence of these MUC6+, TFF2+ cells consistently in our single-cell experiments. We also now validated the presence of these cells in the duodenum with CODEX imaging, and observed that they are consistently present in the duodenum of multiple donors, as shown below:

33. The authors also note the MUC5B+ secretory cells in the duodenum are surprising. The authors should consider validating the presence of this population. Interestingly, the human protein atlas does not report MUC5B expression in the duodenum

(<https://www.proteinatlas.org/ENSG00000117983-MUC5B/tissue>)

After performing additional single-cell sequencing experiments, we did not find additional duodenum samples with populations expressing high levels of MUC5B+ cells, making us less confident that this is a cell type commonly present in the duodenum. It is possible that this is a contaminating cell type from a nearby tissue that was carried over from the dissection or it may be only found in one patient. Either way we have added the following to make it clear that we do not have evidence that it is a general cell type:

“Notably the MUC5B+ cells and exocrine cells were primarily only present in one sample in our dataset, making it possible that they are contaminating cells from a different tissue, where expression of these markers is more common, or a rare cell type in the duodenum.”

• Fig 6

34. Fig 6F heatmap needs four labels for the duodenum, jejunum, ileum, and colon regions.

We have added these labels to figure 6F.

35. Please show how cell type composition and neighborhoods within a spatial tissue region vary across individuals. This is not shown in Supp Fig 7,8 but Supp Fig 7C comes close.

We added examples of both cell types and neighborhoods percentages with donor and region information shown (see Reviewer #1 question #5 for example plots).

36. It is unclear how the authors tested for statistically significant differences in cell type composition across samples, and this is important for many of their claims. The test should control for the compositional nature of the data (good references include Differential abundance testing on single-cell data using k-nearest neighbor graphs (Nature Biotechnology, 2021), scCODA is a Bayesian model for compositional single-cell data analysis (Nature Communications, 2021).

We now apply both of the suggested methods, scCODA and Milo, to identify significant differences in cell type composition across locations of the intestine. We added a milo beeswarm plot showing significantly differentially abundant neighborhoods when comparing the small intestine to the colon in the main figures and have added comparisons between the individual small intestine regions and the colon to a supplemental figure. These methods support the findings that CD8+ T cells are more prevalent in the small intestine, whereas macrophages/monocytes are more abundant in the colon and that smooth muscle/myofibroblasts were less abundant in the duodenum and jejunum than the colon.

h) Beeswarm plot showing the log-fold change between small intestine and colon for groups of nearest neighbor cells from different cell type clusters. Significant changes are indicated in color.

Supplemental Figure 19: Differential cell type abundance on a nearest neighbor graph.

a–c) Beeswarm plot showing the log-fold change between the three main regions of the small intestine and colon for groups of nearest neighbor cells from different cell type clusters in the stromal (a), immune (b), and epithelial (c) compartments. Significant changes are indicated in color.

Supplemental Figure 20: Differential cell type abundance with scCODA.

a) Boxplots comparing the fraction of all cells in each sample composed of each cell type for samples from the colon, ileum, jejunum, and duodenum. b) Log₂FC in abundance of each cell type between the regions listed on the y-axis as estimated with scCODA. Only significant results at an FDR of 0.05 are shown, with all nonsignificant differences plotted as white.

37. What are ITLN enterocytes?

We have decided to exclude ITLN1 from subcategorizing cell types. Though this marker is often used to categorize early secretory epithelial cells [1].

[1] Owen, R.P., White, M.J., Severson, D.T. et al. Single cell RNA-seq reveals profound transcriptional similarity between Barrett's oesophagus and oesophageal submucosal glands. *Nat Commun* 9, 4261 (2018). <https://doi.org/10.1038/s41467-018-06796-9>

38. Can the authors provide further detail on how they decided how many cell neighborhoods to define in their spatial data? A tSNE of the cell composition vectors for all the cells may make it clearer how distinctly single cells separate into the cell neighborhoods.

Deciding upon the number of cellular neighborhoods is an important parameter. In general, we tested out unsupervised clustering across a number of resolutions. By comparing cell type enrichments within resultant clusters and also by mapping these back to the tissue allowed us to determine unique cellular neighborhoods. Consequently, we learned that starting with 30 clusters enabled recognition of unique conserved structures across the intestine. We clarify this now in the methods that for the final dataset we started with 30 clusters and merged down to our final uniquely named neighborhoods.

39. Why is the most diffuse same cell distance defined as taking the number of total cells of a given cell type divided by the total area of the region. Intuitively I would think the most diffuse same cell distance should decrease as the total cells of a given cell type increase, and should increase as the total region area increases.

We agree this was a confusing title. This metric is found by taking the average distance divided by the most diffuse distance. We have clarified the name and explanation of this measurement in the figure, manuscript, and methods.

40. Why was the identification of the cell neighborhoods performed using all eight tissue regions together, rather than analyzing the regions individually? My concern is that a cell neighborhood unique to a tissue region would be missed in the joint analyses done here.

The identification of the cell neighborhoods is performed together with all the regions so that cell type microenvironments can be characterized and compared more easily across the heterogeneity when clustered together. We have kept the cell neighborhood analysis presented in Fig. 2 from this approach.

However, per the reviewer's suggestion we also took the time to annotate cellular neighborhoods for each individual region for all 8 regions. All 20 of the neighborhoods that we identified from analyzing all regions together were also identified by the neighborhood analysis of each individual region by itself as shown in Supplementary Fig. 13A:

In addition, we also identified 2 new neighborhoods: *Neutrophil Enriched* and *Neuroendocrine Enriched* neighborhoods. Both of these uniquely-identified neighborhoods were low percentages as the reviewer suggested and we have discussed the significance of both of these neighborhoods in detail in the text of the manuscript. Briefly, the *Neutrophil Enriched* neighborhood was driven by a high density of Neutrophils within the stroma. This was found in only a subset of the samples and primarily within the colon as shown by Supplementary fig 13C:

The *Neuroendocrine Enriched* neighborhood was solely found in the colon as shown in Supplemental Fig. 13B:

We analyzed the composition and localization of this neighborhood and found it represented the colonic epithelial neighborhood as described more in response to Reviewer #1 Question #3.

41. In Fig 1F, why is a same-cell density measurement reported for Paneth cells in the colon, given their expected absence? Is the annotation of Paneth cells in the colon a result of considering all gut tissue regions together rather than individually when identifying cell types?

We agree that this is distracting from the main idea this figure was trying to convey and now incorporate all regions per cell type.

42. The CD8+ T enriched IEL neighborhood in Fig 2B consists of CD8+ T cells and enterocytes, TA, and cycling TA cells? As TA cells in the small intestine tend to be at the base of the crypt and enterocytes tend to be in the villus, how are these epithelial cells all in the same cell neighborhood with the CD8+ T cells?

Annotating the cellular neighborhoods with all 8 donor samples together, there is not an enrichment of TA and cycling TA cells in the CD8+ T cell Enriched IEL neighborhood.

43. Why is the frequency of inner follicles not highest in the mid-jejunum / ileum (Fig 2D), as that is where we expect Peyer's patches? Do the CODEX measurements identify the Peyer's patches?

Imaging regions were selected in an unbiased manner, such that whatever block of tissue was cut and assembled into the tissue array was what was imaged with CODEX (See answer to Reviewer #1 question #1). The follicle regions that we identify through the neighborhood analysis directly correspond to follicle and Peyer Patch structures pathologically within the tissue. Here we identified 11/64 sections that had these mature structures (See answer to Reviewer #1 question #4).

Nevertheless our data show that the cellular neighborhoods of the *Outer Follicle* and *Adaptive Immune Enriched* neighborhoods are found independent of the existence of *Inner Follicle* neighborhoods suggesting that these neighborhoods are poised to develop into mature follicles. Indeed our data is a single-snapshot in time and likely there are number of follicular structures that may be in various points of development. Furthermore our comparisons of the immune cell neighborhoods in the ileum compared to other areas of the intestine showed an increase in *Adaptive Immune Enriched* neighborhoods and a decrease in *Plasma Cell Enriched* neighborhoods as described in answer to Reviewer #1 question #5.

44. Can the authors please comment on the limitations of the CODEX measurements with regards to cell type detection. What expected cell types are missing due to rarity or lack of appropriate antibody markers, e.g. ILCs, neurons, Tregs, Th1, Th2, M cells, ...?

There will always be tradeoff in multiplexed imaging panels since we only use a panel of 50 or so markers. We now highlight in the results the complementarity of the multiome data for detailed cell type analysis and highlight within the discussion how further studies can be done to change antibody panels instead of broadly characterizing the intestine by cell type for more specific studies involving more specific subtypes of cells.

45. "Using this intuitive formalism, we observe crosstalk between stromal and smooth muscle cell types and structures, which are in turn isolated from epithelial and immune components that are more entwined with one another." Crosstalk generally indicates functional interactions between cell types, whereas the data only can indicate co-associations. The authors can strengthen the case for crosstalk by looking for an enrichment of ligand-receptor interactions within the cells of a neighborhood or community (leveraging their scRNA-Seq data).

We thank the reviewer for this suggestion. We performed our colocalization analysis for cell-cell pairs that were enriched within the *follicle* community. This resulted in 57 cell pairs colocalized, such as the CD8+ T cell-CD4+ T cell pair, each of the pairs having a large list of potential receptor-ligand interactions (Supplementary Fig. 25c).

We also matched CODEX and snRNAseq at the single cell level to look at differentially expressed genes within CODEX-defined multicellular neighborhoods. See response to Reviewer #4 question 3C.

46. When introducing the single cell RNA-Seq and ATAC-Seq analysis in the text, it was unclear that the 10x Multiome assay was applied. I surmised this from the methods, but this should be clarified in the text.

We thank the reviewer for highlighting that this was not clear from the main text. We actually performed separate scRNA and scATAC experiments for samples from three of the donors and the multiome assay on samples from one of the donors in the original submission. We now include the multiome assay performed on five additional donors. We have clarified this in the main text and explicitly state that the multiome assay was performed on samples from six donors and separate scATAC and scRNA experiments were performed on samples from three of the donors. We have also made this clear in Figure 4A that both of these assays were performed.

a) Sections of the intestine from which cells were isolated for separate snRNA-seq and snATAC-seq experiments or multiome experiments.

47. Single nucleus RNA-Seq often has high levels of ambient RNA detected, confounding analysis. Was the level of ambient RNA ever assessed, perhaps by looking at the specificity in expression of lineage markers, and were any methods run to correct ambient RNA or remove empty droplets beyond what is incorporated in Cellranger?

We now run decontX to correct for ambient RNA contamination.

48. Please include in the data availability statement what kind of data is available (raw, processed, images) and whether there are portals to visualize and query the final datasets (spatial tissue maps, UMAP embeddings).

We have now added in the data availability the type of data that is available. This includes a link to the HuBMAP portal where all raw datasets are stored that can be downloaded and also viewed in a processed state. We also provide processed and annotated CODEX datasets via Dryad repository.

49. When doing differential expression analysis for genes expressed in colon vs small bowel (e.g. Fig 4G,H and Supp Fig 16), the 4 tissue donors should be treated as the independent replicates and not the individual single cells. Using Wilcox with single cells as the replicates leads to over inflated p-values. A pseudobulk DE gene method would be appropriate: Confronting false discoveries in single-cell differential expression (Nature Communications, 2021), muscat detects subpopulation-specific state transitions from multi-sample multi-condition single-cell transcriptomics data (Nature Communications, 2020).

We now added plots that average the gene expression per donor, cells, and expression of the two genes that follow the same trend by the single-cell analysis. One donor (B006) had very few (5) plasma cells compared to the other donors and is also with lower expression profiles in the summed figure.

50. When introducing LD score regression, the authors should provide explanation as to what confounding biases (cryptic relatedness, population stratification) this method controls for.

We have added “a method that aims to distinguish heritability from confounding factors such as population stratification and cryptic relatedness” when we introduce LD score regression.

51. The authors run LD score regression using the chromatin accessibility data. Did they also try using the gene expression data? Is there a way to combine results from analyses done using both modalities?

We attempted to use the gene expression data for the LD score regression, but when applying a similar approach we saw some qualitative similarities, but none of the cell types met significance after correction for multiple hypothesis testing. The results of the LD score regression for RNA is shown below, with the log10 uncorrected p values set as the color bar.

Referee #2 (Remarks to the Author):

In this manuscript Hickey et al. present a spatially-resolved cell atlas of the human gastrointestinal tract. They use both single nucleus RNAseq and single cell ATACseq to identify cell states and regulators, as well as CODEX multiplexed imaging, to identify neighborhoods of co-localized cell types. This forms a comprehensive resource that will serve as an important reference map for future human studies. The amount of collected data is impressive. The manuscript is very well written and manages to highlight several interesting and novel findings in a highly approachable manner. I particularly liked the spatial analyses that highlighted some intriguing differences between the small and large intestine including the differences in CD8+ cell abundances, the differential localization of immune cells in the crypt apex/villus vs. crypt base, as well as the unbiased identification of regulators along differentiation trajectories. I have a few, generally minor comments that should be addressed in a revision:

1) The spatial analysis of Figure 3 is very interesting, however the authors should add graphs showing the differential localization of distinct cell types in the crypt/villus zone in the small intestine and in the crypt base/crypt apex zones in the large intestine. For example, are CD8+ cells, CD4+ cells, neutrophils and dendritic cells more abundant in the villus compared to the crypt in the small intestine? Similarly for the plasma cell co-localization with transit amplifying cells, it would be helpful to show this more explicitly, ideally with some blow-up images of example crypt/villus zones.

We thank the reviewer for these suggestions. We have worked in our revision to focus the analysis around layering of the intestine and also crypt structures. We had several approaches to this:

1. We looked at concentric neighboring cells surrounding paneth cells identified in the small intestine. This revealed that adaptive immune cells such as CD8+ T cells, CD4+ T cells, and ICCs are enriched in the intestinal crypt areas as shown quantitatively and qualitatively (staining) by Fig. 1 K-N:

2. We also performed our neighborhood analysis on each individual tissue section separately and found a unique *Neuroendocrine Enriched* neighborhood that is only found in the colon (Fig. 1G). General density analysis of neuroendocrine cells had a higher density in the colon in general (Fig. 1H) and compared to other epithelial cell neighborhoods (Fig. 1I). Furthermore, the density of neuroendocrine cells decreased as with increasing maturity of the

epithelial cell neighborhood and was found to be at the base of the crypt and share similar composition to the small intestine crypt with enriched adaptive immune cells (Fig. 1J).

3. We further analyzed the organization of the mucosa by looking at the community spatial context map and described in detail these changes within the text. Briefly it demonstrated the communities and layering of these communities throughout the epithelium. We have provided magnified versions of the community, neighborhood, and cell type maps corresponding to these quantitative spatial context maps, as well as a percentage plot of immune, stromal, and epithelial cell types that are found across these communities in Fig. 3G-M

4. We also now provide a magnified image of the plasma cell and TA interaction within the colon shown in Supplementary Fig. 25a:

2)As the authors note, secretory cell fractions increase dramatically from the small to the large intestine. A major open question is what dictates these differential differentiation outcomes – are there inherent differences in the stem cells between the two segments, or rather differences in niche signals? Crypt stem cell differentiation into secretory or absorptive cells, in mice at least, depends on notch activity – notch active cells differentiate into absorptive cells whereas notch-low cells differentiate into secretory cells. Do the authors find differential activity of Notch-related pathway TFs such as Hes1 and Atoh1 between the small and large intestinal stem cells? Are there other differences in the stem cell TF activity profiles that could prime them to this skewed differentiation towards secretory cells in the large intestine?

This is an interesting question, and we have looked more into the differences between the stem cells in different regions of the intestine. We first identified motif enrichment in differential peaks between stem cells in the colon and stem cells in the small intestine. Interestingly, the notch pathway TFs ATOH1 and HES1 motifs were not enriched in peaks that are significantly more accessible in colon stem cells compared to small intestine stem cells (see a in figure below). In general, HOX and FOX motifs were the most enriched motifs in peaks that are significantly more accessible in colon stem cells. It is unclear if the activity of these TFs is somehow biasing cells in the colon toward the secretory lineage. We do nominate several FOX TFs, including FOXA2, FOXA3, and FOXP1 as a potential master regulators of goblet cells, and perhaps the increased accessibility at FOX motifs in colon stem cells biases toward the secretory lineage by activating genes controlled by these fox TFs in goblet cells. We do observe higher TF motif activity scores for FOXA3, in colon stem cells than stem cells from the other regions (b). When we look at which of these FOX TFs is actually expressed in stem cells, we see the greatest expression of FOXP1 (c).

We have added the figure above and the following text to the manuscript:

“We noted above that goblet cells are more abundant in the colon when compared to the small intestine, but it is unclear what biases cells to differentiate into goblet cells with greater frequency in the colon. Possible causes include differences in signaling in the crypts vs differences in stem cells between the regions. We identified motifs enriched in differential peaks between small intestine and colon stem cells and found that FOX motifs are enriched in peaks more accessible in colon stem cells (Supplemental Figure 26a). Similarly, FOX TFs have greater motif deviation scores in colon stem cells when compared to small intestine stem cells (Supplemental Figure 26b). Given that we nominate several FOX TFs as potential regulators of goblet cells, the increased FOX activity may partially explain why goblet cells are more abundant in the colon.”

3)The authors make a distinction between CD57 enterocytes, CD66 enterocytes and another class of enterocytes in the CODEX data, with differing frequencies along the small-to-large intestinal axis. It would be helpful to show the expression of CD57 and CD66 mRNAs on the single cell atlas of Figure 4. Do these sub-types of enterocytes have differential gene expression programs or crypt-villus localization patterns?

The CD66 antibody binds several genes, including CD66a, CD66c, CD66d, and CD66e (<https://www.google.com/url?q=https://www.fishersci.pt/shop/products/anti-cd66-clone-b1-1-bd/158766618&sa=D&source=docs&ust=1659292088999745&usg=AOvVaw1-RO3SoSPXow9O5RP4EWuz>). Plots of expression of these different CD66/CEACAM genes are shown on the plot below. In general, CEACAM1, CEACAM5, and CEACAM6 are highest expressed in mature colon cells, including both mature goblet cells and mature enterocytes. In the RNA data, we do not observe clear separation of enterocyte clusters based on expression of CD66 genes, but it is possible that the CD66 enterocytes in the CODEX data represent more mature enterocytes based on our observations in the RNA data. Given that these primarily represent the more mature cells, we would expect them to localize closer to the villi and further from the crypts, which is what we see in the CODEX data.

For CD57, we looked for expression of B3GAT1, which is a key enzyme in the synthesis of the carbohydrate epitope CD57. This gene was only expressed at low levels in the secretory clusters

in the duodenum. Similarly in the CODEX data, we only observe the presence of CD57 enterocytes in the duodenum, which are primarily present in the glands of the duodenum.

We have added the figure below and a summary of these findings to the text:

“Within the CODEX data we made the distinction between CD66+ Enterocytes and CD57+ Enterocytes. When we examined expression of these markers in the snRNA data we found that CD66+ Enterocytes typically represent mature absorptive enterocytes and are most abundant in the colon (Supplemental Figs. 21a,b). CD57+ cells are much less abundant and CD57 expression was primarily only observed at low levels in the cluster of MUC6+ cells, consistent with the observation of CD57+ cells being found in glands in the duodenum in the CODEX data (Supplemental Figs. 21a,b).”

Supplemental Figure 21: Identification of CD57+ and CD66+ cells in the snRNA data.

a) Expression of B3GAT1 and four CEACAM transcripts plotted on the UMAP manifold of epithelial cells from the duodenum, jejunum, ileum, and colon. b) Dotplot representation of the expression of B3GAT1 and four CEACAM transcripts by different epithelial cell types in different regions of the intestine.

4) L cells are usually defined by their expression of GCG (GLP1) rather than PYY.

We have added GCG to the text as a defining feature of L cells.

5) What are the differential markers of the four types of EC cells identified?

We did not observe clear differential markers and have collapsed this into a single group of enterochromaffin cells.

6) Figure 4E – please add legends, the colors are unclear.

We have updated the cell type composition figure in the main text to highlight statistically significant differences in cell type composition as suggested by reviewer 1. The new figure is reproduced below:

7) Figure 5 – on page 23 row 10 there is a typo – should be right instead of left.

This has been corrected.

8) Page 28 row 3 – ‘we analyzed many different regions’ – rephrase to tone down.

We have rephrased this to say “we analyzed eight regions”

9) I could not find the data in the portal mentioned in the data availability statement.

We have now provided information about data availability and provide more details in response to Reviewer #1 question #48.

10)Page 46, rows 27-28 – please use an objective criterion to filter out doublets.

We have clarified our approach for filtering doublets, including indicating the objective criterion used to remove cells. This process includes first simulating doublets with ArchR for ATAC data or DoubletFinder for RNA data and removing predicted doublets from the dataset and later identifying clusters of cells expressing marker genes from multiple lineages. We have summarized this with the following text:

“After quality control and filtering, doublet scores for all multiome cells and all nonmultiome snATAC cells were computed with the ArchR function `addDoubletScores` with `k=10`, `knnMethod = “UMAP”`, and `LSIMethod = 1`. An ArchR project was then created and doublets were filtered with `filterDoublets` with a filter ratio of 1.2.”

“DoubletFinder⁸⁴ was run for each nonmultiome snRNA sample using PCs 1-20. `nExp` was set to $0.08 * nCells^2 / 10000$, `pN` to 0.25, and `pK` to 0.09, and cells classified as doublets were removed prior to downstream analysis.”

“In addition to removing likely doublets based on simulating doublets as described above, we also identified clusters with expression of markers from multiple lineages (e.g. stromal and immune) during downstream clustering and annotation. For example, some cells that initially clustered with immune cells expressed higher levels of stromal genes than would be expected. For these cases we took the following approach: we first clustered all of the cells initially classified as immune cells and identified marker genes for each cluster. We then compared the marker genes to a previously published list of colon marker genes³⁵ to nominate clusters that may not contain singlet immune cells. Next, we moved these cells to the stromal or epithelial compartments, where we clustered them with all the cells initially classified as epithelial or stromal cells. In this case, if the cells had high expression of immune marker genes when compared to stromal cells, we reasoned that they were most likely immune/stromal doublets and removed the cluster of cells prior to downstream analysis.”

Referee #3 (Remarks to the Author):

The image analysis presented in the paper consists of three components:

1. Cell segmentation, using two different published tools: the CODEX Segmenter and CellVisionSegmenter.
2. Cell classification, using STELLAR, presented in a co-submitted manuscript f

3. Statistical analysis of local neighborhoods at different scales.

1) It is stated in the paper that both the CODEX Segmenter and CellVisionSegmenter are used, and references to a GitHub page are included, but no information on where and for what parts of the datasets the two respective cell segmentation methods are used, nor is there any information on how the methods were tuned and what effect faulty cell segmentation may have on the downstream analysis. All segmentation methods have their limitations, especially in parts of the tissue where cell density is high. If different methods are used on different data sets it may very much affect downstream analysis. A better description of when and where different segmentation methods were used, as well as their limitations and effect on cell classification and neighborhood analysis should be discussed.

It is worth noting that, also this paper completely lacks a description of cell segmentation approaches and the potential impact of faulty cell segmentation results on downstream cell classifications. Furthermore, a discussion on the method's sensitivity to variation in staining intensities often observed between samples is missing.

- We have clarified what samples the different segmenters were used, rationale for choosing the segmenters, changes in parameters for the segmenters, described the chance for faulty segmentation, what we do in cell data processing to account for variances in intensity, and provided additional references to segmentation in general and have pasted this new addition here for easier reference:

B001 was segmented using the CODEX Segmenter (with parameters tuned as described [1]) whereas all other donor samples were segmented using the CellVisionSegmenter. CellVisionSegmenter has been shown to work well with segmenting both dense and diffuse cellular tissues with CODEX data [2]. CellVisionSegmenter is an open-source, pre-trained nucleus segmentation and signal quantification software based on the Mask region-convolutional neural network (R-CNN) architecture. Indeed, it was designed and trained on manually annotated images from CODEX multiplexed imaging data within our own group. Consequently, the only parameter that was altered was the growth pixels of the nuclear mask which we found experimentally to work best at a value of 3. Despite this, no segmentation algorithm does a perfect job of segmentation where boundaries identified may capture portions of neighboring cells and nuclear segmentation can limit quantified signal that whole cell segmentation might be able to capture (though also imperfect from lack of consistent cell membrane stains), which can be found discussed in more detail in reviews and primary sources of segmentation [3-11]. For this reason, we performed an in-depth analysis on the different data normalization techniques and unsupervised clustering methods for robust identification of cell types in CODEX intestine data. This revealed that there is some segmentation noise that could affect cell type

identification if using manual gating, but using z-normalization, Vortex or leiden based unsupervised clustering, over-clustering the data, and manually overlaying resultant cell type clusters to the image, results in much higher fidelity cell type identification.

- [1] <https://doi.org/10.1038/s41596-021-00556-8>
- [2] <https://doi.org/10.1186/s12859-022-04570-9>
- [3] <https://doi.org/10.1038/s41598-022-08355-1>
- [4] <https://doi.org/10.1186/s12859-022-04570-9>
- [5] <https://doi.org/10.1016/j.ajpath.2021.05.022>
- [6] <https://doi.org/10.1109/MSP.2012.2204190>
- [7] <https://doi.org/10.1371/journal.pcbi.1007012>
- [8] <https://doi.org/10.1371/journal.pcbi.1005177>
- [9] <https://doi.org/10.1038/s41592-019-0403-1>
- [10] <https://doi.org/10.1038/s41598-020-61808-3>
- [11] <https://doi.org/10.1038/s41592-019-0403-1>
- [12] <https://doi.org/10.3389/fimmu.2021.727626>

2) The co-submitted paper describing STELLAR is well written and describes the combined use of per-cell marker intensity measurements and each cell's neighbors as input to a graph neural network for cell classification. It is trained using a manually annotated training set, and it is nice in the sense that it also includes a confidence measurement making it possible to find cell deviating from those present in the training set, making it possible to find new cell types.

However, STELLAR is not unique in using graph convolutional neural networks to learn latent low-dimensional cell or cell neighborhood representations that jointly capture spatial and molecular similarities. The idea has been published before:

Hu et al. "SpaGCN: Integrating gene expression, spatial location and histology to identify spatial domains and spatially variable genes by graph convolutional network." *Nature methods* 18.11 (2021): 1342-1351.

Solorzano et al. "Machine learning for cell classification and neighborhood analysis in glioma tissue." *Cytometry Part A* 99.12 (2021): 1176-1186.

Partel et al. "Spa2vec: Unsupervised representation of localized spatial gene expression signatures." *The FEBS journal* 288.6 (2021): 1859-1870.

A discussion of similarities and differences as compared to these methods is missing.

We thank the Referee for appreciating the value of our STELLAR framework and for the insightful comment about comparison to SpaGCN [1] and Spage2vec [3]. While both methods leverage spatial and molecular features using graph convolutional neural networks, these methods are unsupervised clustering methods. As such, both methods: (1) require expert assignment of clusters to corresponding cell types based on the marker genes, and (2) analyze each dataset in isolation. On the other hand, STELLAR is a semi-supervised method developed for automatic identification and discovery of novel cell types, and it works by transferring annotations across datasets. STELLAR transferred annotations across different donors of the HuBMAP data without *any manual effort*. As a semi-supervised method, the cell embeddings in STELLAR are directly optimized for the cell type annotation task forcing the learned embeddings to directly capture cell type differences which is not the case with unsupervised approaches.

Compared to [2], the annotation pipeline by Solorzano et al. is based on an ensemble of fully connected neural networks and XGBoost. In STELLAR, we have already compared our method to a fully connected neural network, ensemble methods and XGBoost (Fig 2e,f in co-submitted paper), and showed that STELLAR outperforms these methods by a large margin. For spatial cell niche characterization, authors in [2] use unsupervised method Spage2vec [3] explained above. In response to the Referee's feedback, we have added a paragraph in the Methods section that provides explanation why STELLAR is the preferred tool for annotating HuBMAP data compared to [1,2,3]. We thank the Referee and we will also include these references in our co-submitted manuscript.

[1] Hu et al. "SpaGCN: Integrating gene expression, spatial location and histology to identify spatial domains and spatially variable genes by graph convolutional network." *Nature methods* 18.11 (2021): 1342-1351.

[2] Solorzano et al. "Machine learning for cell classification and neighborhood analysis in glioma tissue." *Cytometry Part A* 99.12 (2021): 1176-1186.

[3] Partel et al. "Spage2vec: Unsupervised representation of localized spatial gene expression signatures." *The FEBS journal* 288.6 (2021): 1859-1870.

3) The final cluster approach is not novel, and has been published before in different versions, with one popular implementation provided (together with many other spatial statistics tools) in the SquidPy toolbox

Palla, G., Spitzer, H., Klein, M. et al. Squidpy: a scalable framework for spatial omics analysis. *Nat Methods* (2022). <https://doi.org/10.1038/s41592-021-01358-2>

Overall, I believe the paper as a whole is interesting and well written, and apart from a discussion of limitations of cell segmentation, the technical soundness is good. In many ways, it is a nice application of a number of known methods to a nice dataset to explore variations in cell types and cell communities in the intestine. The novelty is however limited.

We have added the provided reference and acknowledged that the clustering method is not a novel aspect to the Methods section. We chose these methods based on reliability in their ability to identify unique cell types in CODEX multiplexed data that we have studied in detail elsewhere [1]. We describe how we use Vortex and leiden-based clustering to identify cell types in the training dataset that we used to transfer cell type labels with STELLAR to other donor samples which is more novel and the method described in greater detail here [2].

[1] Hickey, J. W., Tan, Y., Nolan, G. P., & Goltsev, Y. (2021). Strategies for accurate cell type identification in CODEX multiplexed imaging data. *Frontiers in Immunology*, 3317. <https://doi.org/10.3389/fimmu.2021.727626>

[2] Brbic, M., Cao, K., Hickey, J. W., Tan, Y., Snyder, M., Nolan, G. P., & Leskovec, J. (2021). Annotation of Spatially Resolved Single-cell Data with STELLAR. *bioRxiv*. <https://doi.org/10.1101/2021.11.24.469947>

Referee #4 (Remarks to the Author):

Hickey, Becker, Nevins et al. provide a detailed single cell description of colon tissues from four patients across eight different tissue sties. Healthy tissue samples from autopsies are a rare but important resource for the understanding of human tissue and homeostasis. Utilizing CODEX multiplexed imaging they map the single cellular organization at multiple scales from neighbourhoods of ten cells to large environments and identify differences in cellular organization and immune co-localization along the colon. Further, they profiled the single cell gene regulation and expression of these regions of the colon by integrating snRNAseq and snATACseq from the same tissue samples and revealed more detailed descriptions of cellular content. By combining spatial co-localization measurements and region specific single cell sequencing, they nominate a list of receptor-ligand interactions that are involved in different cell neighbourhoods and present SEMA4D-MET as an example of plasma cell – TA2 signaling in the colon. Next, through integration of gene expression and accessible DNA binding motifs, active transcription factors were identified in different cell types and these findings expanded to focus on groups of genes with similar expression changes across cell types and differentiation pseudotime. Transcription factor motifs enriched in similarly expressed targets were used to identify putative regulators of differentiation and function. Finally, they test if disease heritability associated genes (SNP-associated) are enriched in specific cell types linking enteroendocrine cells with BMI and T cells with inflammatory diseases.

Overall, this is a detailed example of the power of a systems biology approach to map tissues which includes both spatial, transcriptome and chromatin accessibility. There is excellent usage and integration of snRNAseq and snATACseq but this has only minorly been connected to the tissue neighbourhoods and communities identified using CODEX imaging. It is a challenge to communicate such a detailed and broad study, but specific areas of clarification and presentation will help make this work more accessible to a general audience. In addition, methodological details and methods as well as the importance of making these datasets accessible as the offers have set up through the HubMap consortium are important aspects of its dissemination as, alone, the dataset is limited to a small number of patients.

Comments:

Multi-layered cell neighborhood and community analysis

1. How are cutoffs for window sizes or differences between communities determined? This would be important for determining the robustness of multi-cellular feature metrics and to enable the quantification of such features across different datasets or tissues. How is the decision made on where to cut a cluster of clusters?

a) Does altering the window size change the definition of a multi-cellular feature? Are different sized tissue structures more accurately identified at different scales? A neighborhood of 10 or 100 closest cells appears arbitrary (The answer was also not apparent in Schürch et al. 2020).

Determining window size cutoffs for cellular neighborhood analysis is an important metric to be chosen. In general, smaller window sizes will identify more local or microstructures whereas larger window sizes will lead to the identification of similarly composed structures that require a larger window size. For our neighborhood analysis here we chose to have window size cutoffs by selecting the 10 nearest neighbors around a given cell. Across many tissues we have tested this has worked well to identify conserved compositions representing a cell's immediate microenvironment or local neighbors [1-6]. This resulted in the ability to recognize microstructures such as the *Microvasculature* and also microenvironments within larger environments such as different cellular neighborhoods within the Muscularis Externa: *Smooth Muscle*, *Innervated Smooth Muscle*, and *Smooth Muscle & Innate Immune* as shown in Fig. 2B.

We chose this strategically to look at the microstructures at the neighborhood level because we also were curious to understand how these microstructures work together and come together to form macrostructures of the intestine. For this analysis we increased the number of nearest neighbors to 100. This allows us to see larger areas of cell communication and identification of macrostructures such as the *Follicle*. We further increase this further to identify Tissue Units from communities using a nearest neighbor size of 300 as can be seen from Fig. 3A-D:

In general, the size of structure does not directly relate to the window size choice, but instead relates to how compartmentalized the conserved cell types are within a given structure. For example, we can compare the *Inner Follicle* neighborhood and the *Microvasculature* neighborhoods. The *Microvasculature* neighborhoods are small structures but composed of conserved enrichment of endothelial cells and several innate immune cells. However, the *Inner Follicle* neighborhood is similarly a conserved composition of cell types within a small window size (B cells and CD4+ T cells), but this composition is conserved over large areas of the tissue as can be seen by the light blue *Inner Follicle* neighborhood seen in Fig. 3C across the intestine:

Nevertheless, we used the reviewer's suggestion to do neighborhood analysis across multiple numbers of nearest neighbors to see how the neighborhoods surrounding Paneth cells would change. Using different window sizes around paneth cells showed the local (Paneth and

Neuroendocrine enriched) environments at low numbers of the nearest neighbors and broader cell neighbors (e.g., CD8+ and CD4+ T cells) as shown in Fig. 1K-N:

We have added such discussion to the methods portion of the neighborhood analysis and also discussion of the results within the manuscript to aid better interpretation of the results.

[1] Schürch, C. M., Bhate, S. S., Barlow, G. L., Phillips, D. J., Noti, L., Zlobec, I., ... & Nolan, G. P. (2020). Coordinated cellular neighborhoods orchestrate antitumoral immunity at the colorectal cancer invasive front. *Cell*, 182(5), 1341-1359. <https://doi.org/10.1016/j.cell.2020.07.005>

[2] Bhate, S. S., Barlow, G. L., Schürch, C. M., & Nolan, G. P. (2022). Tissue schematics map the specialization of immune tissue motifs and their appropriation by tumors. *Cell Systems*, 13(2), 109-130. <https://doi.org/10.1016/j.cels.2021.09.012>

[3] Hickey, J. W., Tan, Y., Nolan, G. P., & Goltsev, Y. (2021). Strategies for accurate cell type identification in CODEX multiplexed imaging data. *Frontiers in Immunology*, 3317. <https://doi.org/10.3389/fimmu.2021.727626>

[4] Jiang, S., Chan, C. N., Rovira-Clavé, X., Chen, H., Bai, Y., Zhu, B., ... & Nolan, G. P. (2022). Combined protein and nucleic acid imaging reveals virus-dependent B cell and macrophage immunosuppression of tissue microenvironments. *Immunity*. <https://doi.org/10.1016/j.immuni.2022.03.020>

[5] Phillips, D., Matusiak, M., Gutierrez, B.R. et al. Immune cell topography predicts response to PD-1 blockade in cutaneous T cell lymphoma. *Nat Commun* 12, 6726 (2021). <https://doi.org/10.1038/s41467-021-26974-6>

[6] Brbic, M., Cao, K., Hickey, J. W., Tan, Y., Snyder, M., Nolan, G. P., & Leskovec, J. (2021). Annotation of Spatially Resolved Single-cell Data with STELLAR. *bioRxiv*. <https://doi.org/10.1101/2021.11.24.469947>

b) How do less or more prevalent cell types influence multi-cellular environment clustering? This would also appear to be dependent on the granularity of the underlying single cell clustering.

Rare cell types will influence clustering just as a unique protein on a cell will differentiate it from another cell type in clustering of the single-cell data. Consequently, we aimed to over-

cluster the neighborhoods and then merge similarly composed neighborhoods. Through testing multiple granularities, we learned that starting with 30 clusters enabled recognition of all the unique conserved 20 structures across the intestine. We clarify this now in the methods that for the final dataset we started with 30 clusters and merged down to our final uniquely named neighborhoods. We have also added this to the Methods section.

2. a) How does tissue orientation and cut impact cell or neighborhood quantification in the CODEX data? Various images are taken from different planes in the tissue resulting in images with different portions of epithelial tissue in an image or different cells across a crypt. Does this impact the proportion of cells or communities quantified in an image or the community that each cell is in? Some plots could be switched to cellular density measurements cells/area instead of as a percentage of cells.

For a detailed discussion on tissue orientation and impact on cell percentage calculations see answer to Reviewer #1 question #1. Briefly, we have 60/64 sections as representative cross-sections of the intestine and have normalized the percentages of cell types and neighborhoods by the individual compartment (Immune, Epithelial, Stromal). Since each CODEX imaging area was constant for each sample, we often opted to try to image more epithelial areas (most of the antibodies and cell types were in this area). Also, because the imaging area was fixed, cells/area would have the same issues impacting their quantification as reporting percentages across the sections (though we did calculate same-cell densities correcting for cell percentages). To address this we segmented our cell type analysis into functional *Tissue Units* which showed similar changes in percentage differences to when we separated cells out by compartment as shown in Supplemental Fig. 15

b) There are also areas of the tissue which have been damaged during processing, how are these artifacts corrected for in spatial analysis.

When we set up the imaging we attempted to select high fidelity regions with limited damage, and while there are some areas where tissue may have been damaged a majority of the tissues and tissue areas are intact and have limited damage. For areas where there are small folds of the tissue, the autofluorescence makes it unreasonable to accurately assign cell types and these cells are removed from the data at the cell type identification stage. For other issues such as tears, this will not make major differences in the statistics because we calculate a majority of the statistics with nearest neighbor methods and since our tissue sizes are so large, this could be thought of more as a separate tissue region. We have added these details to the methods of the manuscript.

3. Visualization and interpretability of tissue maps across scales

The hierarchical maps of tissue organization are of great interest but their current presentation is very difficult to interpret. A reorganization or new approach to these plots could provide much more information to a reader. Inclusion of labels and not just colors would be of great help. It would also be useful to be able to easily compare them across different settings. I understand that this is challenging data to present in this format but as a key finding of the paper an interpretable version is needed.

We thank the reviewer for the suggestion to improve the hierarchical maps of tissue organization. We have worked to improve the interpretation of this graph. We have included descriptions of both color and shape of the legend in the figure and have labels for each level of how it is connected. The graph structure is rich in data where many conclusions could be gathered from its study depending on the reader's interest. To focus the discussion on our analysis, we put a box around conditions that we provide labels for and describe within the text (*Paneth cell*, *Paneth cell enriched neighborhood*, and *Adaptive Immune Enriched Community*) to help direct the reader to the correct area of the plot as well as a red set of brackets to highlight how the stroma is separated from the mucosa as shown in Fig. 3A-F:

Also, per the reviewer's suggestion we also compare and produce the individual structures when we make the graph from only the small intestine or colon and from the 8 different sites of the intestine as shown in Supplementary Fig. 16:

Integration and validation of single cell imaging and sequencing results

3. There is excellent integration of the snRNAseq and snATACseq results but the multiplexed imaging data and sequencing are only minorly linked through the identification of interacting cells and their potential receptor-ligand pairs. Unfortunately, this analysis seems underdeveloped as it is limited to only communities that differ between bowel and colon leaving behind an opportunity to model the molecular interactions present in specific neighborhoods. This information is only listed in a small table and should be presented in the main body as it is more insightful than the numerous UMAP plots of expected cell types. As has been completed in one example, it would be fantastic if the authors were able to plot potential signaling networks in different neighborhoods or even identify cell signaling factors or responses in the single nuc data that is associated with a spatial location or interaction.

A. With such detailed phenotyping completed in the snRNAseq and snATACseq a missing link in the paper is the validation of these results using CODEX and the localization of the identified phenotypes

and signaling events to specific tissue locations. Why are mast, DCs, and neutrophils not in both datasets?

For DCs, we now identify a small number of dendritic cells in the single-nuclei data with the addition of more samples. For neutrophils, the multilobular nuclei are typically lost during the nuclei isolation steps performed prior to the single-nuclei experiments in this study, and therefore we do not expect to identify any neutrophils in the single-nuclei data. Similarly we were unable to differentiate mast cells in the CODEX data because we did not include a mast cell identifying antibody within our panel.

We also now validate receptor-ligand interaction predictions and also MUC6+ epithelial cells with CODEX data, see answer to Reviewer 1 questions 12 and 13.

B. Detailed comparison of communities or communities in different regions

We thank the reviewers for this suggestion and expanded the comparison of communities and have included similar analyses that we did for cell types and neighborhoods of the CODEX data.

- Supplementary Fig. 14A: heatmap of neighborhoods enriched in each community
- Supplementary Fig. 14B: changes in percentages of community type within each area of the intestine
- Supplementary Fig. 14C: percentage comparing small intestine and colon for each of the communities identified
- Supplementary Fig. 14D: heatmap of concentric neighborhoods enriched for Paneth enriched neighborhood

C. Very interesting work detailing the role of different transcription factors through differentiation is nicely presented. Is it feasible to profile transcriptional changes that are associated with different neighborhoods or communities? Are there changes in cell function that are associated with the identified changes in tissue organization? As it stands, the association between the neighborhoods of figures 1-3 and the single cell data of figures 4-6 remain independent biological entities.

We thank the reviewer for suggesting the coupling of CODEX neighborhood information with the snRNA-seq data. Indeed, such analysis would be very useful and of vital importance. However, associating CODEX cells to snRNA-seq cells is very challenging: while methods to match and integrate cells across different modality are available, the majority of such techniques are designed for integrating cross-sequencing modalities. Such methods generally requires hundreds, if not thousands, shared features (eg. the same gene) being measured across datasets, and thus would fail when attempting to match proteomic modalities (with

features of 30-60 VS > 1000) to sequencing modalities. We have previously developed methods for proteomic cell matching (<https://doi.org/10.1101/2021.12.03.471185>), and now have extended the methods capability to match cells (at single-cell level) from weakly correlated modalities like CODEX to scRNAseq (MaxFuse, manuscript in separate preparation). In brief, MaxFuse uses both shared (when available) and unshared features, and implements signal boosting steps (eg. graph smoothing and meta cell construction) to produce cross modality cell pairing and subsequent integration analysis. The overall matching accuracy, evaluated by the respective cell type annotations, was 92.1%. The integration result, visualized via UMAP, is shown in Figure 5A:

A HUBMAP CODEX - scRNA-seq Integration via MaxFuse

With the cross modality single cell pairing information (across snRNAseq and CODEX), we transferred the transcriptome information to individual CODEX cells, and investigated the differentially expressed genes among previously identified cellular neighborhoods. Overall, the signature genes among each cellular neighborhood were largely consistent across small bowel and the colon samples (Supplementary Figure below, left panel). In particular, we showed expression levels of DEGs involved with the follicle neighborhoods (above figure right most UMAP) and identified gene pathways enriched in these spatial organizations, including B cell receptor signaling, T cell polarity regulation, and tolerance induction of self antigen pathways (figure above, right panel).

Reviewer Reports on the First Revision:

Referees' comments:

Referee #1 (Remarks to the Author):

The authors now include 5 additional individuals to substantiate their original observations while making new ones which addresses a major concern with the original submission. In addition, they make an effort to address all other criticisms (both major and minor) and integrated the comments to fundamentally change the presentation of the data and featured conclusions for the better in the resubmission. Issues with general data presentation and statistical approaches were also well addressed.

General comments:

1. Authors improve the relevance of cell distribution analysis in figure 1 by drawing associations to metadata such as BMI and hypertension, opening interesting avenues for follow up work within the field.
2. The new observation that the reduction of CD8 T cells in individuals with a history of hypertension is enriched for the IEL compartment adds valuable insight about the composition of immune niches within the intestine
3. Ligand-receptor analysis provides some intriguing targets for future studies
4. Novel integration of CODEX with snRNA-seq data for DEG analysis is a strong addition to the manuscript

Referee #2 (Remarks to the Author):

The authors have addressed all of my previous comments. I support publication of this revised manuscript.

Referee #3 (Remarks to the Author):

The revised version of the manuscript includes a doubling of the number of individuals included in the analysis, as well as number of novel findings supported by the increased sample size. Over all, the authors have made significant improvements to the presentation and validation of the content, and thoroughly follow up on the critique by the reviewers. In the current format, I judge the work to be of general scientific interest, and I find the conclusions reliable and robust, and appropriate credit is given to previous work.

Referee #4 (Remarks to the Author):

I thank the authors for an impressive revision to address concerns and amplify the findings and impact of the study. Improved visualizations and labeling, the validation of specific findings, the more detailed analysis of spatial communities and neighborhoods and stromal/immune TFs, the submission to HUBMAP and the development of new methods to map scRNAseq to CODEX and identify a small number of spatial pathway enrichments are excellent additions. I would suggest that a plan to provide detailed methods on MaxFuse and its publication in BioRxiv and code on Github would finalize these new additions.